

# Quantifying light absorption and its source attribution of

# insoluble light-absorbing particles in Tibetan Plateau glaciers

# from 2013-2015

Xin Wang[1], Hailun Wei[1], Jun Liu[1], Baiqing Xu[2], and Mo Wang[2]

5   [1] Key Laboratory for Semi-Arid Climate Change of the Ministry of Education, College of Atmospheric
Sciences, Lanzhou University, Lanzhou, 730000, China

[2] Key Laboratory of Tibetan Environment Changes and Land Surface Processes, Institute of Tibetan
Plateau Research, Chinese Academy of Sciences, Beijing 100085, China

10   Correspondence to: X. Wang (wxin@lzu.edu.cn)



**Abstract.** Amounts of insoluble light-absorbing particles (ILAPs) deposited on the surface of snow and ice can significantly reduce the snow albedo and accelerate the snow melting process. In this study, ~67 snow/ice samples were collected in 7 high mountain glaciers over the Tibetan Plateau (TP) regions from May 2013 to October 2015. The

mixing ratio of black carbon (BC), organic carbon (OC), and mineral dust (MD) was measured using an integrating sphere/integrating sandwich spectrophotometer (ISSW) system associated with the chemical analysis by assuming the light absorption of mineral dust due to iron oxide. The results indicate that mass mixing ratios of BC, ISOC, and MD show a large variation of 10-3100 ng g$^{-1}$, 10-17000 ng g$^{-1}$, 10-3500 ng g$^{-1}$, with a mean

value of 218±397 ng g$^{-1}$, 1357±2417 ng g$^{-1}$, 241±452 ng g$^{-1}$ on TP glaciers during the entire snow field campaign, respectively. The chemical elements and the selected carbonaceous particles were also analyzed of the attributions of the particulate light absorption based on a positive matrix factorization (PMF) receptor model. On average, the industrial pollution (33.1%), biomass/biofuel burning (29.4%), and soil dust (37.5%)

were the major sources of the ILAPs in TP glaciers. Although the soil dust assumed to be the highest contributor to the mass loading of ILAPs, we noted that the averaged light absorption of BC (50.7%) and ISOC (33.2%) was largely responsible for the measured light absorption in the high mountain glaciers at the wavelengths of 450-600 nm.



## 1 Introduction

The Tibetan Plateau (TP), known as the highest plateau in the world, and its surrounding areas contain the largest snow and ice mass outside the polar regions (Qin et al., 2006). Ample evidence has indicated that the greatest decrease in length and area and the most

5 negative mass balance of high glaciers in the TP regions is associated with the deposition of black carbon (BC) over the past decade (Yao et al., 2012; Xu et al., 2009a; Xu et al., 2006). The unusual increase in temperature over the TP is now considered one of the major contributors to glacial shrinkage (Ding et al., 2006). Climate models indicated that BC heats the troposphere by absorbing solar radiation (Jacobson, 2001; Jacobi et al.,

2015), and BC reduces snow and ice albedos when it is deposited on their surface, thus leading to the acceleration of snowmelt (Hadley and Kirchstetter, 2012; Hansen and Nazarenko, 2004; Yasunari et al., 2015; Flanner et al., 2009; Flanner et al., 2007). For example, a mixing ratio of 10 ng $g^{-1}$ of BC in snow can reduce snow albedo by 1%, which has a similar effect to that of 500 ng $g^{-1}$ of mineral dust on the albedo of snow and

ice at 500 nm wavelength (Warren and Wiscombe, 1980; Warren, 1982; Wang et al., 2017).

Bond et al. (2014) indicated that the best estimate of climate forcing from BC deposition on snow and sea ice in the industrial era is +0.13 W $m^{-2}$ with 90% uncertainty bounds of +0.04 to +0.33 W $m^{-2}$. In addition to BC, organic carbon (OC) and mineral dust (MD)

also substantially contribute to springtime snowmelt and surface warming through snow darkening effects (Yasunari et al., 2015; Painter et al., 2012; Painter et al., 2010; Kaspari et al., 2014; Wang et al., 2014; Wang et al., 2013; Huang et al., 2011). Water-soluble organic carbon (WSOC) and insoluble organic carbon (ISOC) are the major components of organic carbon in the atmosphere, snow and sea ice. In addition to the strong light

absorption of ISOC and WSOC may also influence the regional and global climate through the heating and evaporating of clouds and by acting as cloud condensation nuclei (CCN) (Chen and Bond, 2010; Witkowska and Lewandowska, 2016; Alexander et al., 2012). Moreover, WSOC also plays a key role in affecting human health due to its toxic effects (McConnell and Edwards, 2008; Wang et al., 2015). Although the mass mixing

ratio of insoluble organic carbon in the snow and ice has been widely investigated in



previous studies, there are still limited studies that measure the mass mixing ratios of both WSOC and ISOC in snow/ice samples, especially across the TP regions.

Due to the importance of the climate effects by ILAPs, numerous snow surveys have been conducted to investigate the light absorption of ILAPs and their potential source

attribution in snow (Clarke and Noone, 1985; Doherty et al., 2010, 2014; Hegg et al., 2010; Huang et al., 2011). For instance, Hegg et al. (2009) found out that the BC attribution in snow are originated from two distinct biomass burning sources, a pollution source, and a marine source obtained at 36 sites in Alaska, Canada, Greenland, Russia, and the Arctic Ocean in early 2007. Huang et al. (2011) conducted the first snow survey

over northern China, and the sources of ILAPs in seasonal snow in the region were explored based on a positive matrix factorization (PMF) with backward trajectory cluster analysis (Zhang et al., 2013a). Wang et al. (2013) indicated that soil dust was found to be the major contributor to snow particulate absorption in Inner Mongolia regions and Qilian mountains over northern China. Recently, vertical profiles of ILAPs in seasonal snow

were performed from 67 north American sites, and the sources of particulate light absorption were explored based on the chemical and optical data (Doherty et al., 2014). However, the assessments of the light absorption and its emission sources of ILAPs on the TP glaciers are sparse due to limited observations. Here, we present a snow survey on collecting the snow/ice samples on 7 high mountain glaciers on the TP regions from

2013-2015. By using an integrating sphere/integrating sandwich spectrophotometer (ISSW) system associated with the chemical analysis, the particulate light absorption of BC, ISOC, and MD in TP glaciers was evaluated. Finally, the relative attribution of emission sources of the ILAPs in these regions was explored based on a positive matrix factorization (PMF) receptor model.

## 2    Site description and methods

### 2.1    Site description and sample collection

As shown in Fig. 1, the spatial distribution of aerosol optical depth (AOD) retrieved from Moderate-resolution Imaging Spectrometer (MODIS) sensors are ranging from 0.1 to 0.4

from south to north near the high mountain glacier regions over TP regions. The Qiyi




glacier (39 °14' N, 97 °45' E) is located in the eastern part of the TP above the equilibrium line altitude (ELA) at 4130 m a.s.l. The Tanggula glacier (33 °04' N, 92 °04' E) is located at 5743 m a.s.l. in the central Qinghai-Tibetan Plateau, and the average snowline in the Tanggula glacier is 5560 m a.s.l. The Yuzhufeng glacier (35 °38' N, 94 °13' E) is the

5 highest peak across the Kunlun Mountains, with an elevation of 6178 m. The Meikuang and Qiumianleiketage glaciers are also located over the Kunlun Mountains, and these glaciers have an average altitude of 5100 m and 5500 m a.s.l, respectively. The Meikuang glacier is located in the eastern Kunlun Mountains, where is characterized by alluvial deposits and sand dunes (Xiao et al., 2002). The glaciers of Hariqin and Meikuang have

10 similar altitudes but are from different mountains (Li et al., 2016). The Yangbajing glacier is located on the south-eastern margin of the Nyenchen Tanglha Mountains, and seated about 90 km northwest of Lhasa, the capital city of Tibet (Liang et al., 1995). To investigate the enrichment of ILAPs via wet and dry deposition on high glaciers, ~67 vertical profiles of snow/ice samples with seasonal transitions were obtained at 7 high

mountain sites from May 2013 to October 2015 (Fig. 1). The collected snow/ice samples were preserved in 0.5-m pure, clean tubes and kept frozen at the State Key Laboratory of Cryospheric Sciences, Cold and Arid Regions Environmental and Engineering Research Institute in Lanzhou. Then, each snow/ice sample was cut vertically into small pieces from the surface to the bottom. Therefore, approximately 189 pieces of the snow/ice

samples were analyzed in this study.

## 2.2 Optical analysis

An integrating sphere/integrating sandwich spectrophotometer (ISSW) instrument that developed by Grenfell et al. (2011) was used to measure the mass mixing ratio of BC in snow by Doherty et al. (2010, 2014) and Wang et al. (2013a). By assuming the major

light absorption of mineral dust is due to iron oxides (e.g., hematite and goethite, hereinafter, simply "Fe"), it is possible to evaluate the absorption properties of OC by combining the chemical analysis and optical method (Zhou et al., 2017; Doherty et al., 2014). The following measured parameters included equivalent BC ($C_{BC}^{equiv}$), maximum BC ($C_{BC}^{max}$), estimated BC ($C_{BC}^{est}$), fraction of light absorption by non-BC ILAPs ($f_{non-BC}^{est}$),

the non-BC absorption Ångström exponent ($Å_{non-BC}$) and the absorption Ångström




exponent of all ILAPs ($\text{Å}_{tot}$), as described by Doherty et al. (2010, 2014). These parameters are defined as follows:

1. $C_{BC}^{max}$ (ng g$^{-1}$): *maximum BC* is the maximum possible BC mixing ratio in snow by assuming all light absorption is due to BC at the wavelengths of 650-700 nm.

2. $C_{BC}^{est}$ (ng g$^{-1}$): *estimated BC* is the estimated snow BC mixing ratio derived by separating the spectrally resolved total light absorption.

3. $C_{BC}^{equiv}$ (ng g$^{-1}$): *equivalent BC* is the amount of BC that would be needed to produce absorption of solar energy by all insoluble particles in snow for the wavelength-integrated from 300-750 nm.

4. $\text{Å}_{tot}$: *absorption Ångström exponent* is calculated for all insoluble particles deposited on the filter between 450 and 600 nm.

5. $\text{Å}_{non\text{-}BC}$: *non-BC absorption Ångström exponent* is defined as the light absorption by non-BC components of the insoluble particles in snow between 450-600 nm.

6. $f_{non-BC}^{est}$ (%): *fraction of light absorption by non-BC light absorbing particles* is the integrated absorption due to non-BC light absorbing particles, which is weighted by the down-welling solar flux from snow at the wavelengths of 300-750 nm.

### 2.3    Chemical analysis

Previous studies on these parameters have concluded that ILAPs are primarily derived from BC, OC, and Fe. We assume that the mass absorption coefficients (MACs) for BC, OC, and Fe are 6.3, 0.3, and 0.9 m$^2$ g$^{-1}$, respectively, at 550 nm and that the absorption Ångström exponents (Å or AAE) for BC, OC, and Fe are 1.1, 6, and 3, respectively (Grenfell et al., 2011; Doherty et al., 2010, 2014; Wang et al., 2013). Meanwhile, to quantify WSOC, about 10 ml of the filter liquor was injected into a total carbon analyzer (TOC-V, Shimadzu). The method detection limit (MDL) used was 4 μg/l with a precision of ±5% (Cong et al., 2015). The definition of TOC (Total Organic Carbon) in this study is calculated as the total WSOC measured by carbon analyzer and the ISOC calculated by ISSW instrument.

The major metallic elements (Al, Cr, Mn, Fe, Ni, Cu, Zn, Cd, Pb) were analyzed by an inductively coupled plasma-mass spectrometry (ICP-MS, X-7 Thermo Elemental) at the





Institute of Tibetan Plateau Research in Beijing. The detection limits are Al, 0.238 ng/ml; Cr, 0.075 ng/ml; Mn, 0.006 ng/ml; Fe, 4.146 ng/ml; Ni, 0.049 ng/ml; Cu, 0.054 ng/ml; Zn, 0.049 ng/ml; Cd, 0.002 ng/ml; Pb, 0.002 ng/ml. Generally speaking, we acidified all snow/ice samples to pH<2 with ultra-pure $HNO_3$, then let settle for 48h. We note that the

measurement precision is ranging from 2-10%. Details on these procedures are given in Gao et al. (2003).

Meanwhile, for the filtrated snow/ice samples, we measured the major anions ($Cl^-$, $NO_2^-$, $NO_3^-$, $SO_4^{2-}$) and cations ($Na^+$, $NH_4^+$, $K^+$, $Mg^{2+}$, $Ca^{2+}$) with an ion chromatograph using a CS12 column for cations and an AS11 column for anions at the Institute of

Tibetan Plateau Research in Beijing. All the detection limit of the ions was 1 μg/l. In addition, except for the anions and cations and trace elements, $CL_{salt}$, MD and biosmoke K ($K_{Biosmoke}$) were determined to assess the mass contributions of the major components in the snow/ice samples. $CL_{salt}$ was estimated as follows in accordance with Pio et al. (2007), by adding to sodium, chloride, and sea-salt contributions of magnesium, calcium,

potassium, and sulfate, as follows:

$$CL_{salt}=Na_{S_s}^+ + Cl^- + Mg_{S_s}^{2+} + Ca_{S_s}^{2+} + K_{S_s}^+ + SO_{4S_s}^{2-}$$
$$=Na_{S_s}^+ + Cl^- + 0.12Na_{S_s}^+ + 0.038Na_{S_s}^+ + 0.038Na_{S_s}^+ + 0.25Na_{S_s}^+ \quad (1)$$

$$Na_{S_s}= Na_{Total} - Al \cdot (Na/Al)_{Crust} \quad (2)$$

Where $(Na/Al)_{Crust}=0.33$ , and represents the Na/Al ratio in the dust materials

(Wedepohl, 1995). With 0.12, 0.038, 0.038, and 0.25 being the mass rations in seawater of magnesium to sodium, calcium to sodium, as well as potassium to sodium and sulfate to sodium, respectively.

The MD content was calculated by a straightforward method, and the Al concentration in dust was estimated at 7% (Zhang et al., 2013b):

$$MD=Al/0.07 \quad (3)$$

We determined $K_{Biosmoke}$ as follows (Pu et al., 2017):

$$K_{Biosmoke}=K_{Total} - K_{Dust} - K_{Ss} \quad (4)$$

$$K_{Dust}=Al \cdot (K/Al)_{Crust} \quad (5)$$

$$K_{Ss}=Na_{Ss} \cdot 0.038 \quad (6)$$





Where $(K/Al)_{Crust}$ is 0.37 and represents the K/Al ratio in the dust materials (Wedepohl, 1995) and $Na_{Ss}$ is estimated by Eq. (2).

### 2.4 Enrichment factor (EF)

To evaluate the relative contributions of trace elements from natural (e.g., mineral and soil dust) versus anthropogenic sources, an inter-annual comparison of $EF_c$ values, which represent the enrichment of a given element relative to its concentration in the crust of the earth. The primary uncertainty in these calculations is attributed to the differences between chemical compositions in the snow and the reference crustal composition. The $EF_c$ is defined as the concentration ratio of a given metal to that of Al, which is a reliable measure of crustal dust, normalized to the same concentration ratio characteristic of the upper continental crust (Wedepohl, 1995), calculated with the following equation:

$$EF_c = \frac{(X/Al)_{snow}}{(X/Al)_{crust}} \qquad (7)$$

### 2.5 Source apportionment

The Positive Matrix Factorization (PMF 5.0) is considered as a generally accepted receptor model to determine source apportionment of the ILAPs when source emission profiles are unknown (Paatero and Tapper, 1994). Details of the PMF procedure used in this study are also similar to the previous work as discussed in Hegg et al. (2009, 2010). Generally, the mass concentration of the chemical species and the uncertainty were used as the input. The final data set used for the PMF analysis contained 189 samples with 18 elements whereby only elements that have high recovery were used. The uncertainty value of each variable in each sample estimated from an empirical equation. The PMF model was run for 3 to 6 factors with 6 random seeds, but only a three-factor solution of the ILAPs in TP glaciers could provide the most meaningful results. $Q$ values (modified values) for the 3-factor solution (both robust and true) were closest to the theoretical $Q$ value of any of the factor numbers for which the model was run, suggesting that the 3-factor solution was optimal.



## 3. Results

### 3.1 Regional averages

Over 67 snow/ice samples were collected at 7 sites from 2013 to 2015 across the Tibetan

plateau field campaign. Each vertical snow/ice sample was cut into several pieces. The

general information of $C_{BC}^{est}$, $C_{BC}^{max}$, $C_{BC}^{equiv}$, $f_{non-BC}^{est}$, $Å_{tot}$, and $Å_{non-BC}$ of the snow/ice

samples are given in Table 1 for each glacier. The lower median values of $C_{BC}^{est}$ could be

found in the Tanggula, Hariqin, and Yangbajing glaciers on the south of the TP regions,

while the other glaciers shows a relative higher range (94-172 ng g⁻¹) on the north edge of

10 the TP regions. Details of the vertical profiles of all snow/ice samples collected in each

site could also be found in Table S1. During the field campaign from 2013-2015, the

lowest concentration of BC in the snow/ice samples is found in the Tanggula glacier, with

a value of $C_{BC}^{est}$ ~10 ng g⁻¹. In contrast, the highest values of $C_{BC}^{est}$, $C_{BC}^{max}$, and $C_{BC}^{equiv}$ are

3100 ng g⁻¹, 3600 ng g⁻¹, and 4700 ng g⁻¹, respectively, taken in the Yangbajing region.

We note that there are no apparent differences in ILAPs between the cold and warm

seasons in high glacier regions.

It is well known that the aerosol composition and the size distribution are key parameters

that affect the absorption Ångström exponent. Doherty et al. (2010) reported that the

value of the absorption Ångström exponent of OC was close to 5, which is consistent

with previous studies with values ranging from 4-6 (Kirchstetter et al., 2004). Several

studies indicated that the absorption Ångström exponent of mineral dust ranged from 2 to

5 (Fialho et al., 2005; Lafon et al., 2006). The variation in the absorption Ångström

exponents for urban and industrial fossil fuel emissions is typically in the range of 1.0-1.5

(Millikan, 1961; Bergstrom et al., 2007), which is slightly lower than that of biomass

25 burning, which primarily falls in the range of 1.5-2.5 (Kirchstetter et al., 2004; Bergstrom

et al., 2007). Although the source attribution of the insoluble light-absorbing particles in

the samples is not a dominant determinant of the value of the absorption Ångström

exponent, fossil fuel burning may have a lower absorption Ångström exponent (<2) than

2-5 (Fialho et al., 2005; Millikan, 1961). Generally, the median of the absorption

30 Ångström exponent for total particulate constituents ($Å_{tot}$) exceeds 1.0 at all locations (Fig.



2). As shown in Fig. 2a, the lowest median value of $Å_{tot}$ (~2.1) is found in the Tanggula glacier, while the other glaciers exhibit much higher values (2.5-2.9). The results indicated that the emission of the ILAPs in the Tanggula glacier likely originated from the combustion sources, which is also consistent with the previous studies (Bond et al.,

1999, 2001; Bergstrom et al., 2007; Schnaiter et al., 2003, 2005; Clarke et al., 2007). Except Hariqin glacier, the other glaciers show an increased trend of the absorption Ångström exponent for non-BC particulate constituents ($Å_{non-BC}$) from the south to north regions in the TP regions (Fig. 2b). $Å_{tot}$ and $Å_{non-BC}$ for all snow/ice samples were in the range of 1.4-3.7 and 1.9-5.8, respectively (Table S1). The lower absorption Ångström

exponent ($Å_{tot}$<2) found in the Meikuang (site 30), Tanggula (sites 51, 56, 57, 58) and Yangbajing glaciers (site 65) suggested that the sites were primarily influenced by fossil fuel emission, whereas the other sites were heavily influenced by soil dust and biomass burning. Another notable feature is that a lower $Å_{non-BC}$ suggests a higher percentage of mineral dust in all snow/ice samples, while a higher $Å_{non-BC}$ reflects that the non-BC

ILAPs in snow and ice were mainly dominated by OC (Wang et al., 2013; see Eq. 3). Histograms of the absorption Ångström exponent by region are shown in Fig. 3. In the Yuzhufeng and Tanggula glaciers, there is a large variation of the absorption Ångström exponent (~1-4), reflecting that the ILAPs are not only dominated by BC in these regions but also influenced by non-BC absorbers such as OC and mineral dust. In contrast, a

common feature in the other regions is that they show a less variable of the absorption Ångström exponent, ranging from 2.5-3.

BC and other ILAPs are integrated into the snowpack and ice surface by dry and wet deposition, such as gravity, turbulence, and precipitation. For instance, Flanner et al. (2012) indicated that BC nucleates ice very poorly via direct deposition of vapor, so most

relevant mechanisms involve liquid water. Qi et al. (2017) exhibited that the major process of wet scavenging is in-cloud scavenging, which occurs in two stages: aerosol activation to form cloud droplets, and removal of droplets by precipitation. Therefore, the investigation of the mixing ratios of ILAPs in each glacier could be useful to analyze the emission sources of the air pollutants. As shown in Fig. 4, a notable feather is that there

are large biases between the median and the average values of the concentration of ILAPs



in snow/ice in each glacier. Due to all of the snow/ice samples were collected in the individual period from 2013-2015, there are large variations of the concentrations of the ILAPs in snow/ice samples. Therefore, we note that median values are more representative in each glacier. The $C_{BC}^{est}$ and $C_{ISOC}$ both show a significant decreasing

trend from the Qiyi glacier to the Yangbajing glacier. We note that the relative higher values of the $C_{BC}^{est}$ and $C_{ISOC}$ in the Qiyi, Qiumianleiketage, Yuzhufeng, and Meikuang glaciers are due to heavy human activities than that in the Hariqin, Tanggula and Yangbajing glaciers, where are shown as the cleanest regions in the TP regions. The result also has a good agreement with the previous study (Ming et al., 2013, their Fig. 3).

Doherty et al. (2010, 2014) and Wang et al. (2013) indicated that the light absorption of Fe across the Northern Hemisphere is mainly originated from the local soil dust. Therefore, the concentrations of Fe in snow/ice samples in each glacier are exhibited in Fig. 4c.

The Yuzhufeng glacier is located in the northern part of the Loess Plateau, close to the

Meikuang glacier. Twelve snow/ice samples were collected in the Yuzhufeng glacier, and the depth of these samples ranged from 15-45 cm (Table S1). As shown in Fig. S2, most values of $C_{BC}^{est}$ in this region range from ~100-1000 ng g$^{-1}$, with a few values lower than 100 ng g$^{-1}$. One notable feature is that the highest concentrations of $C_{BC}^{equiv}$ and $C_{BC}^{max}$ for the surface layer are 2600 ng g$^{-1}$ and 1600 ng g$^{-1}$, respectively, at site 41. while the

concentration of ISOC and $Å_{tot}$ are ~8600 ng g$^{-1}$ and 3.41. Based on the non-BC fraction of the light absorption ($f_{non-BC}^{est}$) is 0.56, we pointed out that the light absorption in the surface glacier at site 41 wasn't only influenced by BC, but also possibly related to the ISOC, and MD. The large variation of the absorption Ångström exponent distribution is an indication of the complicated emission sources of ILAPs in the snow/ice samples. In

the Yuzhufeng glacier, $Å_{tot}$ generally varied between ~2 and 3.7, and the average value of $f_{non-BC}^{est}$ is close to 50%, so these results also reveal that the ILAPs in snow/ice samples are heavily influenced by anthropogenic air pollutants. Large variations of ISOC (measured by ISSW) and WSOC are also observed, with values ranging from ~10-17000 ng g$^{-1}$ and ~410-43000 ng g$^{-1}$, respectively. Except for site 23, the values of $C_{BC}^{est}$ in the

Meikuang glacier are much lower than those in the Yuzhufeng glacier, with values





ranging from ~20-670 ng g$^{-1}$ and with a median value of 130 ng g$^{-1}$. The median value of the mass concentration of ISOC is ~600 ng g$^{-1}$ in the Meikuang glacier. The fraction of total particulate light absorption due to non-BC constituents is typically ~16-62%, and $\mathring{A}_{non\text{-}BC}$ (5.12) in this region is highly similar to that found in the Yuzhufeng glacier (5.06).

This suggests that the ILAPs have similar emission sources in the Yuzhufeng and Meikuang glaciers. In addition, there appears to be no significant difference in the mixing ratios of ILAPs in the snow/ice samples via dry and wet deposition in these glaciers between the cold and warm seasons (Fig. S2, S3, S4, S5, S6).

In the Qiyi glacier (Fig. S4), the $C_{BC}^{est.}$ are much similar than those in the Meikuang

glacier, with values ranging from ~20-720 ng g$^{-1}$, which does not include the highest value of 1900 ng g$^{-1}$ at site 13. The fraction of total particulate light absorption due to non-BC constituents $f_{non\text{-}BC}^{est}$ is typically ~20-70%, with a median value of 41%. This information along with the lower $\mathring{A}_{tot}$ (2.6) indicates that BC plays the dominant role in influencing the light absorption in this region. The vertical profiles of ILAPs in the

snow/ice samples were collected in the warm season from 2014 to 2015. In the warm season, the mixing ratios of ISOC and Fe ranged from 80-10100 ng g$^{-1}$ and 20-340 ng g$^{-1}$, respectively, and the mixing ratios of ILAPs in most of the snow/ice samples increased remarkably from the top to the bottom. This result is highly consistent with a previous study by Doherty et al. (2013). Fig. S5 shows that the vertical profiles of the mass mixing

ratios of BC, ISOC, and Fe for the snow/ice samples in the Tanggula glacier were more complicated than those for the other regions. With the exception of the surface layer at sites 53 and 54, most values of $C_{BC}^{est}$ ranged from 10 to 280 ng g$^{-1}$ in the Tanggula glacier; therefore, this glacier was the cleanest region of all the studied glaciers. At sites 52-54, a notable feature is that the surface mixing ratios of $C_{BC}^{est}$ are significantly larger than those

in the sub-surface layers, possibly because of the accumulation of BC via dry deposition on the snow/ice surface. Doherty et al. (2013) also found that the ILAPs could be scavenged with the snow meltwater, therefore leading to a much higher concentration of BC in the surface snow. At sites 56-58, $f_{non\text{-}BC}^{est}$ was lower than 38%, and $\mathring{A}_{tot}$ ranged from 1-2.5. These results are consistent with the fossil fuel combustion source due to

industrial activities. Because single layer samples are not shown, the vertical profiles of



$C_{BC}^{est}$ are plotted in Fig. S6 for all snow/ice samples, which were collected in the Qiumianleiketage, Hariqin, and Yangbajing glaciers. Except for the sites in Fig. S6d, S6e, S6i, and S6h, the other sites reveal the trapping and scavenging effects of a higher mass concentration of BC in the surface layer due to the melting process. Previous studies have

also illustrated that the ILAPs could become trapped and integrated at the surface of the snowpack due to melting and sublimation to enrich the surface concentrations (Conway et al., 2002; Doherty et al., 2013; Painter et al., 2012).

### 3.2    Relationship between BC and ISOC

The largest contribution to both BC and OC emissions is from biomass burning, but biofuel and fossil fuel burning play key roles in influencing the mass concentration of BC and OC (Bond et al., 2004, 2006; Chen et al., 2010; Akagi et al., 2011). The OC/BC ratio could also represent the relative abilities of aerosol light scattering and absorption; therefore, this ratio is an indication that the origins of the carbonaceous particles are due

to different emission sources during the transport and deposition processes (Chow et al., 1996; Turpin and Huntzicher, 1995). For instance, Bond et al. (2004) revealed that the higher ratio of OC/BC (7-8) was mainly due to open biomass burning and biofuel combustion rather than fossil fuel combustion. Koch (2009) revealed that an OC/BC ratio of ~4.0 was assumed to indicate emission sources due to fossil fuel combustion, while an

OC/BC ratio higher than 12 was due to residential coal combustion (Cao et al., 2005). Further studies have also been performed to analyze the variations in the abundance of OC and BC among sources in the atmosphere such as coal, diesel, gasoline, and wood combustion (Watson et al., 1994, 2001). However, only few studies have revealed the mass mixing ratios of OC and BC in the Loess Plateau, especially in the high glacier

regions via dry and wet deposition. As illustrated in Fig. 5, the ISOC/BC ratios range from 1 to 18 during the warm season and from 0.05 to 18.4 during the cold season, with median values of 6 and 5, respectively. High ISOC/BC ratios (>3) in all TP glaciers mainly revealed the primary emissions of carbonaceous aerosols due to fossil fuel and biofuel burning. The highest ISOC/BC ratio (18.4) found at site 52 resulted that the

emission sources is mainly due to residential coal combustion. Only slight differences





were found in the ISOC/BC ratios (6 and 5) between warm and cold seasons, respectively. The lowest values for the ISOC/BC ratios (<2) varied substantially at several sampling sites, suggesting that the natural combustion sources were the dominant factor. In order to better illustrate the spatial distribution of the mixing ratio of the ISOC/BC in snow/ice,

the median values of the mixing ratio of ISOC/BC in each glacier is given in Fig. S7. The result indicates that the relative lower mixing ratios of ISOC/BC are found in the southern glaciers, while much higher in the northern glaciers. Furthermore, the ISOC concentration exhibited a strong correlation with BC (y=4.350x+343.80, $R^2$=0.54) (Fig. 6), which suggested influences from similar emission sources (e.g., biomass burning and

residential and commercial coal combustion).

### 3.3 Water-soluble organic carbon

It is well known that the WSOC component has a large variation ranging from 20% to 99% of total carbonaceous particles (Mayol-Bracero et al., 2002; Saxena and Hildemann,

1996). There is a very large fraction representing the average WSOC (>80%) to TOC, which can absorb solar light and enhance cloud formation through their direct and indirect climate effects. These results are highly consistent with previous studies showing that fossil fuel combustion plays a key role in leading to the higher fraction of WSOC to total organic carbon (TOC) (Andreae and Rosenfeld, 2008). The concentration of WSOC

during the field campaign varied from 400 to 43600 ng g$^{-1}$, with a median of 1400 ng g$^{-1}$ (Table 1). The median values of the WSOC/TOC ratio were 0.85 for the warm season and 0.9 for the cold season during this field campaign (Fig. 7). No remarkable correlation was observed between WSOC and ISOC in both cold and warm seasons. As WSOC is considered as a stable indicator of the primary biomass burning, with a small contribution

of fossil fuel combustion, we indicate that a mixture emission sources from the biomass burning and fossil fuel combustion might be contributing significantly to the TOC concentrations in these TP glacier regions.

### 3.4 Contributions to particulate light absorption by ILAPs



BC and OC were mainly emitted from biomass burning and biofuel and fossil fuel combustion, while mineral dust was emitted from the local soil or desert regions (Bond et al., 2013; Kaspari et al., 2014; Painter et al., 2007; Chen and Bond, 2010; Streets et al., 2001; Pu et al., 2017). The contributions to particulate light absorption by BC, OC, and

Fe have been investigated using the ISSW measurements across northern China and North America (Wang et al., 2013; Doherty et al., 2014). The fractional contributions to absorption by BC, ISOC, and Fe at 450 nm in surface high glaciers are shown in Fig. 8, and the concentrations of BC, ISOC, and Fe are given in Table S1. BC plays a dominant role in particulate light absorption with values ranging from 20-87% in all high glacier

regions. ISOC is the second highest absorber in high glacier regions, and there are large variations of light absorption of ISOC during the field campaign with values ranging from 0.5-58%. The light absorption due to ILAPs in the TP glacier regions is not only from industrial and biomass burning but also with a small contribution from local soil dust. Note that the median fraction of light absorption due to Fe is ~13%, with the highest

light absorption of iron being higher than 40% in the Yangbajing glacier. This result is an indication that mineral dust plays a key role in affecting the spectral absorption properties due to the soil and mineral dust at site 67. Although the total light absorption was dominated by BC and ISOC in all selected glaciers, we note that the relative spatial distribution of the total light absorption due to Fe (>17%) also play key roles to affected

the snowmelt in the southern glaciers than that of northern glaciers across the TP regions (Fig. S8).

### 3.5    Enrichment factor (EF)

Briefly, EF values ranging from 0.1 to 10 indicate significant input from crustal sources.

Conversely, EF values that larger than 10 exhibit a major contribution from anthropogenic activities. Referring to the EF analysis (Fig. 9), the mean EF of Fe < 5 in each glacier can be assumed to customarily originate from crustal sources. Recent studies have also indicated that light-absorbing particles in snow are dominated by local soil dust in some typical regions over northern China (Wang et al., 2013), and northern America

(Doherty et al., 2014). Comparable with Fe, the other trace metals with the mean EF of



$\geqslant 5.0$ were moderately to highly enriched predominantly from anthropogenic emissions (Hsu et al., 2010). For example, Pacyna (2001) reported that fossil fuel combustion is a major source of Cr. Cu primarily originates from emissions from fossil fuel combustion and industrial processes, while Pb and Zn are known to be drawn from the traffic-related

activities and coal burning (Contini et al., 2014; Christian et al., 2010). Together with high EF values observed for Cu, Zn, and Cd in our snow/ice samples clearly suggested that the TP glaciers have already been polluted by human activities, such as biomass burning, fossil fuel burning, and the coal burning. For instance, high level of Pb and Cr have demonstrated a link to coal combustion (Mokhtar et al., 2014; Zhang et al., 2013a,

2013b), abundant Cu and Cd were associated with traffic-related dusts (Cheng et al., 2010). Therefore, we concluded that the natural dust source and anthropogenic emission source are both non-negligible to the ILAPs in the TP glaciers.

### 3.6     Source apportionment

Given the importance of the climate effect in our understanding of the ILAPs in TP glaciers, we present a PMF receptor model to analyze the source attribution of ILAPs in these glaciers. In this study, two datasets including the mass concentrations of the chemical components and the ILAPs in snow/ice and the associated uncertainty datasets were used to run the PMF 5.0 model. The details of the techniques have already been

illustrated by Hegg et al. (2009, 2010) and Pu et al. (2017). The factor loadings (apportionment of species mass to individual factors) for the 3-factor solution of the source profiles based on the PMF 5.0 model are given in Fig. 10 (in both measured mass concentration and the % total mass allocated to each factor). It is evident that the first factor (top panel) was obviously characterized by high loadings of $Cl^-$, $CL_{salt}$,  $SO_4^{2-}$,

and $NO_3^-$, which are well known markers for the urban or local industrial pollutions (Alexander et al., 2015). Although $Cl^-$ to $Na^+$ are usually considered as a potential product of emission source of sea salt, but also a high loading of $Cl^-$ to $CL_{salt}$ indicated another source in addition to sea salt such as industrial emission and coal combustion (Hailin et al., 2008; Kulkarni, 2009). Additionally, the highest loading of $NH_4^+$ is also

suggested as an indicator of coal combustion (pang et al., 2007). Compared with the first



factor, the highest loading of Al (90.3%) and Fe (87.3%) are well-known markers for the urban or regional soil dust (Pu et al., 2017). Therefore, the second factor or source profile is easily interpretable as a natural soil dust source. But a notable feature is that the relative high mass loading of $C_{BC}^{max}$ (76.4%) to that of the previously identified soil dust

source as reported by Pu et al. (2017). It is well known that $K^+$ and $K_{Biosmoke}$ are the major indicators of biomass burning source (Zhang et al., 2013a). Therefore, it is easily interpretable that the highest loadings of $K^+$ and $K_{Biosmoke}$ are well representative the biomass burning source (Fig. 10c). However, it is also important to note that the lowest mass loading of $C_{BC}^{max}$ in this factor is a bit unexpected. Indeed, the $C_{BC}^{max}$ is not only

attributed to the biomass burning emission, but also associated with the industrial activities associated with the local soil dust (Bond et al., 2006). Therefore, we interpreted the third factor normally considered a predominantly biomass burning product. However, the major emission of BC in TP glaciers originated from the local soil source instead of the biomass burning and industrial pollution than previous studies (Pu et al., 2017; Zhang

et al., 2013a).

Finally, the chemical composition and mean source apportionment of the ILAPs to the three sources in the TP glaciers were given in Fig. 11. Note that the apportionment is of the light absorption by insoluble particles in the surface glaciers. On average, the source appointment of the ILAPs in all TP glaciers by soil dust is close to 37.5%, while the

industrial emission and biomass burning contributes 33% and 29.4%. Specifically, the largest biomass burning contribution of the light-absorption of ILAPs was found in the Qiyi glacier, which is close to the human activity regions (Guan et al., 2009). In the Meikuang, Qiumianleiketage, Yangbajing, and Tanggula glaciers, the soil dust contribution of light absorption is much larger (>47.9%) than that of industrial pollution

and biomass burning, especially in the Meikuang glacier. In these regions, the percent of the MD light absorption is ranging from 20.4-31.1%, while the light absorption by biomass burning is in the range of 18.5-35.8%. Industrial pollution constitutes a major fraction in the Yuzhufeng glacier. Chemical analysis shows that the percentages of the chemical species in the Yuzhufeng and Meikuang glaciers are much similar. The

attribution of the total anions by chloride, nitrate, and sulphate is higher 52% and 48%





than the other chemical species in the Yuzhufeng and Meikuang glaicers. In the Hariqin glacier, the largest attribution of the sulphate is up to 45.4%. As shown in Fig. 11, the source apportionment of the light absorption by insoluble particles in the surface glaciers is dominated by soil dust and the industrial pollution in most glaciers, only with a large

fraction of the light absorption due to biomass burning in the Yuzhufeng glacier. These results are highly consistent with the previous studies (Andersson et al., 2015; Li et al., 2016). They found that the contributions of coal-combustion-sourced BC are the most significant for the TP glaciers. Based on the model simulations, Zhang et al. (2015) revealed that the largest contribution to annual mean BC burden and surface deposition in

the entire TP regions is from biofuel and biomass (BB) emissions in South Asia, followed by fossil fuel (FF) emissions from South Asia.

## 4    Conclusions

In this study, the ILAPs observations in 7 high glacier regions across the Tibetan Plateau

are presented using the ISSW technique along with chemical analysis. Approximately 67 vertical profiles of snow/ice samples are analyzed during the warm and cold seasons from 2013-2015. There are no apparent differences in the mixing ratios of ILAPs in the snow/ice samples during seasonal transitions. The results indicate that the variations of $\mathring{A}_{tot}$ and $\mathring{A}_{non-BC}$ for all snow/ice samples range from 1.4-3.7 and 1.9-5.8, respectively,

excluding site 16. The lower absorption Ångström exponent ($\mathring{A}_{tot} < 2$) suggested that the sites 30, 51, and 56-58, 65 were primarily influenced by fossil fuel emission, whereas the rest of the sites were heavily influenced by mineral dust and biomass burning. Another notable feature is the large variation of the ILAPs in the snow/ice samples. By excluding some of the highest ILAPs values in the snow/ice samples, the values of $C_{BC}^{est}$, $C_{ISOC}$, and

$C_{Fe}$ range from 100-1000 ng g$^{-1}$, 10-2700 ng g$^{-1}$, and 10-1000 ng g$^{-1}$, respectively. Among the samples, the lower concentrations of BC were found in the Tanggula, Hariqin and Yangbajing glaciers, with the median concentrations of 33 ng g$^{-1}$, 24 ng g$^{-1}$, and 28 ng g$^{-1}$, respectively. The large contribution of high ISOC/BC ratios (>3) at all high glacier sites mainly indicated that the primary emissions of carbonaceous aerosols were due to

fossil fuel and biofuel burning.

BC and ISOC play dominant roles in particulate light absorption, with values ranging from 20-87% and 0.5-58%, respectively, in all high glacier regions. However, the relative spatial distribution of the total light absorption due to Mineral dust also plays key roles to affect the snowmelt in the TP glaciers. In this study, we also present a PMF receptor
5    model to analyze the source attributions of ILAPs in these glaciers. We found that the anthropogenic air pollution is much heavy across the northern glaciers due to human activities, but the major emissions of the light absorption by insoluble particles in TP glaciers originated from the local soil and industrial pollution sources, followed by the biomass burning source. We note that the $C_{BC}^{max}$ is not only attributed to the biomass
10    burning emission, but also associated with the industrial activities associated with the local soil dust. Therefore, the natural dust source and anthropogenic emission source are both non-negligible to the ILAPs in the TP glaciers.

## 5    Data availability

All datasets and codes used to produce this study can be obtained by contacting Xin Wang (wxin@lzu.edu.cn).

*Competing interests.* The authors declare that they have no conflicts of interest.

*Acknowledgements.* This research was supported by the Foundation for Innovative Research Groups of the National Natural Science Foundation of China (41521004), the
25    National Natural Science Foundation of China under grant (41775144 and 41522505), and the Fundamental Research Funds for the Central Universities (lzujbky-2015-k01, lzujbky-2016-k06 and lzujbky-2015-3).





**Table 1**. Statistics of the snow/ice variables measured using an ISSW for each glacier.

| Region | Latitude | Longitude | | $C_{BC}^{equiv}$ | $C_{BC}^{max}$ | $C_{BC}^{est}$ | $f_{non-BC}^{est}$ | $Å_{tot}$ | ISOC | WSOC | Fe |
|---|---|---|---|---|---|---|---|---|---|---|---|
| | (N) | (E) | | (ng g⁻¹) | (ng g⁻¹) | (ng g⁻¹) | (%) | | (ppm) | (ppm) | (ppb) |
| Qiyi glacier | 39°14'28" | 97°45'27" | average | 414 | 299 | 238(116, 313) | 42(15, 66) | 2.59 | 1.21 | 5.43 | 181.3 |
| | | | median | 176 | 128 | 94(29, 124) | 41(17, 70) | 2.62 | 0.66 | 2.25 | 93.94 |
| | | | minimum | 26 | 29 | 25(13, 35) | 21(—, 53) | 0.8 | 0.08 | 0.62 | 19.8 |
| | | | maximum | 2651 | 2230 | 1877(1182, 2109) | 73(41, —) | 3.73 | 11.59 | 43.55 | 2414 |
| Qiumianleiketage | 36°41'47" | 90°43'44" | average | 421 | 296 | 238(139, 402) | 44(24, 81) | 2.80 | 1.43 | 1.99 | 231.7 |
| | | | median | 307 | 215 | 172(64, 218) | 44(24, 81) | 2.76 | 1.06 | 2.23 | 184 |
| | | | minimum | 139 | 93 | 62(19, 93) | 37(12, 64) | 2.45 | 0.54 | 1.17 | 105.9 |
| | | | maximum | 995 | 662 | 558(143, 678) | 56(27, 86) | 3.08 | 3.97 | 2.56 | 625.2 |
| Meikuang glacier | 35°40'24" | 94°11'10" | average | 493 | 328 | 260(119, 331) | 42(15, 37) | 2.65 | 2.14 | 2.80 | 218.8 |
| | | | median | 197 | 156 | 133(76, 153) | 44(16, 69) | 2.64 | 0.61 | 3.05 | 125.5 |
| | | | minimum | 24 | 23 | 19(17, 24) | 16(—, 17) | 1.37 | 0.13 | 0.62 | 32.24 |
| | | | maximum | 4696 | 2817 | 2292(109, 2938) | 62(23, 85) | 3.56 | 16.89 | 8.06 | 1224 |
| Yuzhufeng glacier | 35°38'43" | 94°13'36" | average | 457 | 312 | 233(94, 295) | 51(—, 37) | 2.84 | 1.51 | 2.85 | 438.4 |
| | | | median | 317 | 201 | 160(116, 204) | 48(26, 87) | 2.95 | 1.02 | 2.76 | 212.3 |
| | | | minimum | 52 | 35 | 24(8, 35) | 15(—, 37) | 1.82 | 0.07 | 0.8 | 45.1 |
| | | | maximum | 2630 | 1608 | 1169(72, 1603) | 110(6, 49) | 3.7 | 9.16 | 5.92 | 3513 |
| Hariqin glacier | 33°08'23" | 92°05'34" | average | 476 | 327 | 256(100, 385) | 48(26, 82) | 2.79 | 1.59 | 1.67 | 171.2 |
| | | | median | 54 | 37 | 23(9, 30) | 48(26, 82) | 2.87 | 0.22 | 1.08 | 52.15 |
| | | | minimum | 36 | 24 | 13(4, 22) | 19(—, 41) | 1.96 | 0.08 | 0.72 | 26.69 |
| | | | maximum | 3990 | 2702 | 2131(682, 2784) | 64(32, 84) | 3.52 | 9.64 | 4.92 | 1049 |
| Tanggula glacier | 33°04'08" | 92°04'24" | average | 253 | 171 | 152(76, 177) | 37(15, 63) | 2.28 | 0.95 | 1.82 | 173.0 |



| Region | Latitude | Longitude | | $C_{BC}^{equiv}$ | $C_{BC}^{max}$ | $C_{BC}^{est}$ | $f_{non\text{-}BC}^{est}$ | $\mathring{A}_{tot}$ | ISOC | WSOC | Fe |
|---|---|---|---|---|---|---|---|---|---|---|---|
| | (N) | (E) | | (ng g$^{-1}$) | (ng g$^{-1}$) | (ng g$^{-1}$) | (%) | | (ppm) | (ppm) | (ppb) |
| | | | median | 62 | 47 | 53(37, 65) | 36(13, 59) | 2.18 | 0.19 | 1.41 | 62.08 |
| | | | minimum | 13 | 12 | 9(6, 18) | 8(—, 19) | 1.08 | 0.01 | 0.45 | 10.22 |
| | | | maximum | 2770 | 1849 | 1637(596, 2031) | 86(25, 90) | 3.63 | 6.97 | 5.91 | 2129 |
| Yangbajing glacier | 30°11'17" | 90°27'23" | average | 382 | 292 | 247(212, 591) | 46(16, 71) | 2.42 | 0.62 | 1.21 | 182.1 |
| | | | median | 61 | 46 | 30(19, 44) | 48(18, 75) | 2.46 | 0.13 | 0.85 | 97.99 |
| | | | minimum | 28 | 23 | 15(10, 24) | 27(7, 52) | 1.34 | 0.02 | 0.41 | 31.51 |
| | | | maximum | 4674 | 3634 | 3080(1876, 3884) | 61(26, 85) | 2.92 | 5.22 | 4.65 | 911.2 |

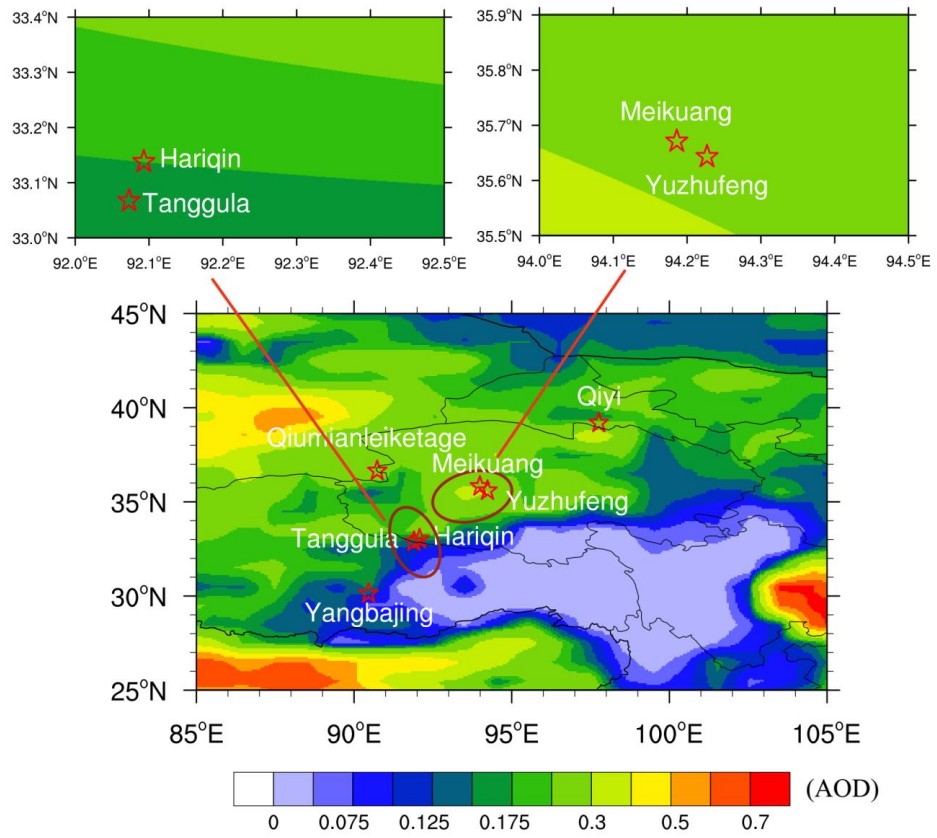

**Figure 1.** Spatial distribution of the averaged AOD retrieved from Aqua-MODIS over Tibetan Plateau

from 2013 to 2015. The red stars are the sampling locations (see also Table 1): 1, Qiyi glacier (97.76 °

E, 39.24 ° N, 4850 m a.s.l); 2, Qiumianleiketage glacier (90.73 ° E, 36.70 ° N, 5240 m a.s.l); 3,

Meikuang glacier (94.19 °E, 35.67 °N, 4983 m a.s.l); 4, Yuzhufeng glacier (94.23 °E, 35.65 °N, 5200

m a.s.l); 5, Hariqin glacier (92.09 °E, 33.14 °N, 5100 m a.s.l); 6, Tanggula glacier (92.07 °E, 33.07 °N,

5600 m a.s.l); 7, Yangbajing glacier (90.46 °E, 30.19 °N, 5655 m a.s.l).





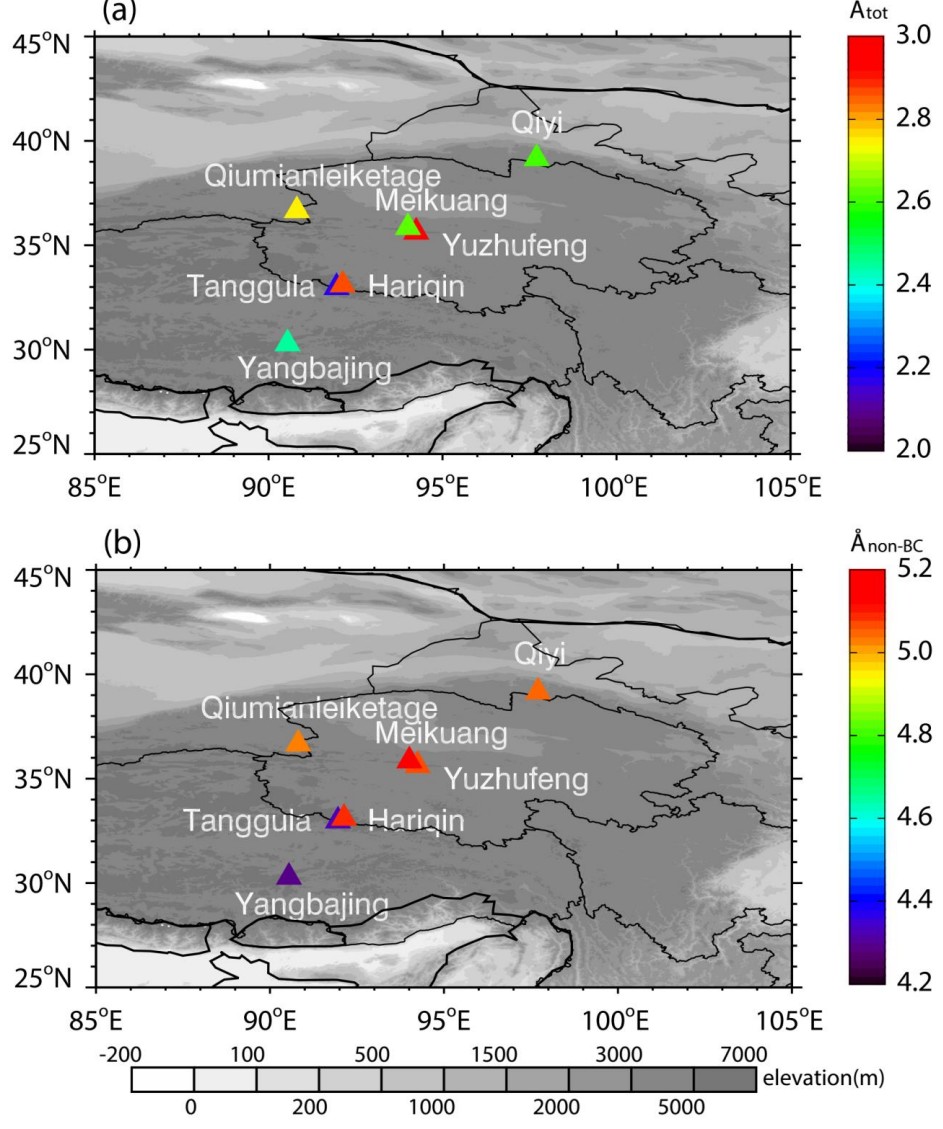

**Figure 2.** The spatial distribution of the median absorption Ångström exponent for (a) total particulate constituents ($Å_{tot}$), and (b) non-BC particulate constituents ($Å_{non-BC}$) in each glacier (see Fig. 1).



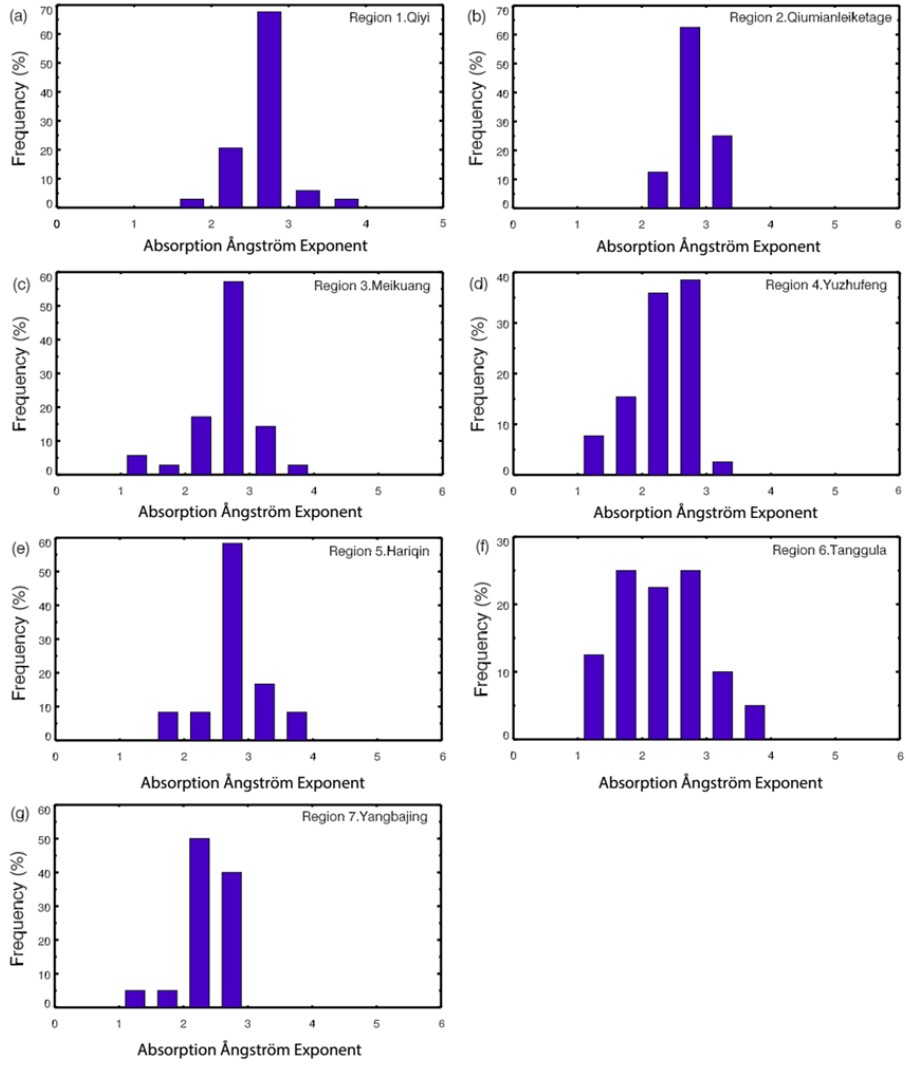

**Figure 3.** Histograms of the frequency of $\mathring{A}_{tot}$ (450-600 nm) for snow/ice samples in each of the glacier region. Samples from all vertical profiles are included.





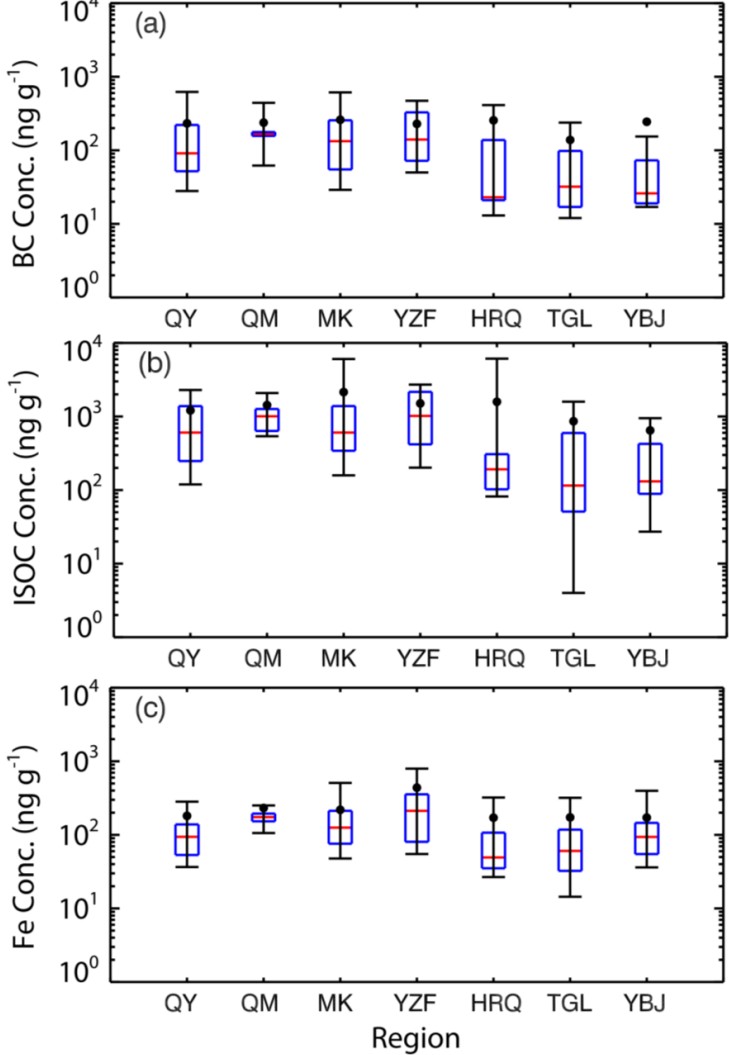

**Figure 4.** Box plots of the region variations in (a) BC concentration, (b) ISOC concentration, and (c) Fe concentration of the seven glaciers. QY, QM, MK, YZF, HRQ, TGL, and YBJ represent the following glaciers: Qiyi, Qiumianleiketage, Meikuang, Yuzhufeng, Hariqin, Tanggula, Yangbajing glaciers, respectively. Error bars are 10th, 25th, median, 75th, and 90th percentiles of the data. The dot symbol represents the average concentrations of the ILAPs in snow/ice sample in each glacier.





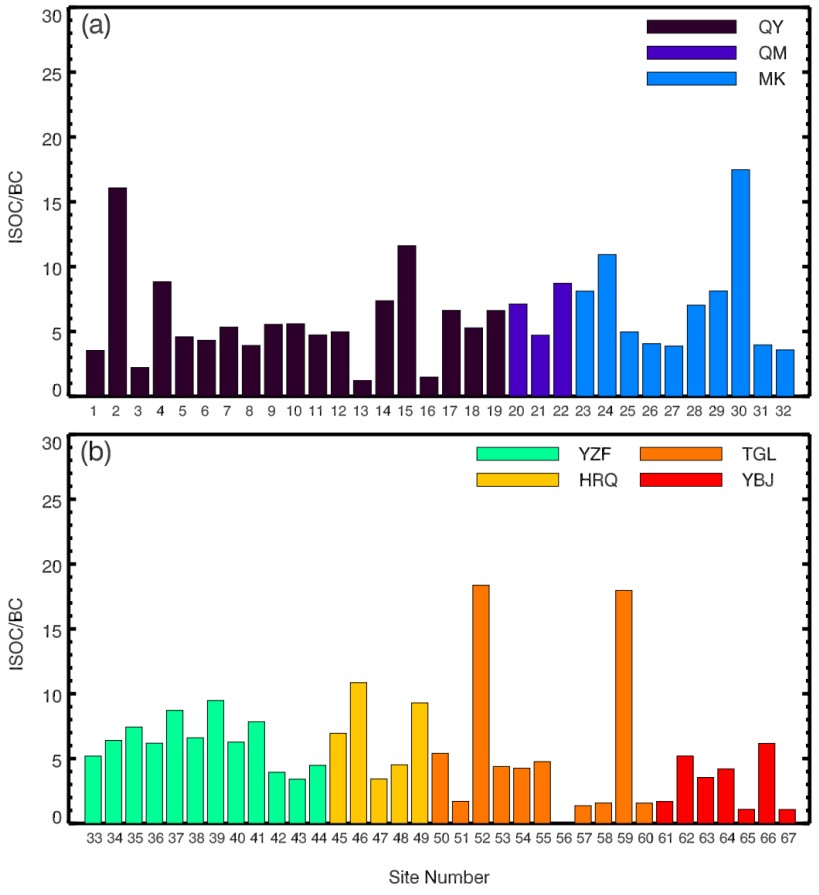

**Figure 5.** The mass mixing ratios of ISOC and BC for each surface snow/ice sample.





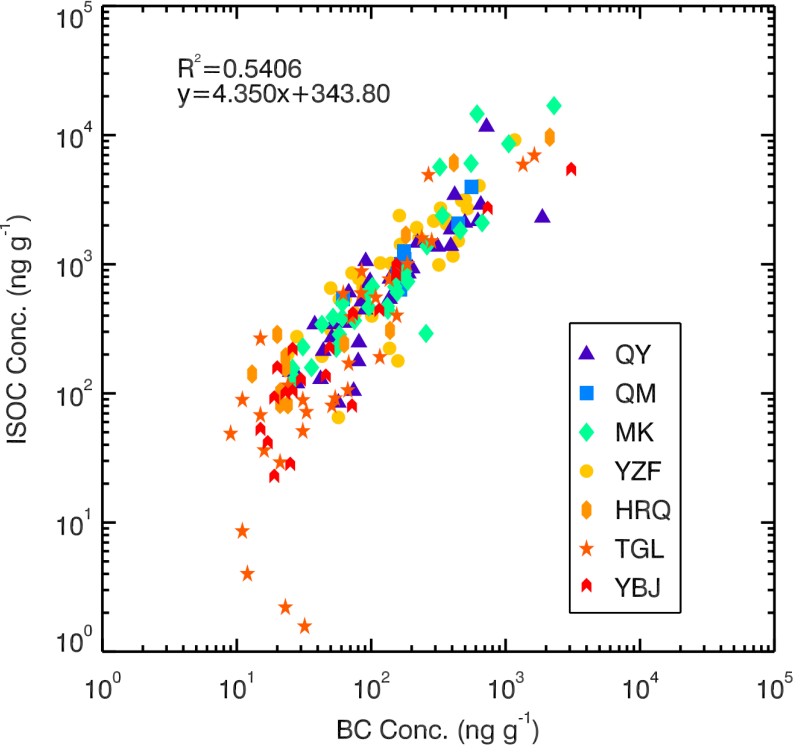

**Figure 6.** The ratios of ISOC to BC of all snow/ice samples in seven high glacier regions.





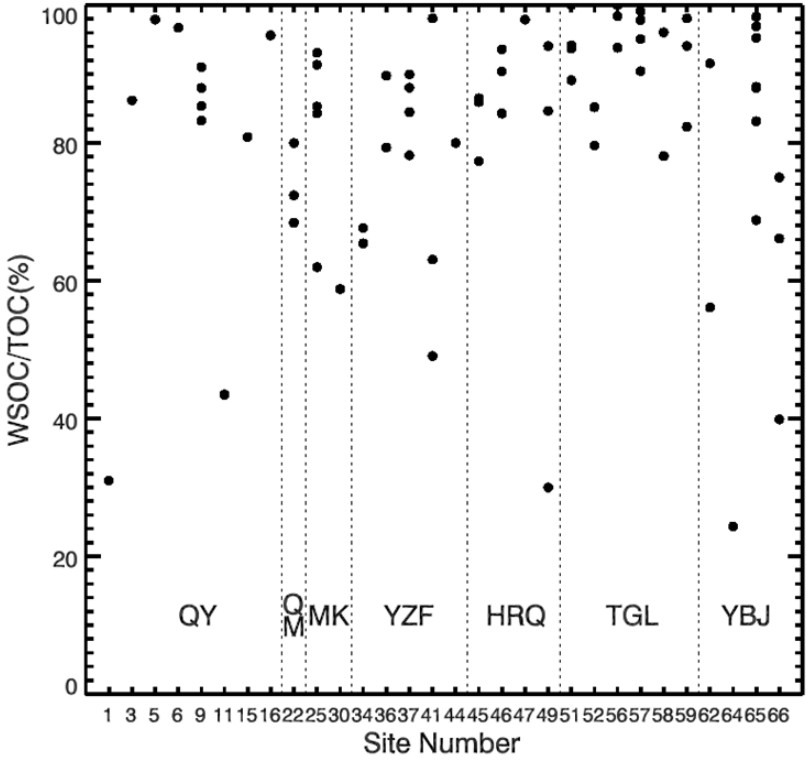

**Figure 7.** The mass fraction of WSOC to TOC of each snow/ice sample.





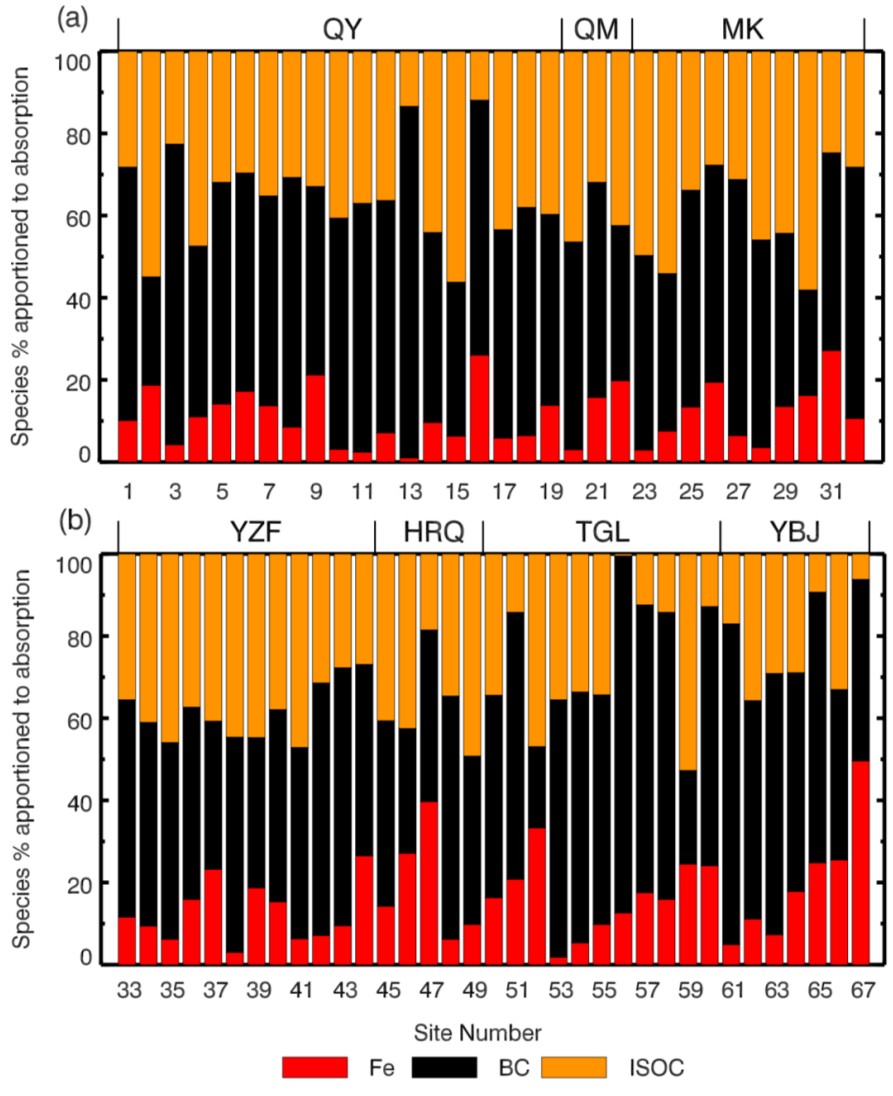

**Figure 8.** Relative contributions to total light absorption by BC, ISOC, and Fe oxide (assumed to be in the form of goethite) for surface snow/ice samples in each sampling site (See Table 1).





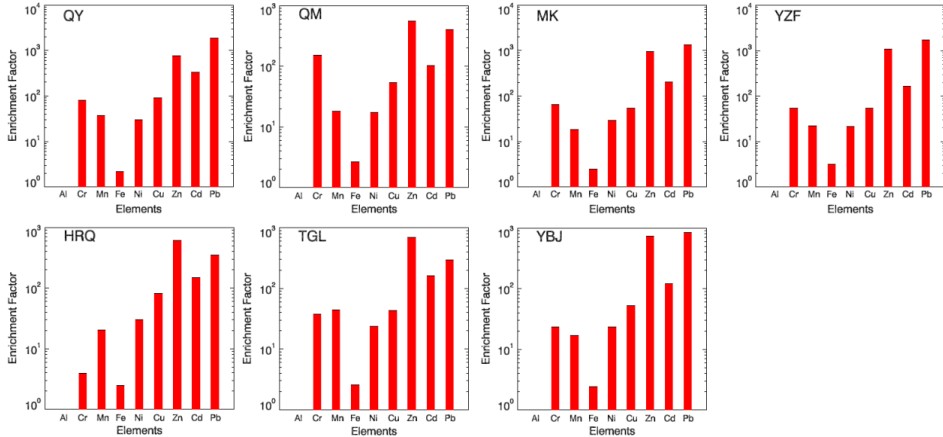

**Figure 9.** Average enrichment factors of trace metals in surface snow/ice samples at each region.



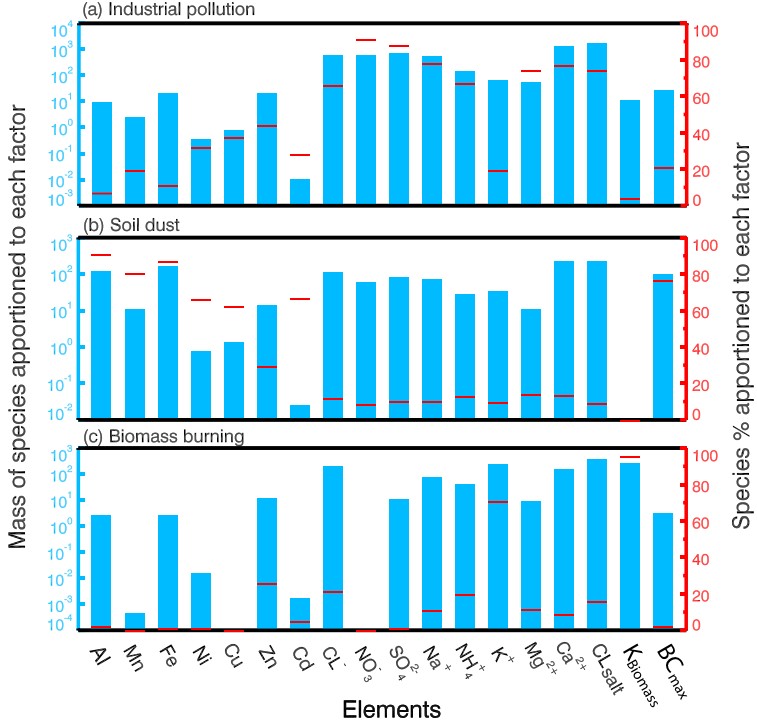

**Figure 10.** Source profiles for the three factors/sources that were resolved by the PMF 5.0 model.



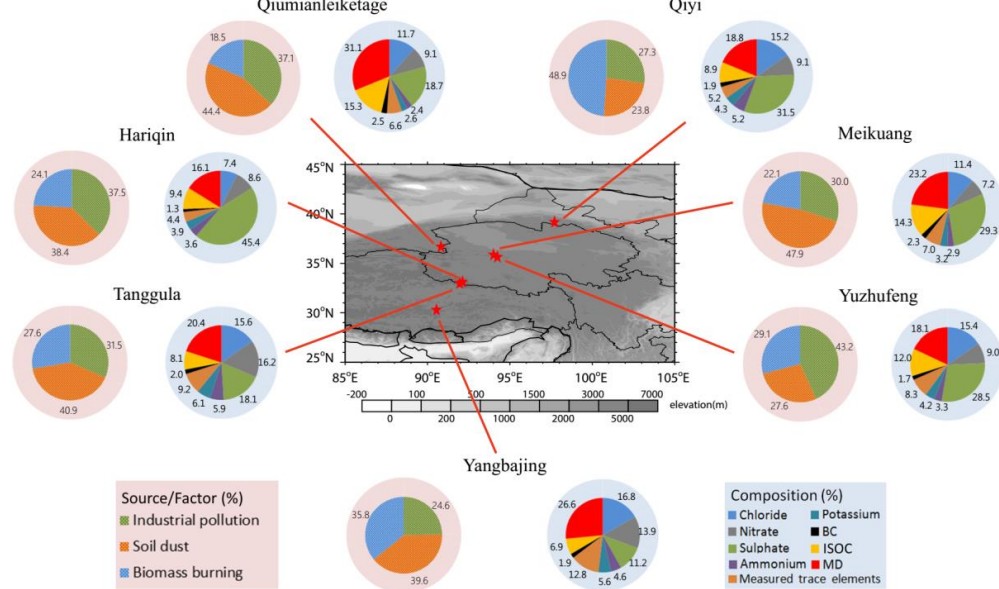

**Figure 11.** Chemical composition and source apportionment for the seven glaciers in the TP

regions. Note that the apportionment is of the light absorption by insoluble particles in the surface

glaciers.



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
