# Peer review of "Quantifying the light absorption and source attribution of insoluble light-absorbing particles on Tibetan Plateau glaciers between 2013 and 2015"

_The Cryosphere, 2018_

## Referee Comment (RC1) · Anonymous Referee #1 · 9 Jun 2018

This paper reports BC, ISOC, MD concentrations in TP glaciers using an integrating sphere/integrating sandwich spectrophotometer (ISSW). The data are valuable and I am happy to see these can be published. I have two major concerns: 1. The manuscript seems to be prepared in 2016. There are very few updated literatures were cited. Since 2017, lots of LAIs (BC, OC, MD) data measured by TOR methods (DRI) in TP glaciers have been published, and assessment of LAIs impacts on surface albedo and glacier melt has been also reported. I encourage the authors to check these references and add necessary discussion on BC and MD loadings, sources and impacts on accelerating glacier melt. Here I list some of new literatures in the region. 1. Li X., S Kang, G. Zhang, B. Que, L. Tripatheea, R. Paudyal, Z. Jing, Y. Zhang, F.

[Figure]

Yan, G. Li, X. Cui, R. Xu, Z. Hu, C. Li. 2017. Light-absorbing impurities in a southern Tibetan Plateau glacier: Variations and potential impact on snow albedo and radiative forcing. Atmospheric Research, 200: 77-87. Doi: 10.1016/j.atmosres.2017.10.002. 2. Li X. F., S. Kang, X. He, B. Qu, L. Tripathee, Z. Jing, R. Paudyal, Y. Li, Y. Zhang, F. Yan, G. Li, C. Li. 2017. Light-absorbing impurities accelerate glacier melt in the Central Tibetan Plateau. Science of the Total Environment, 587-588: 482-490. Doi: 10.1016/j.scitotenv.2017.02.169. 3. Zhang Y., S Kang, M. Sprenger, Z. Cong, T. Gao, C. Li, S. Tao, X. Li, X. Zhong, M. Xu, W. Meng, B. Neupane, X. Qin, M. Sillanpää. 2018. Black carbon and mineral dust in snow cover on the Tibetan Plateau. The Cryosphere, 12: 413-431. Doi: 10.5194/tc-12-413-2018. 4. Zhang Y. L., S. Kang, C. Li, T. Gao, Z. Cong, M. Sprenger, Y. Liu, X. Li, J. Guo, M. Sillanpää, K. Wang, J. Chen, Y. Li, S. Sun. 2017. Characteristics of black carbon in snow from Laohugou No. 12 glacier on the northern Tibetan Plateau. Science of the Total Environment: 607-608: 1237-1249. Doi: 10.1016/j.scitotenv.2017.07.100. 5. Zhang Y.L., S. Kang, Z. Cong, J. Schmale, M. Sprenger, C. Li, W. Yang, T. Gao, M. Sillanpää, X. Li, Y. Liu, P. Chen, X. Zhang. 2017. Light-absorbing impurities enhance glacier albedo reduction in the southeastern Tibetan Plateau. Journal of Geophysical Research - Atmosphere, 122. Doi: 10.1002/2016JD026397. 6. Zhang Y.L., S. Kang, M. Xu., M. Sprenger, T. Gao, Z. Cong, C. Li, J. Guo, Z. Xu, Y. Li, G. Li, X. Li, Y. Liu, H. Han. 2017. Light-absorbing impurities on Keqikaer Glacier in western Tien Shan: concentrations and potential impact on albedo reduction. Sciences in Cold and Arid Regions, 9(2): 97-111. Doi: 10.3724/SP.J.1226.2017.00097. 7. Schmale J., M. Flanner, S. Kang, M. Sprenger, Q. Zhang, J. Guo, Y. Li, M. Schwikowski, D. Farinotti. 2017. Modulation of snow reflectance and snowmelt from Central Asian glaciers by anthropogenic black carbon. Scientific Reports, 7: 40501. Doi: 10.1038/srep40501. 8. Niu H., S. Kang, Y. Zhang, X. Y. Shi, X. F. Shi, S. Wang, G. Li, X. Yan, T. Pu, Y. He. 2017. Distribution of light-absorbing impurities in snow of glacier on Mt. Yulong, southeastern Tibetan Plateau. Atmospheric Research, 197: 474-484. Doi: 10.1016/j.atmosres.2017.07.004. 9. Niu H., S. Kang, X. Shi, R. Paudyal, Y. He, G. Li, S. Wang, T. Pu, X. Shi, 2017.

In-situ measurements of light-absorbing impurities in snow of glacier on Mt. Yulong and implications for radiative forcing estimates. Science of the Total Environment. 581-582: 848-856. Doi: 10.1016/j.scitotenv.2017.01.032. 10. Ji Z., S. Kang, Q. Zhang, Z. Cong, P. Chen, M. Sillanpää. 2016. Investigation of mineral aerosols radiative effects over High Mountain Asia in 1990–2009 using a regional climate model. Atmospheric Research, 178-179: 484-496. Doi : 10.1016/j.atmosres.2016.05.003. 11. Jenkins, M., S. Kaspari, S., S. Kang., B. Grigholm, B., P.A. Mayewski. 2016. Tibetan Plateau Geladaindong black carbon ice core record (1843‒1982): Recent increases due to higher emissions and lower snow accumulation. Advances in Climate Change Research, 7(3): 132-138. Doi: 10.1016/j.accre.2016.07.002. 12. Yang J., S. Kang, Z. Ji, D. Chen. 2018. Modeling the origin of anthropogenic black carbon and its climatic effect over the Tibetan Plateau and surrounding regions. Journal of Geophysical Research: Atmospheres, 123. Doi: 10.1002/2017JD027282.

2. Geographical information is poor. Some glacier name are not correct. And sampling method and site should be represented clearly. In a glacier, you can collect samples from surface snow, surface ice and snowpit. These information are absent. I emphasize this because LAIs concentration mainly depends on which kind of samples collected in the glacier. Usually old snow and ice have higher LAIs concentrations than fresh snow and snowpit (with one or two magnitudes).

Other minor comments: 1. About the phrases in this manuscript. The author use the term of "Light-absorbing particles" in the title, while in the main text, "particulates" were used. What is the differences between these two phrases? I suggest to use the same one in the whole manuscript. In the main text, the author use the term of "High glacier", it is not a proper phrase. The author should revise these words. "Cold season and warm season" in the Tibetan Plateau which should be "monsoon season and non-monsoon season". "soil dust" should be "mineral dust".

2. Introduction: The authors mentioned "BC, OC and MD contribute to spring snowmelt and surface warming through snow darkening effects (Page 3 Line 19-21)". Then the

authors give the research progress of BC and OC, what is the role of MD in this study? The authors should give their points on MD. Page 4 Line 17-18: Check the recent literatures and the authors should point out the differences/advantages between previous studies and this study. In the Tibetan Plateau, Li et al. (2016) use the dual-carbon isotopes to distinguish the different sources of BC, which is helpful for interpretation of BC sources.

3. For the sampling: In the abstract, the author used "∼67 snow/ice samples" (Page 2 Line 3), do the authors mean about 67 snow/ice samples or more/less than 67 snow/ice samples? The author can give the exact number of samples. But then in the section 2.5 the author mentioned "189 samples". Do you mean 67 snowpits? Dis these snow samples collect from the accumulation zone of the glaciers? What is the "sites" mean in the main text? (for example in Page 10 Line11 "site65"). For the same glacier, for example Qiyi glacier, as shown in Table S1, I can't find any information for the site 1 or site 2? What is the differences between them (which part of the glacier)? And in the main text, did the authors also show the results from surface snow samples? The author need to clarify type of samples in the section 2.

Section 2.1: What is the glacier name at Tanggula Mountains? "Tanggula glacier" is not the exact name of the glacier. The same question for "Yangbajing glacier" (I think it is Gurenhekou glacier).

Section 2.1 and 2.2: When the snow/ice sample were prepared for analysis, what is the procedure on how to get the samples for ISSW analysis and WSOC analysis? What kind of filters and vials you used? Do you have blank samples? And duplicate samples?

Page 5 Line 16: "0.5-m pure, clean tubes", what is the material of this tube? What is the diameter of this tube? Do you have any photos provided in the SI?

Page 5 Line 18-20: "was cut vertically into small pieces from", please indicate a resolution.

Section 2.3: equation 2 and equation 6, there are many saline lakes on the Tibetan Plateau, Na and K may be affected by the moisture evaporated from these saline lakes. How the authors to eliminate this effects when to discuss the NaSs?

4. Section3: The author used "site 65" (Page 10 Line 11) confused me. Do the authors mean for the same glacier, the ILPS were affected by different sources? (Page10 Line 9-13)

Page 11 Line19-20, this is not a complete sentence.

Page 12 Line 25-26: this sentence is contradict with the next sentence.

Page14 Line7-8, biomass burning? Do you mean the agricultural/straw burning?

5. Section 3.6: lack of several important references to discuss the potential sources of BC. Yang et al., 2018 JGR; Li et al., 2016, Nature Communications; Zhang et al., 2018 TC.

Page17 Line12-14: "originated from the local soil source instead of the biomass burning and industrial pollution than previous studies". This sentence is contradict with the authors stated above.

Page17 Line20-22: "Qiyi glacier" is located in the northeast TP. In this region (Laohugou glacier), Li et al. (2016) indicated that 67% of BC was from fossil fuel combustion. You result is different. Why? Potential reasons? I am inclined to the dual carbon isotopes results.

Page17 Line16 to Page 18 Line 11: the authors only give the specific result of each glacier, what is the general characteristics of sources for glacier in south or north TP? The author should supplement the related references to discuss the sources.

6. Conclusions: in Page18 Line22, Page18 Line29-30, Page19 Line8, the authors mentioned the sources of ILAPs repeat.

7. Unit used in this manuscript: Page7 Line1-3, the unit should be "ng ml-1"

[Figure]

8. Figure 4: Please clear indicate the mean of red line, the blue box, and the upper and bottom black line. Figure 5: the author should define the meaning of QY QYM MK YZF TGL HRQ YBJ in the figure. In this figure, the authors can show the average ratios of ISOC to BC. The caption should change Figure 6: the relationships between ISOC and BC rather than ratios?

9. Other comments: Page7 Line 6: Gao et al. (2003) is not the proper references here. Cong et al., 2010 or Li et al., 2009 may be better. These data were measured used the same equipment in the same institute.

Page 3 Line 4-6, "most negative mass balance "'with the deposition of black carbon", I don't believe. Estimates from related studies and simulations, the contribution from BC, OC, MD can reach to about 30%.

Page 6 Line 19-20, "Previous from BC, OC and Fe" needs references.

Page 8 Line 23, "elements"? Parameters?

Page 7 line 5, delete the sentence.

Page 7 Line 10-11, the sentence should be insert into the section 2.

Page7 Line 15-16, the samples were collected in "warm season", then the results were attributed to the warm season? For snowpit, it can contain the cold season and warm season snow (non-monsoon and monsoon snow).

Page 8 Line 2-4, "with previous studies". The references here is not related to the study area (See Li et al. 2016 NC).

---

## Referee Comment (RC2) · Anonymous Referee #2 · 5 Jul 2018

Referee comments on Wang et al: Quantifying light absorption and its source attribution of insoluble light-absorbing particles in Tibetan Plateau glaciers from 2013-2015

This manuscript present interesting and valuable measurements of light-absorbing particles (LAP) from Tibetan plateau glaciers. The LAP concentrations are surprisingly high, when compared to other studies from the Tibetan plateau and the Himalaya. A strength of the manuscript is the extent of data presented. The amount of work that goes into collecting samples from such remote and harsh locations as the Tibetan plateau, often seems to be neglected in the larger community. Here the authors have performed measurements covering large areas that presumably have different sources of LAP, as well as different meteorology affecting the glaciers. Publishing these measurements would be of benefit to the public, and this should be possible after some structural and interpretational changes are made. Along with the suggested changes, the language needs to be reviewed carefully, in order to make sure that future readers will interpret the claims and results of the paper correctly. See below for both major and specific comments.

Major comments:

In the section describing the sampled glaciers (2.1) information should be added. It currently lacks crucial information such as: any estimates on the area or volume of the glaciers? Are they 'typical' valley-type glaciers? or what are the general characteristics of the glaciers? Nearby emission sources? Are the glacier fronts heavily debris covered? (See also additional specific comments given below).

In the same section I would like to see more details on the snow sample collection. For example: what was the elevation of sample locations; which ones were snow? ice? In the supplementary material I see what the cold and warm seasons correspond to, but this information is valuable to have in the manuscript itself also (In table 1 possibly).

Although the measurement method for the filters has been used previously, it would be beneficial to have a few words on the principles of the method (in section 2.2), and how measurements were carried out in practice for this manuscript. After reading Doherty et al. (2010; 2014; 2016) it is evident that the instrument have gone through modifications with time. Please provide information on how the instrument used in this manuscript is similar/different compared to Doherty et al. and if it contains the latest updates or not. Also, how did the authors take into account filter samples with a high mineral dust load? With higher load a bias can be introduced to the data (described well in Doherty et al. 2016, doi/10.1002/2015JD024375, and references therein). Further, information about the filters used should be provided since it can make a significant difference on the undercatch (look for example in Doherty et al. 2014).

In the current manuscript text the discussion on the Ångström exponent is not sufficient and I'm not entirely sure that the interpretation is correct. For the $Å_{non-BC}$ fraction, more information in section 2.2 on how it was determined is needed. For the results, I do not think that the differentiation between fossil fuel and biomass burning aerosol can be determined in the way it is done here (combined with the PMF it is possible nonetheless to distinguish fossil vs. biomass burning). I assume the authors already have studied Doherty et al.'s work, but I would urge the authors review it once more (especially Doherty et al. 2014 regarding Ångström) to guide their interpretations and text (and of course reference to what is done the same way).

Section 3.1 would benefit from more structure. In its current form, I have a hard time following in a logical order. At the moment it starts with an introduction to the concentrations of LAP in the samples, followed by statements on the Ångström exponent, and then back to more discussion on LAP concentrations. Additionally, it would be valuable to see the results obtained in this study compared in a larger Tibetan/Himalayan perspective with results from other nearby measurements of LAP in snow.

Section 3.2 needs to be majorly changed. In the references provided in this manuscript the OC/BC ratio is not used in such a way that the authors here claim. From the ratio it is not (unfortunately) as straight forward to say that a ratio of XX corresponds to the aerosol particles originating from fossil or biomass burning. As an example, what about secondary formation of organics? This section therefore needs further work, or to possibly taken out from the manuscript.

After the revisions of the manuscript results have been carried out, the conclusions should obviously be updated as well, so that they are reflecting the findings in this paper.

Specific comments:

Lines 4-6: The second sentence of the introduction almost seems contradictory to the following sentence. I believe this contradiction could be removed with careful language editing.

Lines 8-9: This is incorrect use of the Jacobi et al. 2015 reference for that statement.

Lines 17-19: Do you mean climate forcing from BC deposition world-wide? Please clarify. Also believe reference should be Bond et al. 2013, not 2014.

Lines 29-2 (page 4): Insoluble organic carbon in previous studies, please provide the references.

Line 3: The abbreviation 'ILAPs' should first be written out.

Lines 3-6: Did all of these references actually perform source attribution of the ILAP in the snow? If not, please adjust references referred to.

Lines 6-9: This sentence needs to be reworked, confusing at the moment.

Lines 14-16: What did the results of Doherty et al. 2014 show? In the previous sentences you provide some highlight from each study, but not for Doherty et al. Could be useful for readers.

Lines 17-18: At this time there has been an increasing number of observations on ILAP in Tibetan snow. The other referee provided a comprehensive list on this.

Lines 29-30: Units for the AOD numbers? And for what time period is this? Also, it would be good to include some information on what an AOD number of XX means (e.g. 0.4 meaning high optical depth, etc.)

Lines 1-2: How is the glacier located above the ELA?

Lines 2-3: The glacier must encompass an elevation range, and not only located at 5743 m a.s.l. The average snowline is based on what? Reference on this number?

Lines 4-5: Is the Yuzhufeng glacier part of the highest peak? Please clarify.

Lines 8-9: The surrounding areas characteristics, why is this information important? Is such information available for other glaciers?

Line 12: What is the point of the Liang et al. 1995 reference?

Line 15: I don't see in table S1 (or anywhere else) if the sample is ice or snow. Please provide this information.

Line 16: What was the volume of the tubes? And what kind of tubes were they?

Lines 18-20: Please provide more information on how the snow/ice samples were cut. After this (and elsewhere in the manuscript) I'm missing some description on the filtering of snow samples.

Lines 24-28: This sentence is confusing and needs to be reworked. I do not find the argument made in the references given, but once this sentence is reorganized it may be more clear.

Lines 19-20: What does this opening sentence have to do with the chemical analysis?

Lines 20-23: What does the MACs have to do with the chemical analysis? This information should instead be included in the optical analysis section (2.2.).

Line 24: Do you mean 10 mL of water from the filtered meltwater? Please clarify. In addition, a few more words describing the total carbon analyzer would be beneficial.

Lines 29-30: Was it the filtered meltwater again that was analyzed for major metallic elements?

Line 3-4: Did you acidify all samples or not? I find it confusing with the opening of the sentence 'generally speaking'.

Line 5: Measurement precision range, do you mean ±?

Line 6: What are $EF_c$ ? Please write out the abbreviation.

Line 25: What does $Q$ values stand for?

Line 1: Should not this section be referred to as Results and Discussion? And not only Results as it is now.

Line 4: I'm confused with the number of samples. Here you state over 67 and in section 2.1 ~67. Please clarify.

Line 5: The second sentence of section 3.1 is vague and not necessary for this section. As mentioned previously, please elaborate on this in the methods section.

Lines 7-10: Here the authors provide a range for the higher values, what is the range for the lower values?

Lines 15-16: This is interesting that there is no difference between seasons since several other authors have observed the contrary. This should be elaborated on in the revised manuscript.

Lines 17-29: These sentences should rather be included in the methods section, assisting with data interpretation.

Lines 2-5: Low Åtot means that the LAP originated from combustion sources?

Lines 16-21: I find it confusing with these couple of sentences here as the previous sentences discusses Ånon-BC, and the sentences before that Åtot. First discussion on Åtot, and then into Ånon-BC.

Lines 27-29: I do not understand this reasoning and how the sentences leading up to this reasoning support it.

Line 29-30: Section 3.1 started with introducing the LAP concentrations and now they come back (continued in the following sentences. The structure of this section would be better by having the concentration discussion intact.

Line 2: What do you mean by individual period?

Lines 4-5: How is a decreasing trend observed? Please include information in the manuscript that show the trend.

Line 7: How is that due to heavy human activities?

Lines 5-8: Please check this sentence structure.

Line 8-9: In what sense is there good agreement with Ming et al. 2013?

Line 16: What does the sample depth range presented here correspond to?

Line 20: In table S1 I find ISOC to be 9.16 ppm, not 8600 ng g$^{-1}$.

Lines 5-6: The previous sentences suggests that it is not similar emission sources. Please clarify.

Lines 6-8: A similar statement has been made earlier in the manuscript. Please combine these observations.

Line 17-18: Do you mean increased from the bottom to the top?

Lines 20-21: How were the samples more complicated?

Line 26-28: Would it not lead to lower LAP concentrations at the surface the way this sentence describes it now? with LAP being scavenged with meltwater?

Line 6-7: This is the wrong Conway et al reference, should be 1996 instead of 2002. Please check this also in the reference list.

Lines 15-17: Please provide a reference for this.

Lines 17-19: I do not find this to be the case in given reference.

Lines 23-27: How is that?

Lines 1-2: Is this a general statement or results of this paper? My guess is the former, and if it is that, I would place this statement in the introduction of this paper.

Line 13-15: That there is a small contribution from dust is, to my knowledge, not consistent with that results presented earlier in the manuscript. Please clarify this.

Lines 11-12: What do you mean 'for the anthropogenic emission source'? I find this sentence confusing.

As a last note, I wanted to comment on the in-text citations, are they done by year or alphabetical order? I did not find an order to this. Please check this throughout your manuscript.

---

## Author Comment (AC1) · 30 Aug 2018

Response to reviewer

We are very grateful for the reviewer's critical comments and suggestions, which have helped us improve the paper quality substantially. We have addressed all of the comments carefully as detailed below in our point-by-point responses. Our responses start with "R:".

This paper reports BC, ISOC, MD concentrations in TP glaciers using an integrating sphere/integrating sandwich spectrophotometer (ISSW). The data are valuable and I

am happy to see these can be published.

R: Thanks for all of the comments. We have carefully responded the following questions and concerns.

Issues: Comment 1: The manuscript seems to prepared in 2016. There are very few updated literatures were cited. Since 2017, lots of LAIs (BC, OC, MD) data measured by TOR methods (DRI) in TP glaciers have been published, and assessment of LAIs impacts on surface albedo and glacier melt has been also reported. I encourage the authors to check these references and add necessary discussion on BC and MD loadings, sources and impacts on accelerating glacier melt. Here I list some of new literatures in the region. 1. Li X., S Kang, G. Zhang, B. Que, L. Tripatheea, R. Paudyal, Z. Jing, Y. Zhang, F. Yan, G. Li, X. Cui, R. Xu, Z. Hu, C. Li. 2017. Light-absorbing impurities in a southern Tibetan Plateau glacier: Variations and potential impact on snow albedo and radiative forcing. Atmospheric Research, 200: 77-87. Doi: 10.1016/j.atmosres.2017.10.002. 2. Li X. F., S. Kang, X. He, B. Qu, L. Tripathee, Z. Jing, R. Paudyal, Y. Li, Y. Zhang, F. Yan, G. Li, C. Li. 2017. Light-absorbing impurities accelerate glacier melt in the Central Tibetan Plateau. Science of the Total Environment, 587-588: 482-490. Doi: 10.1016/j.scitotenv.2017.02.169. 3. Zhang Y., S Kang, M. Sprenger, Z. Cong, T. Gao, C. Li, S. Tao, X. Li, X. Zhong, M. Xu, W. Meng, B. Neupane, X. Qin, M. Sillanpää. 2018. Black carbon and mineral dust in snow cover on the Tibetan Plateau. The Cryosphere, 12: 413-431. Doi: 10.5194/tc-12-413-2018. 4. Zhang Y. L., S. Kang, C. Li, T. Gao, Z. Cong, M. Sprenger, Y. Liu, X. Li, J. Guo, M. Sillanpää, K. Wang, J. Chen, Y. Li, S. Sun. 2017. Characteristics of black carbon in snow from Laohugou No. 12 glacier on the northern Tibetan Plateau. Science of the Total Environment: 607-608: 1237-1249. Doi: 10.1016/j.scitotenv.2017.07.100. 5. Zhang Y.L., S. Kang, Z. Cong, J. Schmale, M. Sprenger, C. Li, W. Yang, T. Gao, M. Sillanpää, X. Li, Y. Liu, P. Chen, X. Zhang. 2017. Light-absorbing impurities enhance glacier albedo reduction in the southeastern Tibetan Plateau. Journal of Geophysical Research - Atmosphere, 122. Doi: 10.1002/2016JD026397. 6. Zhang Y.L., S. Kang, M.

Xu., M. Sprenger, T. Gao, Z. Cong, C. Li, J. Guo, Z. Xu, Y. Li, G. Li, X. Li, Y. Liu, H. Han. 2017. Lightabsorbing impurities on Keqikaer Glacier in western Tien Shan: concentrations and potential impact on albedo reduction. Sciences in Cold and Arid Regions, 9(2): 97-111. Doi: 10.3724/SP.J.1226.2017.00097. 7. Schmale J., M. Flanner, S. Kang, M. Sprenger, Q. Zhang, J. Guo, Y. Li, M. Schwikowski, D. Farinotti. 2017. Modulation of snow reflectance and snowmelt from Central Asian glaciers by anthropogenic black carbon. Scientific Reports, 7: 40501. Doi: 10.1038/srep40501. 8. Niu H., S. Kang, Y. Zhang, X. Y. Shi, X. F. Shi, S. Wang, G. Li, X. Yan, T. Pu, Y. He. 2017. Distribution of light-absorbing impurities in snow of glacier on Mt. Yulong, southeastern Tibetan Plateau. Atmospheric Research, 197: 474-484. Doi: 10.1016/j.atmosres.2017.07.004. 9. Niu H., S. Kang, X. Shi, R. Paudyal, Y. He, G. Li, S. Wang, T. Pu, X. Shi, 2017. In-situ measurements of light-absorbing impurities in snow of glacier on Mt. Yulong and implications for radiative forcing estimates. Science of the Total Environment. 581-582: 848-856. Doi: 10.1016/j.scitotenv.2017.01.032. 10. Ji Z., S. Kang, Q. Zhang, Z. Cong, P. Chen, M. Sillanpää. 2016. Investigation of mineral aerosols radiative effects over High Mountain Asia in 1990–2009 using a regional climate model. Atmospheric Research, 178-179: 484-496. Doi : 10.1016/j.atmosres.2016.05.003. 11. Jenkins, M., S. Kaspari, S., S. Kang., B. Grigholm, B., P.A. Mayewski. 2016. Tibetan Plateau Geladaindong black carbon ice core record (1843‒1982): Recent increases due ËŸto higher emissions and lower snow accumulation. Advances in Climate Change Research, 7(3): 132-138. Doi: 10.1016/j.accre.2016.07.002. 12. Yang J., S. Kang, Z. Ji, D. Chen. 2018. Modeling the origin of anthropogenic black carbon and its climatic effect over the Tibetan Plateau and surrounding regions. Journal of Geophysical Research: Atmospheres, 123. Doi: 10.1002/2017JD027282.

R: We have updated more relative literatures in the manuscript (included all of the references suggested by the reviewer), which were published since 2017.

Comment 2: Geographical information is poor. Some glacier names are not correct. And sampling method and site should be represented clearly. In a glacier, you can collect samples from surface snow, surface ice and snowpit. This information are absent. I emphasize this because LAIs concentration mainly depends on which kind of samples collected in the glacier. Usually old snow and ice have higher LAIs concentrations than fresh snow and snowpit (with one or two magnitudes).

R: We checked the glacier names in this study very carefully, and we have corrected the "Yangbajing and Tanggula glaciers" as "Gurenhekou and Xiaodongkemadi glaciers". In order to better illustrate the glaciers, we added a photograph as revised Figure 2 to show the major feature of the glaciers. Due to the samples were only collected approximately for each six months in the TP glaciers, we note that most of the collecting samples were ice samples, which were less than 1 m from the glacier surface (Details could be found in Table S1). We also agreed that the LAIs concentrations in the surface layer were relatively higher than that in the subsurface layers shown in Figure S2-S6, only except some dirty layer in the subsurface layer in this study.

Comment 3: About the phrases in this manuscript. The author use the term of "Light-absorbing particles" in the title, while in the main text, "particulates" were used. What are the differences between these two phrases? I suggest to use the same one in the whole manuscript. In the main text, the author use the term of "High glacier", it is not a proper phrase. The author should revise these words. "Cold season and warm season" in the Tibetan Plateau which should be "monsoon season and non-monsoon season". "soil dust" should be "mineral dust".

R: We have modified "particulates" as "particles", and removed the term of "High glacier" throughout the manuscript. We have modified the cold and warm season as "monsoon and non-monsoon season". We have also changed "soil dust" to "mineral dust".

Comment 4: Introduction: The authors mentioned "BC, OC and MD contribute to spring snowmelt and surface warming through snow darkening effects (Page 3 Line 19-21)". Then the authors give the research progress of BC and OC, what is the role of MD

in this study? The authors should give their points on MD. Page 4 Line 17-18: Check the recent literatures and the authors should point out the differences/advantages between previous studies and this study. In the Tibetan Plateau, Li et al. (2016) use the dual-carbon isotopes to distinguish the different sources of BC, which is helpful for interpretation of BC sources.

R: We have added the recent literatures in analyzing the properties of MD in the TP glaciers as follows: It is well known that the light absorption by MD is mostly related to iron oxides. For instance, the increased radiation forcing by MD in snow has affected the timing and magnitude of runoff from the Upper Colorado River Basin (Painter et al., 2007, 2010). In addition, we have added the literature of Li et al. (2016) in the introduction suggested by the reviewer as follows. Recently, Li et al. (2016) exhibited that similar contributions from fossil fuel (46±11%) and biomass (54±11%) combustion of the BC sources based on the dual-carbon isotopes technique from aerosol and snowpit samples in the TP regions. Reference Painter, T. H., Barrett, A. P., Landry, C. C., Neff, J. C., Cassidy, M. P., Lawrence, C. R., McBride, K. E., and Farmer, G. L.: Impact of disturbed desert soils on duration of mountain snow cover, Geophys. Res. Lett., 34, L12502, doi: 10.1029/2007gl030284, 2007. Painter, T. H., Deems, J. S., Belnap, J., Hamlet, A. F., Landry, C. C., and Udall, B.: Response of Colorado River runoff to dust radiative forcing in snow, P. Natl. Acad. Sci. USA, 107, 17125-17130, 2010.

Comment 5: For the sampling: In the abstract, the author used "âĹij67 snow/ice samples" (Page 2 Line 3), do the authors mean about 67 snow/ice samples or more/less than 67 snow/ice samples? The author can give the exact number of samples. But then in the section 2.5 the author mentioned "189 samples". Do you mean 67 snowpits? Is these snow samples collect from the accumulation zone of the glaciers? What is the "sites" mean in the main text? (for example in Page 10 Line 11 "site65"). For the same glacier, for example Qiyi glacier, as shown in Table S1, I can't find any information for the site 1 or site 2? What are the differences between them (which part of the glacier)?

And in the main text, did the authors also show the results from surface snow samples? The author needs to clarify type of samples in the section 2.

R: We have revised the sentence more clearly as "We collected 67 ice samples in seven glaciers on the Tibetan Plateau. Then each ice sample was cut vertically into small pieces from the surface to the bottom. Therefore, 189 pieces of the ice samples were analyzed in this study. All of the parameters for each ice samples were listed in Table S1. In order to analyze light absorption of ILAPs in the ice samples more clearly, we arranged seven glaciers from north to south according to their latitude and longitude, and samples in each glacier were sorted by sampling time. Therefore, the ice samples numbered in chronological order from 1 to 19 was in the Qiyi glacier, while sites 20-22, 23-32, 33-44, 45-49, 50-60, and 61-67 in the Qiumianleiketage, Meikuang, Yuzhufeng, Hariqin, Xiaodongkemadi, and Gurenhekou glaciers, respectively."

Comment 6: Section 2.1: What is the glacier name at Tanggula Mountains? "Tanggula glacier" is not the exact name of the glacier. The same question for "Yangbajing glacier" (I think it is Gurenhekou glacier).

R: Also see our relay to comment 2. We agree with the reviewer. Therefore, we have updated all of the glacier name very carefully throughout the manuscript, such as changing the "Yangbajing and Tanggula glaciers" to "Gurenhekou and "Xiaodongke-madi glaciers"ïïjŇrespectively.

Comment 7: Section 2.1 and 2.2: When the snow/ice samples were prepared for analysis, what is the procedure on how to get the samples for ISSW analysis and WSOC analysis? What kind of filters and vials you used? Do you have blank samples? And duplicate samples?

R: An integrating sphere/integrating sandwich spectrophotometer (ISSW) instrument that developed by Grenfell et al. (2011) was used to measure the mass mixing ratio of BC in snow by Doherty et al. (2010, 2014) and Wang et al. (2013a). This ISSW spectrophotometer measures the light attenuation spectrum from 400 to 700 nm. The total

light attenuation spectrum is extended over the full spectral range by linear extrapolation from 400 to 300 and from 700 to 750 nm (Grenfell et al., 2011). Light attenuation is nominally only sensitive to ILAPs on the filter because of the diffuse radiation field and the sandwich structure of two integrated spheres in the ISSW (Doherty et al., 2014). Meanwhile, to quantify WSOC, about 10 ml of the filter liquor was injected into a total carbon analyzer (TOC-V, Shimadzu) and the method detection limit (MDL) used was 4 $\mu$g l^(-1) with a precision of $\pm$5% (Cong et al., 2015). For filtrate processing, we used 0.2-$\mu$m nuclepore filters, as were used in Doherty et al., (2010, 2014), and Wang et al., (2013). In this study, the transmitted light detected by the system for an ice sample, $S(\lambda)$, are compared with the signal detected for a blank filter, $S\_0 (\lambda)$, and the relative attenuation (Atn) is expressed as: Atn=ln⁡[S_0 $(\lambda)$/S$(\lambda)$]

Comment 8: Page 5 Line 16: "0.5-m pure, clean tubes", what is the material of this tube? What is the diameter of this tube? Do you have any photos provided in the SI?

R: We have corrected the sentences as "The collected ice samples were preserved in 0.5-m pure clean plastic bag with a diameter of 20 cm, and kept frozen at the State Key Laboratory of Cryospheric Sciences, Cold and Arid Regions Environmental and Engineering Research Institute in Lanzhou."

Comment 9: Page 5 Line 18-20: "was cut vertically into small pieces from", please indicate a resolution.

R: See our reply to comment 5.

Comment 10: Section 2.3: equation 2 and equation 6, there are many saline lakes on the Tibetan Plateau, Na and K may be affected by the moisture evaporated from these saline lakes. How the authors to eliminate this effects when to discuss the NaSs?

R: Actually, there are several saline lakes on the Tibetan Plateau. In order to avoid the moisture evaporation, all of the ice samples were collected far from the saline lakes and the industrial areas to eliminate this effect.

[Figure]

Comment 11: Section3: The author used "site 65" (Page 10 Line 11) confused me. Do the authors mean for the same glacier, the ILAPs were affected by different sources? (Page10 Line 9-13)

R: We have changed the site number as the ice sample number to illustrated our result more clearly, which is similar with the above comment 5 (Table S1). The ice samples collected in each glacier were marked with same color or labeled with the glacier names in all of the Figures. The sentence has been deleted due to the reconstruction of this manuscript.

Comment 12: Page 11 Line 19-20, this is not a complete sentence.

R: This sentence has been rewritten as "One notable feature is that the highest concentrations of $C\_{BC}^{max}$ and ISOC for the surface layer are 1600 ng g-1 and 9160 ng g-1 at site 41.".

Comment 13: Page 12 Line 25-26: this sentence is contradict with the next sentence

R: The sentence has been rewritten as "At sites 52-54, a notable feature is that the surface mixing ratios of $C\_{BC}^{est}$ are significantly larger than those in the subsurface layers, possibly because of the accumulation of BC via dry/wet deposition on the surface samples."

Comment 14: Page 14 Line 7-8, biomass burning? Do you mean the agricultural/straw burning?

R: Biomass burning represents an important source of atmospheric air pollutants, and it is mainly the burning of agriculture/straw.

Comment 15: Section 3.6: lack of several important references to discuss the potential sources of BC. Yang et al., 2018, JGR; Li et al., 2016, Nature Communications; Zhang et al., 2018, TC.

R: We have added the recent references in section 3.5 based on the reviewer's suggestions. Referenceïij Li, C. L., Bosch, C., Kang, S. C., Andersson, A., Chen, P. F., Zhang, Q. G., Cong, Z. Y.,Chen, B., Qin, D. H., and Gustafsson, O.: Sources of black carbon to the Himalayan-Tibetan Plateau glaciers, Nat. Commun., 7, 12574, doi: 10.1038/ncomms12574, 2016. Yang J., Kang, S., Ji, Z., Chen, D.: Modeling the origin of anthropogenic black carbon and its climatic effect over the Tibetan Plateau and surrounding regions, J. Geophys. Res.-Atmos., 123, doi: 10.1002/2017JD027282, 2018. Zhang Y., Kang, S., Sprenger, M., Cong, Z., Gao, T., Li, C., Tao, S., Li, X., Zhong, X., Xu, M., Meng, W., Neupane, B., Qin, X., Sillanpää. M.: Black carbon and mineral dust in snow cover on the Tibetan Plateau, The Cryosphere, 12, 413-431, 2018.

Comment 16: Page17 Line12-14: "originated from the local soil source instead of the biomass burning and industrial pollution than previous studies". This sentence is contradict with the authors stated above.

R: According to the PMF results, we summarized the second factor as a natural mineral dust, but a notable feature is that the relative high mass loading of $"C"\_"BC"^{"max"}$ (76.4%) to that of the previously identified mineral dust source as reported by Pu et al. (2017) and Zhang et al. (2013). Indeed, the $"C"\_"BC"^{"max"}$ is not only attributed to the biomass burning emission, but also associated with the industrial activities associated with the local mineral dust (Bond et al., 2006). Compared with the previous studies, the BC emission in studied TP glaciers are also associated with the local mineral dust source instead of the biomass burning and industrial pollution.

Comment 17: Page17 Lines 20-22: "Qiyi glacier" is located in the northeast TP. In this region (Laohugou glacier), Li et al. (2016) indicated that 67% of BC was from fossil fuel combustion. You result is different. Why? Potential reasons? I am inclined to the dual carbon isotopes results.

R: Our result is much different with the previous study by Li et al. (2016). The most important reason is that we only separate the possible emission sources of the ILAPs (not only include BC, but also include OC and MD) in seven glaciers (such as biomass

burning, industrial pollution and the mineral dust) by using the PMF receptor model. But Li et al. (2016) used the dual-carbon isotopes technique to calculate the attribution of BC concentration by fossil fuel or biomass combustion. However, we thank the reviewer to provide a very useful technique to analyze the fraction of the BC emission sources for the collected glacier samples in our further study.

Comment 18: Page 17 Line 16 to Page 18 Line 11: the authors only give the specific result of each glacier, what are the general characteristics of sources for glacier in south or north TP? The author should supplement the related references to discuss the sources.

R: We have added several related references to compare with our major findings.

Comment 19: Conclusions: in Page 18 Line 22, Page 18 Line 29-30, Page 19 Line 8, the authors mentioned the sources of ILAPs repeat.

R: We reconstructed the conclusion section, and deleted the sentences in Page 18 Line 22, and Page 18 Lines 29-30.

Comment 20: Unit used in this manuscript: Page 7 Lines 1-3, the unit should be "ng ml-1"

R: We have modified "ng/ml" as "ng ml-1".

Comment 21: Figure 4: Please clear indicate the mean of red line, the blue box, and the upper and bottom black line. Figure 5: the author should define the meaning of QY QM MK YZF TGL HRQ YBJ in the figure. In this figure, the authors can show the average ratios of ISOC to BC. The caption should change Figure 6: the relationships between ISOC and BC rather than ratios?

R: We have defined the error bars shown in Figure 4 are 10th, 25th, median, 75th, and 90th percentiles of the data, while the dot symbol represents the average concentrations of the ILAPs in each glacier. We have adjusted the structure of the article and Figure 5 & 6 has been deleted.

Comment 22: Other comments: Page 7 Line 6: Gao et al. (2003) is not the proper references here. Cong et al., 2010 or Li et al., 2009 may be better. These data were measured used the same equipment in the same institute.

R: We have updated the sentence as "Details on these procedures are given in Li et al. (2009) and Cong et al. (2010)". ReferenceïijŽ Cong, Z. Y., Kang, S. C., Zhang, Y. L., and Li, X. D.: Atmospheric wet deposition of trace elements to central Tibetan Plateau, Appl. Geochem., 25, 1415-1421, 2010. Li C. L., Kang S. C., Zhang Q.: Elemental composition of Tibetan Plateau top soils and its effect on evaluating atmospheric pollution transport, Environ. Pollut., 157, 8-9, 2009.

Comment 23: Page 3 Lines 4-6, "most negative mass balance "'with the deposition of black carbon", I don't believe. Estimates from related studies and simulations, the contribution from BC, OC, MD can reach to about 30%.

R: The sentence has been rewritten as "Ample evidence has indicated that the deposition of insoluble light-absorbing particles (ILAPs) was one of the major factors (up to 30%) to lead the greatest decrease in length and area of negative mass balance in the TP glaciers over the past decade (Xu et al., 2006, 2009a; Yao et al., 2012; Qian et al., 2015; Li et al., 2017)". ReferenceïijŽ Li X., Kang, S., Zhang, G., Que, B., Tripatheea, L., Paudyal, R., Jing, Z., Zhang, Y., Yan, F., Li, G., Cui, X., Xu, R., Hu, Z., Li. C.: Light-absorbing impurities in a southern Tibetan Plateau glacier: Variations and potential impact on snow albedo and radiative forcing, Atmos., Res., 200, 77-87, 2017. Qian, Y., Yasunari, T. J., Doherty, S. J., Flanner, M. G., Lau, W. K. M., & Jing, M.: Light-absorbing particles in snow and ice: measurement and modeling of climatic and hydrological impact, Adv. Atmos. Sci., 32, 64-91, 2015. Xu, B. Q., Yao, T. D., Liu, X. Q., and Wang, N. L.: Elemental and organic carbon measurements with a two-step heating-gas chromatography system in snow samples from the Tibetan Plateau, Ann. Glaciol., 43, 257-262, 2006. Xu, B. Q., Cao, J. J., Hansen, J., Yao, T. D., Joswia, D. R., Wang, N. L., Wu, G. J., Wang, M., Zhao, H. B., Yang, W., Liu, X. Q., and He, J. Q.: Black soot and the survival of Tibetan glaciers, P. Natl. Acad. Sci. USA, 106, 22114-22118,

2009a. Yao, T. D., Thompson, L., Yang, W., Yu, W. S., Gao, Y., Guo, X. J., Yang, X. X., Duan, K. Q., Zhao, H. B., Xu, B. Q., Pu, J. C., Lu, A. X., Xiang, Y., Kattel, D. B., and Joswiak, D.: Different glacier status with atmospheric circulations in Tibetan Plateau and surroundings, Nature Climate Change, 2, 663-667, 2012. Comment 24: Page 6 Lines 19-20, "Previous from BC, OC and Fe" needs references. R: The sentence has been rewritten as "Previous studies on these parameters have concluded that ILAPs are primarily derived from BC, OC, and Fe (Qian et al., 2015; Wang et al., 2015; Yasunari et al., 2015; Pu et al., 2017)". Referenceïijǽ Pu, W., Wang, X., Wei, H. L., Zhou, Y., Shi, J. S., Hu, Z. Y., Jin, H. C., and Chen, Q. L.: Properties of black carbon and other insoluble light-absorbing particles in seasonal snow of northwestern China, The Cryosphere, 11, 1213-1233, 2017. Qian, Y., Yasunari, T. J., Doherty, S. J., Flanner, M. G., Lau, W. K. M., Ming, J., Wang, H. L., Wang, M., Warren, S. G., and Zhang, R. D.: Light-absorbing Particles in Snow and Ice: Measurement and Modeling of Climatic and Hydrological impact, Adv. Atmos. Sci., 32, 64-91, 2015. Wang, X., Pu, W., Zhang, X. Y., Ren, Y., and Huang, J. P.: Water-soluble ions and trace elements in surface snow and their potential source regions across northeastern China, Atmos. Environ., 114, 57-65, 2015. Yasunari, T. J., Koster, R. D., Lau, W. K. M., and Kim, K. M.: Impact of snow darkening via dust, black carbon, and organic carbon on boreal spring climate in the Earth system, J. Geophys. Res.-Atmos., 120, 5485-5503, 2015.

Comment 25: Page 8 Line 23, "elements"? Parameters?

R: The major chemical elements used in this study were shown in Figure 9.

Comment 26: Page 7 Line 5, delete the sentence.

R: We have deleted the sentence.

Comment 27: Page 7 Lines 10-11, the sentence should be insert into the section 2.

R: The sentence has been moved into the section 2.

Comment 28: Page7 Lines 15-16, The samples were collected in "warm season", then

the results were attributed to the warm season? For snowpit, it can contain the cold season and warm season snow (non-monsoon and monsoon snow).

R: Sorry for the misleading, we note that all of the samples collected in Qiyi glacier were collected in the monsoon season. Therefore, we have modified the sentence as "Compared with the other TP glaciers, we noted that the vertical profiles of ILAPs in the Qiyi glacier were collected in the monsoon season from 2014 to 2015 (Table S1)."

Comment 29: Page 18 Line 2-4, "with previous studies". The reference here is not related to the study area (See Li et al. 2016 NC).

R: We have updated the sentence as "This result is highly consistent with the previous study (Andersson et al., 2015)". ReferenceïijŽ Andersson, A., Deng, J., Du, K., Zheng, M., Yan, C., and Sköld, M.: Regionally-varying combustion sources of the January 2013 severe haze events over eastern China, Environ. Sci. Technol., 49, 2038-2043, 2015.

Please also note the supplement to this comment:
https://www.the-cryosphere-discuss.net/tc-2018-86/tc-2018-86-AC1-supplement.pdf

---

## Author Comment (AC2) · 30 Aug 2018

Response to reviewer 2

We are very grateful for the reviewer's critical comments and suggestions, which have helped us improve the paper quality substantially. We have addressed all of the comments carefully as detailed below in our point-by-point responses. Our responses start with "R:".

This manuscript presents interesting and valuable measurements of light-absorbing particles (LAP) from Tibetan plateau glaciers. The LAP concentrations are surprisingly

high, when compared to other studies from the Tibetan plateau and the Himalaya. A strength of the manuscript is the extent of data presented. The amount of work that goes into collecting samples from such remote and harsh locations as the Tibetan plateau often seems to be neglected in the larger community. Here the authors have performed measurements covering large areas that presumably have different sources of LAP, as well as different meteorology affecting the glaciers. Publishing these measurements would be of benefit to the public, and this should be possible after some structural and interpretational changes are made. Along with the suggested changes, the language needs to be reviewed carefully, in order to make sure that future readers will interpret the claims and results of the paper correctly. See below for both major and specific comments.

R: Thanks for all of the comments. We have carefully responded the following questions and concerns.

Issues:

Comment 1: In the section describing the sampled glaciers (2.1) information should be added. It currently lacks crucial information such as: any estimates on the area or volume of the glaciers? Are they 'typical' valley-type glaciers? or what are the general characteristics of the glaciers? Nearby emission sources? Are the glacier fronts heavily debris covered? (See also additional specific comments given below).

R: We have added some information in this section as follows: The Qiyi glacier (39°14' N, 97°45' E) is located in the eastern part of the TP, with an elevation of 6178 m. It is classified bucket-valley glacier according to its shape, and is classified subcontinental glacier according to the physical characteristics of glacier. The Xiaodongkemadi glacier (33°04' N, 92°04' E) is located in the central Qinghai-Tibetan Plateau. It is 2.8 km in length and the average snowline is 5560 m a.s.l. The annual mean air temperature at the equilibrium line altitude is in the range -5∼-7℃. The surrounding region is mainly tundra. The Yuzhufeng glacier (35°38' N, 94°13' E) is the highest peak across the

[Figure]

eastern Kunlun Mountains, with an elevation of 6178 m. The Meikuang and Qiumian-leiketage glaciers are also located over the Kunlun Mountains, and these glaciers have an average altitude of 5100 m and 5500 m a.s.l, respectively. The Meikuang glacier is located in the eastern Kunlun Mountains, where is characterized by alluvial deposits and sand dunes, the glacier covers an area of 1.1 km2, is 1.8 km in length (Xiao et al., 2002). The Qiumianleiketage glacier is located in the Heyuan District of the Nagora River, the largest river in the Qaidam Basin, which originated in the Kunlun Mountains of the Qinghai-Tibet Plateau. The length of the glacier is 2.6 km, and the area is 1.73 km2. The glaciers of Hariqin and Meikuang have similar altitudes but are from different mountains. The Gurenhekou glacier is located on the south-eastern margin of the Nyenchen Tanglha Mountains, and seated about 90 km northwest of Lhasa, the capital city of Tibet (Liang et al., 1995).

Comment 2: In the same section I would like to see more details on the snow sample collection. For example: what was the elevation of sample locations; which ones were snow? ice? In the supplementary material I see what the cold and warm seasons correspond to, but this information is valuable to have in the manuscript itself also (In table 1 possibly).

R: Parts of the detail information of the snow sample collection were listed in the method section in Page 5, Line 25-Page 6, Line 12 (Also see the reply to comment 1). We have submitted this table as Table 1 in the previous manuscript in this journal, and the former editor suggested that this Table should be moved to the supporting information, because it is too long and not appropriated to show in the main manuscript. However, if all of the reviewers and the editor agreed, we can change the Table S1 to Table 1 in the final manuscript.

Comment 3: Although the measurement method for the filters has been used previously, it would be beneficial to have a few words on the principles of the method (in section 2.2), and how measurements were carried out in practice for this manuscript. After reading Doherty et al. (2010; 2014; 2016) it is evident that the instrument has

gone through modifications with time. Please provide information on how the instrument used in this manuscript is similar/different compared to Doherty et al. and if it contains the latest updates or not. Also, how did the authors take into account filter samples with a high mineral dust load? With higher load a bias can be introduced to the data (described well in Doherty et al. 2016, doi/10.1002/2015JD024375, and references therein). Further, information about the filters used should be provided since it can make a significant difference on the undercatch (look for example in Doherty et al. 2014).

R: Yes, we have added information on optical analysis as follows: An updated integrating sphere/integrating sandwich spectrophotometer (ISSW) was used to measure the mass mixing ratio of BC in snow, which is similar with the instrument developed by Grenfell et al. (2011). Compared with the ISSW spectrophotometer developed by Grenfell et al. (2011), the major difference is that we used two integrating sphere to calculate the relative attenuation instead of the integrating sandwich diffuser to reduce the diffuse radiation during the measuring process. This ISSW spectrophotometer measures the light attenuation spectrum from 400 to 700 nm. The total light attenuation spectrum is extended over the full spectral range by linear extrapolation from 400 to 300 and from 700 to 750 nm (Grenfell et al., 2011). Light attenuation is nominally only sensitive to ILAPs on the filter because of the diffuse radiation field and the sandwich structure of two integrated spheres in the ISSW (Doherty et al., 2014). Briefly, the transmitted light detected by the system for an ice sample, $S(\lambda)$, are compared with the signal detected for a blank filter, $S\_0 (\lambda)$, and the relative attenuation (Atn) is expressed as: Atn=ln‡[S_0 $(\lambda)$/S$(\lambda)$] The following measured parameters included equivalent BC ("C" _"BC" ˆ"equiv" ), maximum BC ("C" _"BC" ˆ"max" ), estimated BC ("C" _"BC" ˆ"est" ), fraction of light absorption by non-BC ILAPs ("f" _(non-BC)ˆ"est" ), the non-BC absorption Ångström exponent (Ånon-BC) and the absorption Ångström exponent of all ILAPs (Åtot), were calculated by using the wavelength dependence of the measured spectral light absorption and by assuming that the MACs of the BC, OC, and Fe are 6.3, 0.3, and 0.9 m2 g-1, respectively, at 550 nm and that the absorption Ångström

exponents (Å or AAE) for BC, OC, and Fe are 1.1, 6, and 3, respectively (Doherty et al., 2010, 2014; Grenfell et al., 2011; Wang et al., 2013). These parameters are defined as follows: 1. $C\_BC^{max}$ (ng g-1): maximum BC is the maximum possible BC mixing ratio in snow by assuming all light absorption is due to BC at the wavelengths of 650-700 nm. 2. $C\_BC^{est}$ (ng g-1): estimated BC is the estimated snow BC mixing ratio derived by separating the spectrally resolved total light absorption. 3. $C\_BC^{equiv}$ (ng g-1): equivalent BC is the amount of BC that would be needed to produce absorption of solar energy by all insoluble particles in snow for the wavelength-integrated from 300-750 nm. 4. Åtot: absorption Ångström exponent is calculated for all insoluble particles deposited on the filter between 450 and 600 nm. 5. Ånon-BC: non-BC absorption Ångström exponent is defined as the light absorption by non-BC components of the insoluble particles in snow between 450-600 nm. 6. $f\_{(non-BC)}^{est}$ (%): fraction of light absorption by non-BC light absorbing particles is the integrated absorption due to non-BC light absorbing particles, which is weighted by the down-welling solar flux from snow at the wavelengths of 300-750 nm. It is well known that the aerosol composition and the size distribution are key parameters that affect the absorption Ångström exponent. Doherty et al. (2010) reported that the value of the absorption Ångström exponent of OC was close to 5, which is consistent with previous studies with values ranging from 4-6 (Kirchstetter et al., 2004). Several studies indicated that the absorption Ångström exponent of mineral dust ranged from 2 to 5 (Fialho et al., 2005; Lafon et al., 2006). The variation in the absorption Ångström exponents for urban and industrial fossil fuel emissions is typically in the range of 1.0-1.5 (Millikan, 1961; Bergstrom et al., 2007), which is slightly lower than that of biomass burning, which primarily falls in the range of 1.5-2.5 (Kirchstetter et al., 2004; Bergstrom et al., 2007). Although the source attribution of the insoluble light-absorbing particles in the samples is not a dominant determinant of the value of the absorption Ångström exponent, fossil fuel burning may have a lower absorption Ångström exponent (<2) than 2-5 (Millikan, 1961; Fialho et al., 2005). In this study, we noted that the absorption Ångström exponent (Åtot) is due to a mix state of BC and non-BC impurities on our

filters, and the calculations of Åtot and Ånon-BC could be found in the study of Doherty et al. (2014). The OC mixing ratio was also determined according to Eq. (2) in Wang et al. (2013), and the Fe concentration was determined according to the inductively coupled plasma-mass spectrometry (ICP-MS) measurements.

Comment 4: In the current manuscript text the discussion on the Ångström exponent is not sufficient and I'm not entirely sure that the interpretation is correct. For the Ånon-BC fraction, more information in section 2.2 on how it was determined is needed. For the results, I do not think that the differentiation between fossil fuel and biomass burning aerosol can be determined in the way it is done here (combined with the PMF it is possible nonetheless to distinguish fossil vs. biomass burning). I assume the authors already have studied Doherty et al.'s work, but I would urge the authors review it once more (especially Doherty et al. 2014 regarding Ångström) to guide their interpretations and text (and of course reference to what is done the same way).

R: Previous studies indicated that the absorption Ångström exponent is driven primarily by the composition of the aerosol and secondarily by the aerosol size distribution. Therefore, variations in Åtot can reflect variations in the source of ILAPs in the snow. Sure, Åtot can't distinguish fossil or biomass burning, however, combining with the "C" _"BC" ^"est" and Ånon-BC, we can get the general status of the BC and non-BC contents in the snow or ice. Therefore, we note the parameter of absorption Ångström exponent is still valuable in this study. But we also reconstructed section 3.1 based on the recent study by Doherty et al. (2014). Reference: Doherty, S. J., Dang, C., Hegg, D. A., Zhang, R. D., and Warren, S. G.: Black carbon and other light-absorbing particles in snow of central North America, J. Geophys. Res.-Atmos., 119, 12807-12831, 10.1002/2014JD022350, 2014.

Comment 5: Section 3.1 would benefit from more structure. In its current form, I have a hard time following in a logical order. At the moment it starts with an introduction to the concentrations of LAP in the samples, followed by statements on the Ångström exponent, and then back to more discussion on LAP concentrations. Additionally, it

would be valuable to see the results obtained in this study compared in a larger Tibetan/Himalayan perspective with results from other nearby measurements of LAP in snow.

R: In the section 3.1, we firstly analyzed the general description of the total concentration of ILAPs and the absorption Ångström exponent in seven glaciers firstly. Then, we discussed the distribution of ILAPs concentration in each glacier region. We have also reconstructed this section and added more comparison with the other nearby measurements of ILAPs in snow.

Comment 6: Section 3.2 needs to be majorly changed. In the references provided in this manuscript the OC/BC ratio is not used in such a way that the authors here claim. From the ratio it is not (unfortunately) as straight forward to say that a ratio of XX corresponds to the aerosol particles originating from fossil or biomass burning. As an example, what about secondary formation of organics? This section therefore needs further work, or to possibly taken out from the manuscript.

R: We have taken out this section and reconstructed the result section based on the reviewer's suggestion.

Comment 7: Page 3 Lines 4-6: The second sentence of the introduction almost seems contradictory to the following sentence. I believe this contradiction could be removed with careful language editing.

R: Thanks for the reminder. We have converted the relative sentences as "Ample evidence has indicated that the deposition of insoluble light-absorbing particles (ILAPs) was one of the major factors (up to 30%) to lead the greatest decrease in length and area of negative mass balance in the TP glaciers over the past decade (Xu et al., 2006, 2009a; Yao et al., 2012; Qian et al., 2015; Li et al., 2017)." Reference: Li, S., Yao, T., Yang, W., Yu, W., and Zhu, M.: Glacier Energy and Mass Balance in the Inland Tibetan Plateau: Seasonal and Interannual Variability in Relation to Atmospheric Changes, Journal of Geophysical Research: Atmospheres, 10.1029/2017jd028120, 2018. Qian,

Y., Yasunari, T. J., Doherty, S. J., Flanner, M. G., Lau, W. K. M., Ming, J., Wang, H. L., Wang, M., Warren, S. G., and Zhang, R. D.: Light-absorbing Particles in Snow and Ice: Measurement and Modeling of Climatic and Hydrological impact, Adv. Atmos. Sci. , 32, 64-91, 10.1007/s00376-014-0010-0, 2015. Xu, B. Q., Yao, T. D., Liu, X. Q., and Wang, N. L.: Elemental and organic carbon measurements with a two-step heating-gas chromatography system in snow samples from the Tibetan Plateau, Annals of Glaciology, Vol 43, 2006, 43, 257-262, Doi 10.3189/172756406781812122, 2006. Xu, B. Q., Cao, J. J., Hansen, J., Yao, T. D., Joswia, D. R., Wang, N. L., Wu, G. J., Wang, M., Zhao, H. B., Yang, W., Liu, X. Q., and He, J. Q.: Black soot and the survival of Tibetan glaciers, Proc. Nat. Acad. Sci., 106, 22114-22118, 10.1073/pnas.0910444106, 2009a. Xu, B. Q., Wang, M., Joswiak, D. R., Cao, J. J., Yao, T. D., Wu, G. J., Yang, W., and Zhao, H. B.: Deposition of anthropogenic aerosols in a southeastern Tibetan glacier, J. Geophys. Res.-Atmos., 114, 10.1029/2008jd011510, 2009b. Yao, T. D., Thompson, L., Yang, W., Yu, W. S., Gao, Y., Guo, X. J., Yang, X. X., Duan, K. Q., Zhao, H. B., Xu, B. Q., Pu, J. C., Lu, A. X., Xiang, Y., Kattel, D. B., and Joswiak, D.: Different glacier status with atmospheric circulations in Tibetan Plateau and surroundings, Nat. Clim. Chang., 2, 663-667, Doi 10.1038/Nclimate1580, 2012.

Comment 8: Page 3 Lines 8-9: This is incorrect use of the Jacobi et al. 2015 reference for that statement.

R: We have deleted the reference.

Comment 9: Page 3 Lines 17-19: Do you mean climate forcing from BC deposition world-wide? Please clarify. Also believe reference should be Bond et al. 2013, not 2014.

R: We have corrected this mistake, and we believed that the climate forcing in here shows the global deposition of black carbon on the snow and sea ice in the industrial era. Therefore, the we have modified the sentence as "Bond et al. (2013) indicated that the best estimate of global climate forcing from BC deposition on snow and sea

ice . . ."

Comment 10: Page 3 Lines 29-2 (page 4): Insoluble organic carbon in previous studies, please provide the references.

R: We have modified the sentence as "Although the mass mixing ratio of insoluble organic carbon in the snow and ice has been widely investigated in previous studies(Xu et al., 2006, 2009; Flanner et al., 2009; Wang et al., 2013), there are still limited studies that measure the mass mixing ratios of both WSOC and ISOC in ice samples, especially across the TP regions (Li et al., 2016; Yan et al., 2016)." Reference: Flanner, M. G., Zender, C. S., Hess, P. G., Mahowald, N. M., Painter, T. H., Ramanathan, V., and Rasch, P. J.: Springtime warming and reduced snow cover from carbonaceous particles, Atmos. Chem. Phys., 9, 2481-2497, 2009. Li, C. L., Chen, P. F., Kang, S. C., Yan, F. P., Li, X. F., Qu, B., and Sillanpaa, M.: Carbonaceous matter deposition in the high glacial regions of the Tibetan Plateau, Atmos. Environ., 141, 203-208, 2016. Wang, X., Doherty, S. J., and Huang, J. P.: Black carbon and other light-absorbing impurities in snow across Northern China, J. Geophys. Res.-Atmos., 118, 1471-1492, 2013. Xu, B. Q., Yao, T. D., Liu, X. Q., and Wang, N. L.: Elemental and organic carbon measurements with a two-step heating-gas chromatography system in snow samples from the Tibetan Plateau, Ann. Glaciol., 43, 257-262, 2006. Xu, B. Q., Cao, J. J., Hansen, J., Yao, T. D., Joswia, D. R., Wang, N. L., Wu, G. J., Wang, M., Zhao, H. B., Yang, W., Liu, X. Q., and He, J. Q.: Black soot and the survival of Tibetan glaciers, Proc. Nat. Acad. Sci., 106, 22114-22118, 2009. Yan, F. P., Kang, S. C., Li, C. L., Zhang, Y. L., Qin, X., Li, Y., Zhang, X. P., Hu, Z. F., Chen, P. F., Li, X. F., Qu, B., and Sillanpaa, M.: Concentration, sources and light absorption characteristics of dissolved organic carbon on a medium-sized valley glacier, northern Tibetan Plateau, The Cryosphere, 10, 2611-2621, 2016.

Comment 11: Page 4 Line 3: The abbreviation 'ILAPs' should first be written out.

R: The sentence has been modified as "Due to the importance of the climate effects

by insoluble light-absorbing particles (ILAPs), numerous snow surveys have been con-
ducted to investigate the light absorption of ILAPs and their potential source attribution
in snow (Clarke and Noone, 1985; Doherty et al., 2010, 2014; Hegg et al., 2010; Huang
et al., 2011)". Reference: Clarke, A. D., and Noone, K. J.: Soot in the Arctic Snowpack
- a Cause for Perturbations in Radiative-Transfer, Atmos. Environ., 19, 2045-2053,
1985. Doherty, S. J., Dang, C., Hegg, D. A., Zhang, R. D., and Warren, S. G.: Black
carbon and other light-absorbing particles in snow of central North America, J. Geo-
phys. Res.-Atmos., 119, 12807-12831, 2014. Doherty, S. J., Warren, S. G., Grenfell, T.
C., Clarke, A. D., and Brandt, R. E.: Light-absorbing impurities in Arctic snow, Atmos.
Chem. Phys., 10, 11647-11680, 2010. Hegg, D. A., Warren, S. G., Grenfell, T. C.,
Doherty, S. J., and Clarke, A. D.: Sources of light-absorbing aerosol in arctic snow and
their seasonal variation, Atmos. Chem. Phys., 10, 10923-10938, 2010. Huang, J. P.,
Fu, Q. A., Zhang, W., Wang, X., Zhang, R. D., Ye, H., and Warren, S. G.: Dust and
Black Carbon in Seasonal Snow across Northern China, Bull. Amer. Meteor. Soc., 92,
175-181, 2011.

Comment 12: Page 4 Lines 3-6: Did all of these references actually perform source
attribution of the ILAP in the snow? If not, please adjust references referred to. Lines
6-9: This sentence needs to be reworked, confusing at the moment.

R: We have carefully checked the references throughout the manuscript, and rewrit-
ten the sentence as "Due to the importance of the climate effects by insoluble light-
absorbing particles (ILAPs), numerous snow surveys have been conducted to investi-
gate the light absorption of ILAPs and their potential source attribution in snow (Xu et
al., 2009a, b; Doherty et al., 2010, 2014; Hegg et al., 2010; Wang et al., 2015; Wang
et al., 2017). For instance, Hegg et al. (2009) found out that the light absorption in
Arctic snow by ILAPs is mainly originated from two distinct biomass burning sources,
a pollution source, and a marine source based on the EPA PMF receptor model." Ref-
erence: Doherty, S. J., Warren, S. G., Grenfell, T. C., Clarke, A. D., and Brandt, R.
E.: Light-absorbing impurities in Arctic snow, Atmos. Chem. Phys., 10, 11647-11680,

2010. Doherty, S. J., Dang, C., Hegg, D. A., Zhang, R. D., and Warren, S. G.: Black carbon and other light-absorbing particles in snow of central North America, J. Geophys. Res.-Atmos., 119, 12807-12831, 2014. Hegg, D. A., Warren, S. G., Grenfell, T. C., Doherty, S. J., Larson, T. V., and Clarke, A. D.: Source Attribution of Black Carbon in Arctic Snow, Environ. Sci. Technol., 43, 4016-4021, 2009. Hegg, D. A., Warren, S. G., Grenfell, T. C., Doherty, S. J., and Clarke, A. D.: Sources of light-absorbing aerosol in arctic snow and their seasonal variation, Atmos. Chem. Phys., 10, 10923-10938, 2010. Wang, M., Xu, B., Cao, J., Tie, X., Wang, H., Zhang, R., Qian, Y., Rasch, P. J., Zhao, S., Wu, G., Zhao, H., Joswiak, D. R., Li, J., and Xie, Y.: Carbonaceous aerosols recorded in a southeastern Tibetan glacier: analysis of temporal variations and model estimates of sources and radiative forcing, Atmos. Chem. Phys., 15, 1191-1204, 2015. Wang, X., Pu, W., Ren, Y., Zhang, X., Zhang, X., Shi, J., Jin, H., Dai, M., and Chen, Q.: Observations and model simulations of snow albedo reduction in seasonal snow due to insoluble light-absorbing particles during 2014 Chinese survey, Atmos. Chem. Phys., 17, 2279-2296, 2017. Xu, B. Q., Cao, J. J., Hansen, J., Yao, T. D., Joswia, D. R., Wang, N. L., Wu, G. J., Wang, M., Zhao, H. B., Yang, W., Liu, X. Q., and He, J. Q.: Black soot and the survival of Tibetan glaciers, Proc. Nat. Acad. Sci., 106, 22114-22118, 2009a. Xu, B. Q., Wang, M., Joswiak, D. R., Cao, J. J., Yao, T. D., Wu, G. J., Yang, W., and Zhao, H. B.: Deposition of anthropogenic aerosols in a southeastern Tibetan glacier, J. Geophys. Res.-Atmos., 114, 2009b.

Comment 13: Page 4 Lines 14-16: What did the results of Doherty et al. 2014 show? In the previous sentences you provide some highlight from each study, but not for Doherty et al. Could be useful for readers.

R: Sorry, we have added the results of Doherty et al. (2014), and the sentence has been rewritten as "Recently, vertical profiles of ILAPs in seasonal snow were performed from 67 North American sites, and biomass/biofuel burning, soil and fossil fuel pollution are explored as the major sources of particulate light absorption based on the chemical and optical data, and these were (Doherty et al., 2014)".

Comment 14: Page 4 Lines 17-18: At this time there has been an increasing number of observations on ILAP in Tibetan snow. The other referee provided a comprehensive list on this.

R: We have corrected this mistake.

Comment 15: Page 4 Lines 29-30: Units for the AOD numbers? And for what time period is this? Also, it would be good to include some information on what an AOD number of XX means (e.g. 0.4 meaning high optical depth, etc.)

R: Aerosol optical depth (AOD) is a description of the attenuation of light by aerosol, and it is a dimensionless quantity. Figure 1 showed the spatial distribution of the averaged AOD retrieved from Aqua-MODIS over Tibetan Plateau from 2013 to 2015. According to the reviewer's suggestion, we have added this information to the caption in Figure 1.

Comment 16: Page 5 Lines 1-2: How is the glacier located above the ELA? Lines 2-3: The glacier must encompass an elevation range, and not only located at 5743 m a.s.l. The average snowline is based on what? Reference on this number ? Lines 4-5: Is the Yuzhufeng glacier part of the highest peak? Please clarify.

R: Page 5 Lines 1-2: The sentence has been rewritten as "The Qiyi glacier (39°14' N, 97°45' E) is located in the eastern part of TP, with an elevation of 4850 m". Lines 2-3: The average snowline is just a general background of Xiaodongkemadi glacier, and doesn't correlate with our study. Therefore, we deleted the sentence. Lines 4-5: The sentence has been rewritten as "The Yuzhufeng glacier (35°38' N, 94°13' E) is the highest peak across the eastern Kunlun Mountains, with an elevation of 6178 m".

Comment 17: Page 5 Lines 8-9: The surrounding areas characteristics, why is this information important? Is such information available for other glaciers? Line 12: What is the point of the Liang et al. 1995 reference?

R: Page 5 Lines 8-9: this information was used to describe the topography of the

Meikuang glacier. The Meikuang glacier is located in the eastern Kunlun Mountains, where is characterized by alluvial deposits and sand dunes (Xiao et al., 2002), meanwhile, ∼47.9% of the sources of ILAPs in the Meikuang glacier was from soil dust in the PMF analysis, so there was a good correspondence. For line 12: It should be a general description to illustrate Yangbajing glacier and we deleted the citation of Liang et al. (1995).

Comment 18: Page 5 Line 15: I don't see in table S1 (or anywhere else) if the sample is ice or snow. Please provide this information. Line 16: What was the volume of the tubes? And what kind of tubes were they?

R: Due to the samples were only collected approximately for each six months in the TP glaciers, we note that most of the collecting samples were ice samples, which were less than 1 m from the glacier surface (Details could be found in Table S1). Line 16: We have modified the sentence as "The collected snow/ice samples were preserved in 0.5-m pure clean plastic bag with a diameter of 20 cm, and kept frozen at the State Key Laboratory of Cryospheric Sciences, Cold and Arid Regions Environmental and Engineering Research Institute in Lanzhou".

Comment 19: Page 5 Lines 18-20: Please provide more information on how the snow/ice samples were cut. After this (and elsewhere in the manuscript) I'm missing some description on the filtering of snow samples. Lines 24-28: This sentence is confusing and needs to be reworked. I do not find the argument made in the references given, but once this sentence is reorganized it may be more clearly.

R: We collected 67 ice samples in seven glaciers on the Tibetan Plateau, and cut each sample from top to bottom at a length of approximately 10 cm. Finally, we obtained a total of 189 small samples. For the filtration process, we have added the description in section 2.2. Lines 24-28: The sentence has been rewritten as "The OC mixing ratio was also determined according to Eq. (2) in Wang et al. (2013), and the Fe concentration was determined according to the inductively coupled plasma-mass spectrometry (ICP-

MS) measurements".

Comment 20: Page 6 Lines 19-20: What does this opening sentence have to do with the chemical analysis?

R: We have reconstruction the method section, and the sentence in Lines 19-20 has been deleted.

Comment 21: Page 6 Lines 20-23: What does the MACs have to do with the chemical analysis? This information should instead be included in the optical analysis section (2.2.). Line 24: Do you mean 10 mL of water from the filtered meltwater? Please clarify. In addition, a few more words describing the total carbon analyzer would be beneficial. Lines 29-30: Was it the filtered meltwater again that was analyzed for major metallic elements?

R: Page 6 Line 20-23: We have moved this information into section 2.2. Line 24: 10 ml refers to the amount of sample solution after filtration that used to measure the WSOC concentration, and more descriptions of the total carbon analyzer were described by Cong et al. (2015). Lines 29-30: We used the sample solution before filtration when analyzed the major metallic elements.

Comment 22: Page 7 Line 3-4: Did you acidify all samples or not? I find it confusing with the opening of the sentence 'generally speaking'. Line 5: Measurement precision range, do you mean $\pm$?

R: No, for each ice sample, we acidified all samples before filtration and then used ICP-MS to measure the trace element contents. Meanwhile, we deleted "generally speaking". Line 5: The relative deviation between most of the measured values of the elements and the standard reference values is within $\pm 10\%$.

Comment 23: Page 8 Line 6: What are EFc ? Please write out the abbreviation. Line 25: What does Q values stand for?

R: We have changed "EFc" to "EF", which is consistent with the caption of section 2.4.

EF is the Enrichment factor defined as the concentration ratio of a given metal to that of Al. It is a reliable measure of crustal dust, normalized to the same concentration ratio characteristic of the upper continental crust (Wedepohl, 1995), calculated with the following equation: "EF=" ãĂŰ"(X/AI)" ãĂŮ_"snow" /ãĂŰ"(X/AI)" ãĂŮ_"crust" Line 25: Q is the goodness of fit parameter calculated excluding points not fit by the model, which is defined as samples for which the uncertainty-scaled residual is greater than 4. The best solution in PMF is typically identified by the lowest Q value along with the path and may be imagined as the bottom of a trough in the multidimensional space.

Comment 24: Page 9 Line 1: Should not this section be referred to as Results and Discussion? And not only Results as it is now. Line 4: I'm confused with the number of samples. Here you state over 67 and in section 2.1 ∼67. Please clarify. Line 5: The second sentence of section 3.1 is vague and not necessary for this section. As mentioned previously, please elaborate on this in the methods section.

R: We have changed "Results" to "Results and Discussion". Line 4: The number of samples is 67 and we have modified them. Line 5: We moved the sentence into the method section, and reconstructed the method section.

Comment 25: Page 9 Lines 7-10: Here the authors provide a range for the higher values, what is the range for the lower values? Lines 15-16: This is interesting that there is no difference between seasons since several other authors have observed the contrary. This should be elaborated on in the revised manuscript. Lines 17-29: These sentences should rather be included in the methods section, assisting with data interpretation.

R: According to the Table 1, the range for the lower median values of "C" _"BC" ˆ"est" is 23-53 ng g-1 in the Xiaodongkemadi, Hariqin, and Gurenhekou glaciers. Yes, we found that there is no apparent difference of ILAPs in each glacier between monsoon and non-monsoon seasons in these glacier regions. We noted several reasons could lead this discrepancy. First of all, all of the ice samples were collected in an individual time.

Another major issue is that except the long-range transport of ILAPs, local air pollutants can also affect the ILAPs in the glacier via wet and dry deposition. For instance, Li et al. (2016) indicated that the Fossil fuel contributions of BC is much higher in the Laohugou No. 12 glacier (close to Qiyi glacier) due to human activities. Finally, based the study by Xu et al. (2009), the ILAPs in the TP glacier also can be affected by the European air to reach that location on the eastern plateau, and thus lead the greater proportion of the heavy leading of the BC in the glacier. For instance, the mass concentration of BC in the Muztagh Ata and Xiaodongkemadi glaciers in northern part of Tibetan Plateau is much higher than that in the East Rongbuk, Noijin Kangsang and Zuoqiupo glacier, which are located in the southern part of Tibetan Plateau. Above all, we noted that the reasons could be very complicated, and the possible explanation has been added in the result section in Page 13, Line 12-23. Lines 17-29ïjŽWe have moved the sentences into the method section based on the reviewer's suggestion.

Comment 26: Page 10 Lines 2-5: Low Åtot means that the LAP originated from combustion sources? Lines 16-21: I find it confusing with these couple of sentences here as the previous sentences discusses Ånon-BC, and the sentences before that Åtot. First discussion on Åtot, and then into Ånon-BC. Lines 27-29: I do not understand this reasoning and how the sentences leading up to this reasoning support it. Line 29-30: Section 3.1 started with introducing the LAP concentrations and now they come back continued in the following sentences. The structure of this section would be better by having the concentration discussion intact.

R: Based on previous studies, the variation in the absorption Ångström exponents for urban and industrial fossil fuel emissions is typically in the range of 1.0-1.5 (Millikan, 1961; Bergstrom et al., 2007), which is slightly lower than that of biomass burning ranging from 1.5 to 2.5 (Kirchstetter et al., 2004; Bergstrom et al., 2007). Meanwhile, the lower median values of the absorption Ångström exponents in Xiaodongkemadi glacier is about 2.1, so the results indicated that the emission of the ILAPs in Xiaodongkemadi glacier likely originated from the combustion sources. Lines 27-29: We have deleted

the sentence. Lines 29-30: See our reply to comment 5.

Comment 27: Page 11 Line 2: What do you mean by individual period? Lines 4-5: How is a decreasing trend observed? Please include information in the manuscript that show the trend. Line 7: How is that due to heavy human activities? Lines 5-8: Please check this sentence structure. Line 8-9: In what sense is there good agreement with Ming et al. 2013? Line 16: What does the sample depth range presented here correspond to? Line 20: In table S1 I find ISOC to be 9.16 ppm, not 8600 ng g-1.

R: As shown in Table S1, the sampling time for each sample in seven glaciers was different, so we noted that all of the ice samples were collected in the individual period from 2013 to 2015. Lines 4-5: The sentences have been modified as "Based on our sampling locations shown in Figure 1, the Qiyi, Qiumianleiketage, Meikuang, Yuzhufeng glaciers were located in the northern part of Tibetan Plateau, while Xiaodongkemadi, Hariqin, Gurenhekou glaciers were located in the southern part of Tibetan Plateau. As shown in Figure 4, the median values of the "C" _"BC" ˆ"est" and "C" _"ISOC" obviously showed a significant decreasing trend from the northern TP to the southern TP." Lines 5-8: The sentence has been rewritten as "Comparing with Hariqin, Xiaodongkemadi and Gurenhekou glaciers, the relative higher values of the "C" _"BC" ˆ"est" and "C" _"ISOC" in Qiyi, Qiumianleiketage, Yuzhufeng, and Meikuang glaciers are mainly influenced by human activities." Lines 8-9: We have modified the sentence as "The mass concentration of BC in northern TP glaciers was higher than that in southern TP glaciers, which shows a good agreement with Ming et al. (2013)." Line 16: We have rewritten this sentence more clearly as "The depth of these ice samples collected in Yuzhufeng glacier is ranging from 15 to 45 cm as Table S1 shown." Line 20: We have corrected this mistake. Reference: Ming, J., Xiao, C. D., Du, Z. C., and Yang, X. G.: An overview of black carbon deposition in High Asia glaciers and its impacts on radiation balance, Adv. Water Resour., 55, 80-87, 2013.

Comment 28: Page 12 Lines 5-6: The previous sentences suggest that it is not similar emission sources. Please clarify. Lines 6-8: A similar statement has been made earlier

in the manuscript. Please combine these observations. Line 17-18: Do you mean increased from the bottom to the top? Lines 20-21: How were the samples more complicated? Line 26-28: Would it not lead to lower LAP concentrations at the surface the way this sentence describes it now? with LAP being scavenged with meltwater?

R: We agreed with the reviewer, and deleted the sentences in Page 12 Lines 5-6. Lines 6-8: The similar statement in the earlier manuscript has been deleted. Lines 17-18: Except sites 7, 8, and 9, the mixing ratios of ILAPs in the ice samples increased remarkably from the top to the bottom from the Figure S4. Lines 20-21: Comparing with the other glaciers, there is no significant trend of the vertical profile in the mixing ratios of ILAPs in Xiaodongkemadi glacier (see Figure S5), so we indicated that the vertical profiles of the mass mixing ratios of BC, ISOC, and Fe for the ice samples in Xiaodongkemadi glacier were more complicated than those in the other regions. Lines 26-28: During snow melting process, black carbon (BC) and other insoluble light-absorbing particulate impurities (ILAPs) were retained at the snow surface, because their scavenging efficiency with meltwater was <100%. Therefore, the mass concentrations of ILAPs in surface snow increased with snow melting (Doherty et al., 2013). Reference: Doherty, S. J., Grenfell, T. C., Forsström, S., Hegg, D. L., Brandt, R. E., and Warren, S. G.: Observed vertical redistribution of black carbon and other insoluble light‐absorbing particles in melting snow, J. Geophys. Res.-Atmos., 118, 5553-5569, 2013.

Comment 29: Page 13 Line 6-7: This is the wrong Conway et al reference, should be 1996 instead of 2002. Please check this also in the reference list.

R: Sorry, we have modified the reference as follows: "Previous studies have also illustrated that the ILAPs could become trapped and integrated at the surface of the snowpack due to melting and sublimation to enrich the surface concentrations (Conway et al., 1996; Painter et al., 2012; Doherty et al., 2013)". Reference: Conway, J. H., Hardin, R. H., and Sloane, N. J. A.: Packings in Grassmannian spaces, Experimental mathematics, 5, 139-159, 1996. Doherty, S. J., Grenfell, T. C., Forsström, S.,

Hegg, D. L., Brandt, R. E., and Warren, S. G.: Observed vertical redistribution of black carbon and other insoluble light‐absorbing particles in melting snow, J. Geophys. Res.-Atmos., 118, 5553-5569, 2013. Painter, T. H., Bryant, A. C., and Skiles, S. M.: Radiative forcing by light absorbing impurities in snow from MODIS surface reflectance data, Geophys. Res. Lett., 39, L17502, doi: 10.1029/2012gl052457, 2012.

Comment 30: Page 14 Lines 15-17: Please provide a reference for this. Lines 17-19: I do not find this to be the case in given reference. Lines 23-27: How is that?

R: We have provided a reference in Lines 15-17, and modified the reference in Lines 17-19. Then, the sentences have been rewritten as "There is a very large fraction representing the average WSOC (>80%) to TOC, which can absorb solar light and enhance cloud formation through their direct and indirect climate effects (Ram et al., 2010). These results are highly consistent with previous study showing that fossil fuel combustion plays a key role in leading to the higher fraction of WSOC to total organic carbon (TOC) (Zhang et al., 2012)". Reference: Ram, K., Sarin, M. M., Strawa, A. W., Kirchstetter, T. W., and Puxbaum, H.: Spatio-temporal variability in atmospheric abundances of ec, oc and wsoc over northern india, J. Aerosol Sci., 41, 88-98, 2010. Zhang, X., Liu, Z., Hecobian, A., Zheng, M., Frank, N. H., and Edgerton, E. S.: Spatial and seasonal variations of fine particle water-soluble organic carbon (wsoc) over the southeastern united states: implications for secondary organic aerosol formation, Atmos. Chem. Phys., 12, 6593-6607, 2012.

Comment 31: Page 15 Lines 1-2: Is this a general statement or results of this paper? My guess is the former, and if it is that, I would place this statement in the introduction of this paper. Line 13-15: That there is a small contribution from dust is, to my knowledge, not consistent with that results presented earlier in the manuscript. Please clarify this.

R: Yes, it's just a general statement, and we have moved this sentence into the introduction section. In the former section, we analyzed the fraction of BC and non-BC contents in the ice samples. We should note that the non-BC contents are not only

include OC, but also include MD. However, we demonstrated the median fraction of light absorption by ILAPs is only ∼13% due to MD based on the PMF receptor model in section 3.4. As a result, we indicated that there is no conflict between our major results.

Comment 32: Page 16 Lines 11-12: What do you mean 'for the anthropogenic emission source'? I find this sentence confusing.

R: Except the natural dust source, we referred to attribute the major industrial pollution and biomass burning sources to the anthropogenic emission sources, so we noted that the natural dust source and anthropogenic emission source are both non-negligible to the light absorption by ILAPs in the TP glaciers according to the previous analysis.

Comment 33:As a last note, I wanted to comment on the in-text citations, are they done by year or alphabetical order? I did not find an order to this. Please check this throughout your manuscript.

R: We have arranged the in-text citations in chronological order throughout the manuscript.

Please also note the supplement to this comment:
https://www.the-cryosphere-discuss.net/tc-2018-86/tc-2018-86-AC2-supplement.pdf

---

## Referee Report (RR1)

Second review of Wang et al. 'Quantifying light absorption and its source attribution of insoluble light-absorbing particles in Tibetan Plateau glaciers from 2013-2015.'

After a second review of Wang et al. it is evident that some improvements have been made to the manuscript. There are things remaining that need to be addressed however. Linguistics are one major obstacle. It appears that it was not thoroughly checked after the first review. Therefore, please go through this cumbersome process for your manuscript, as it sometimes hinders a correct reading of your manuscript. It also seems like some comments have been addressed with a correct response in the author's reply to the referee comments, but the necessary/appropriate change in the revised manuscript has not been done. Please check this throughout the manuscript (see also some specific examples outlined below). In the description section on the glaciers (2.1), there is still room for improvement. For example, elevations have been added for most glaciers, but I do not find any information if it is the starting point of the glaciers or the end-point. A range for where the glacier starts to the altitude where the glacier snout ends would be desired, if such information exists, as well as some estimate on the areas of the glaciers. The results section still would benefit from being synthesized in a more organized way. For example, I do not find the information on why there is no difference in ILAPs concentrations between the monsoon and non-monsoon seasons explanatory. Likewise, elevated concentrations of ILAPs in the surface of profiles would benefit from more structure, as it currently seems unorganized. Further, I did not find any comparisons of the results acquired here with other studies of ILAPs from the TP, as mentioned in the author's reply. Please see also page specific comments below.

Page specific comments

p. 3 lines 4-7: This is a good addition to the text. This sentence needs to be reworked however (past vs. present tense; did the reference come up with the number 30%, or where did it comes from?).

p. 3 lines 7-9: This sentence does not fit with the previous sentence. In a way it contradicts what has been said previously. Something along the lines that ILAPS have a major role in TP negative glacier mass balance, as well as temperature. This should be sorted out once the linguistics have been checked.

p. 3 lines 9-10: In this sentence, BC is introduced, whereas in the previous lines ILAPs is introduced. I would urge the writers to introduce BC as a part of ILAPs (as well as the other constituents of ILAPs) before diving into BC. As it currently reads, it is confusing and jumps from one thing to another.

p. 3 lines 18-20: Similar as comment above, I would introduce BC first and then provide this suitable reference to show where BC in the snow of TP comes from.

p. 3 line 23: mineral dust (MD) has already been introduced (line 15).

p. 4 lines 7-8: Please provide references for this statement.

p. 4 lines 10-14: Did all of these references provide source attribution in the snow? It is true that they reported ILAPs in snow and ice, but not source attribution.

p. 4 lines 22-26: This sentence needs to be reworked linguistically.

p. 4 lines 27-29: How can it be a snow survey when it was only ice sampled? Also, here I think you can highlight the uniqueness of your data, for example: have all of these 7 glaciers been sampled previous for ILAPs?

p. 5 lines 7-10: This sentence (or possibly sentences) could be more informative on what the AOD numbers imply for the areas investigated here.

p. 5 lines 10-12: Qiyi glacier, which classification scheme uses bucket-valley glacier? What does it actually mean with subcontinental according to physical characteristics? Please clarify what you mean 'with an elevation of 6178 m', highest point of glacier or lowest?

p. 5 lines 13-16: Xiaodongkemadi glacier, how do you know the average snowline and mean temperature there? Your own measurements or reference?

p. 5 lines 16-17: Yuzhufeng glacier, is the glacier actually on top of the mountain peak? Or does it start from the peak flowing down?

p. 5 line 18: Meikuang and Qiumianleiketage are located 'in' Kunlun Mountains not 'over'.

p. 5 lines 25-26: Hariqin and Meikuang have similar altitudes but are from different mountains? This sentence needs to be reworked. Which are the mountains?

p. 5 lines 26-28: Elevation range for Gurenhekou glacier?

p. 5 lines 28-30: I would argue that this sentence is added to the beginning of section 2.1 as it is an introductory statement. And is it actually enrichment of ILAPs that was studied? I would not argue that considering the results presented. How were the ice samples collected? Through drilling or?

p. 6 lines 11-14: I do not think my comments on the filters and filtering procedure in the last review (see major comments in last review) was adequately addressed in the revised manuscript section on the filtering. For example, how were the samples melted? 0.2 μm refers to what on the filter? (pore size I assume?). It is good to have the reference to previous works on how the filtering was done, but I still feel like some essential information on the filtering should be included here as suggested above.

p. 7 lines 2-9: This sentence is very long, I suggest that it is shortened and it will also become more clear what the authors did.

p. 7 lines 12-13: The definition for $C_{BC}^{est}$ is not clear, please revise.

p. 8 lines 8-9: Please write out how the calculations were done, even though it is in Doherty et al. 2014, I think it would be valuable for the reader to see it how you have done it.

p. 8 lines 9-11: Are these statements part of the optical analysis? Please place them in the appropriate section.

p. 8 lines 13-14: consists of instead of 'derived from'

p. 8 line 15: liquid instead of 'liquor'.

p. 8 line 25-28: Please clarify what you are saying in these sentences, it is not clear what this means.

p. 8 line 30: With it mentioned here, it is unclear if the samples for ICP-MS are filtered ice samples or not?

p. 9 line 16: What did the straightforward method entail? Would be good to provide more information on this.

p. 9 line 28: What are the anthropogenic sources? You give examples of natural, so please provide it for anthropogenic also.

p. 9 line 28: Write out the $EF_c$ abbreviation.

p. 10 line 29- p. 11 line 2: You provide a higher range for glaciers in the northern TP, but not for the southern glaciers (and the lower range). Please be consistent.

p. 11 line 3-5: This statement is not part of the results, please remove and place it elsewhere if necessary.

p. 11 lines 12-15: The way the sentence reads now, it could be interpreted that the previous studies have found that ILAPs originated from combustion sources for this glacier, whereas I believe you want to say that due to your Å it indicates that the ILAPs come from combustion sources (and like others have indicated through their studies on Å).

p. 11 lines 25-29: I do not see how these statements connect with the previous sentence and how they are relevant for the results discussion.

p. 11 line 30: Change 'feather' to feature.

p. 12 lines 4-7: This statement would be more useful in the beginning of the results section, since you then talk about difference between north and south.

p. 12 lines 9-11: This has already been mentioned in the beginning of the results section, please add it to that.

p. 12 lines 13-15: Is it actually local? How is that? What is the explanation for that?

p. 13 lines: 10-22: I do not see how this explains the non-existing difference between monsoon and non-monsoon seasons. Please reformulate.

p. 13 lines 28-30: This is valuable information that these samples were collected from the monsoon season, but I'm left wondering what it means compared to with other TP glaciers? Does it mean that your samples should be considered as high concentrations since they were collected during the monsoon? (although you have previously stated that you found no difference between seasons) Please elaborate on this for clarity.

p. 14 lines 2-3: I commented on this previously. I find it confusing the way it is written now, if concentrations are increasing from the top to the bottom, or vice versa. I believe Doherty et al. (2013) found that the concentrations were highest at the surface. Please carefully review and restructure this.

p. 14 lines 4-6: How were the profiles more complicated compared to other profiles? In the following sentences, I do not find an explanation on this. Please add the details or take it away.

p. 14 lines 11-13: This was commented on earlier and your response was sufficient. However, you have not changed this in the revised manuscript as it still reads that ILAPS are scavenged with meltwater and that leads to higher concentrations at the surface. If ILAPs are 'washed out' from the surface layer, it would lead to lower concentrations in the surface snow. Please review and clarify.

p. 14 lines 17-22: This discussion is interesting and is touched upon earlier in the manuscript (the argument from the previous studies for example could be introduced earlier in the manuscript. To me, it seems like there are profiles in this study where the surface layer is enriched with ILAPs and then times where this is not observed. Could this information that is scattered throughout the results be collected and made into a section that is thoroughly discussed and reviewed? (possibly its own section 3.2 or something similar). I think this would only make your data stand out more and would be easier interpreted by future readers.

p. 14 lines 26-28: What types of samples is this based on? Please clarify.

p. 14 lines 28-30: The first part of the sentence almost sounds like it is part of your own results. Please rephrase this statement.

p. 14 line 30- p. 15 lines 1-2: You are only referring to one study, so it is not 'previous studies'.

p. 15 lines 4-6: How do you interpret the ratios that you are presenting? Please add in the manuscript.

p. 15 lines 7-10: I do not understand how the previous sentences lead you to this last statement in this section (3.2). Please explain this further. How are you indicating this? The last part of the statement feels more like a claim to be put in the conclusions of the manuscript.

p. 15 lines 13-16: This was supposed to removed and placed in your introduction according to your author's response.

p. 15 lines 16-18: I do not see the point of this sentence in this section. Please remove.

p. 15 lines 18-20: I would argue that fig. S7 is more informative and sums up the results better than fig. 7. Consider changing fig. S7 to the manuscript and putting the current fig. 7 as fig. S7. If this is done

please revise section 3.3 by referring to the median numbers presented in current fig. S7. Of course specific numbers could still be highlighted in the text and then referenced to in fig. S7.

p. 16 lines 22-23: This statement should be moved to conclusions.

p. 18 lines 4-8: How do these statements not contradict the arguments made in the 3.3 section, where dusts role is downplayed and BC is lifted up as the major light absorber? Please clarify these statements and synchronize the results.

Figure 1. The second half of the first sentence is redundant. Should not the wavelength information (I assume it is 500 nm?) also be included?

Figure 2. Pictures are always good, and a nice addition to the figures. But, as the text now mentions it is 'ice sampling locations' how about indicating in the pictures where the samples from each glacier were taken? These pictures could be considered to be placed in the supplementary materials.

---

## Editor Decision (ED1)

Comments

After the first revision the paper has considerably improved but need more explanation before it could be accepted for publication in the journal. Some specific points have been listed below:

Still for the sampling which I am confused. The author collected "ice samples" (is it ice core?), I am not sure the authors mean supraglacial ice, granular ice or snowpits (which depends on where ice core retrieved, above ELA or ablation zone)? As in the supplementary information, depth of ice sample is more than 50 cm. For a snow pit, it is easy to dig out with 50 cm depth, but ice, you have to drill down to 50 cm depth manually. As my experience, drilling down to get an ice core of 50cm depth is quite hard work manually. Please clarify.

In the supplementary Table S1, for examples, in the regions 1 Qiyi glacier, are the sites from 1 to 19 located at the same latitude and longitude? The author only showed one latitude and longitude coordinates.

Page 6 line 4, "Then each ice sample was cut vertically into small pieces from the surface to the bottom". This sentence is confused. Each ice sample means each ice core? The resolution of each pieces is not the same as shown in Table S1. Especially for the surface sample. Why you choose the surface layer is 22 cm for site 10, and 12.5cm for site 13?

WSOC measurements: In the previous studies of WSOC from the glaciers, precipitation and river waters in the Tibetan Plateau, e.g., Li Xiangying et al. (2018), Hu Zhaofu et al. (20187), Liu Yanmei et al. (2016), Li Chaoliu et al. (2018), Qu Bin et al. (2017), the concentrations of WSOC is much lower than the data the author measured. What kind of bottles (or vials) did the authors use to collect ice samples in this study? Any pretreatment? Any blanks of WSOC measurements? When the author cut the ice pieces, any pretreatment to shaving the outer layer (avoiding contamination) where contacted with the plastic bags? Dose the author mean the pretreatment as same as that for ice core? Any explanation for the very higher WSOC concentrations in this study?

The author mentioned "10 ml refers to the amount of sample solution after filtration that used to measure the WSOC concentration," what kind of filters used to filter the WSOC samples?

For Fe analysis, dose the author use the bulk sample or the filtered sample? How many days you used for acidifying the samples?

In the response to Reviewer 2, the author declared that "we referred to attribute the major industrial pollution and biomass burning sources to the anthropogenic emission sources." As an important biomass burning, how does the author to eliminate the impact of forest fire (natural source)?

For the MD data (average concentration is 241±452 ng g-1 on TP glaciers in the abstract) in this study, I can't see the Al concentrations, but I think the unit should be ppm rather than ppb as compared with previous studies from glaciers in the Tibetan Plateau. Al concentrations may be much higher than that of Fe in glaciers, thus calculation should be larger the current data.

In the conclusion, "The lower absorption Ångström exponent (Åtot <2) suggested that the sites 30, 51, and 56-58, 65 were primarily influenced by fossil fuel emission, whereas the rest of the sites were heavily influenced by mineral dust and biomass burning." Sites 30, 5a, and 56-58, 65, were corresponding to which glacier and which season? Why for the same glacier, the influence of fossil fuel emission, mineral dust and biomass burning are difference? Any reasons? In the conclusion, I prefer to see the clear summary of region rather than different sites.

---

## Author Response (AR2)

Response to reviewer 1

We are very grateful for the reviewer's critical comments and suggestions, which have helped us improve the paper quality substantially. We have addressed all of the comments carefully as detailed below in our point-by-point responses. Our responses start with "R:".

After a second review of Wang et al. it is evident that some improvements have been made to the manuscript. There are things remaining that need to be addressed however. Linguistics are one major obstacle. It appears that it was not thoroughly checked after the first review. Therefore, please go through this cumbersome process for your

- 10 manuscript, as it sometimes hinders a correct reading of your manuscript. It also seems like some comments have been addressed with a correct response in the author's reply to the referee comments, but the necessary/appropriate change in the revised manuscript has not been done. Please check this throughout the manuscript (see also some specific examples outlined below). In the description section on the glaciers (2.1), there is still
- 15room for improvement. For example, elevations have been added for most glaciers, but I do not find any information if it is the starting point of the glaciers or the end-point. A range for where the glacier starts to the altitude where the glacier snout ends would be desired, if such information exists, as well as some estimate on the areas of the glaciers. The results section still would benefit from being synthesized in a more
- 20 organized way. For example, I do not find the information on why there is no difference in ILAPs concentrations between the monsoon and non-monsoon seasons explanatory. Likewise, elevated concentrations of ILAPs in the surface of profiles would benefit from more structure, as it currently seems unorganized. Further, I did not find any comparisons of the results acquired here with other studies of ILAPs from the TP, as mentioned in the author's reply. Please see also page specific comments below.

25

5

R: We have carefully responded the following questions and concerns based on the reviewer's comments. Major changes have been listed as follows:

- (1) The introduction section is totally written (See introduction section).
- (2) We provide several new figures to give more information about the sampling
- 30
- locations and the topographical maps in seven glaciers (Fig. 1, Fig. 2, and Fig. S2).
- (3) We have reconstructed section 2.1, and added the necessary information about the topography of each glacier and the sampling locations (See section 2.1).

- (4) A new section 3.3 is added to investigate the scavenging and washing efficiencies of the ILAPs in the vertical ice samples in the TP glaciers.
- (5) We have checked the necessary changes throughout the manuscript very carefully.We wish the reviewer could be satisfy with the significant changes in the revised
- 5 manuscript.

**Page specific comments**

Comment 1: p. 3 lines 4-7: This is a good addition to the text. This sentence needs to be reworked however (past vs. present tense; did the reference come up with the number 30%, or where did it come from?).

10 R: The sentence has been deleted, and the introduction has been totally rewritten. Comment 2: p. 3 lines 7-9: This sentence does not fit with the previous sentence. In a way it contradicts what has been said previously. Something along the lines that ILAPS have a major role in TP negative glacier mass balance, as well as temperature. This

should be sorted out once the linguistics have been checked.

15 R: See comment 1.

Comment 3: p. 3 lines 9-10: In this sentence, BC is introduced, whereas in the previous lines ILAPs is introduced. I would urge the writers to introduce BC as a part of ILAPs (as well as the other constituents of ILAPs) before diving into BC. As it currently reads, it is confusing and jumps from one thing to another.

20 R: See comment 1.

Comment 4: p. 3 lines 18-20: Similar as comment above, I would introduce BC first and then provide this suitable reference to show where BC in the snow of TP comes from.

R: Changed as suggested.

25 Comment 5: p. 3 line 23: mineral dust (MD) has already been introduced (line 15).R: Changed as suggested.

Comment 6: p. 4 lines 7-8: Please provide references for this statement.

R: We have provided the relative references as suggested.

Comment 7: p. 4 lines 10-14: Did all of these references provide source attribution in

30 the snow? It is true that they reported ILAPs in snow and ice, but not source attribution. R: We have modified this sentence as "Due to the importance of the climate effects by ILAPs, numerous snow surveys have been conducted to investigate the light absorption of ILAPs (Xu et al., 2009a, b; Doherty et al., 2010; Huang et a., 2011; Wang et al., 2013; Dang et al., 2014), and their potential source attribution in snow and ice (Hegg et al., 2010; Zhang et al., 2013; Doherty et al., 2014; Jenkins et al., 2016; Li et al., 2016; Pu et al., 2017)."

Comment 8: p. 4 lines 22-26: This sentence needs to be reworked linguistically.

5 R: The sentence has been revised as "Doherty et al. (2014) found that the source attribution of particulate light absorption in seasonal snow is dominated by biomass/biofuel burning, soil dust and fossil fuel pollution based on the chemical and optical data from 67 North American sites."

Comment 9: p. 4 lines 27-29: How can it be a snow survey when it was only ice sampled?

Also, here I think you can highlight the uniqueness of your data, for example: have all of these 7 glaciers been sampled previous for ILAPs?
R: We have changed the "snow survey" as "an ice field campaign", and highlight the uniqueness of this large ice survey in the last paragraph of the introduction section.

Comment 10: p. 5 lines 7-10: This sentence (or possibly sentences) could be more informative on what the AOD numbers imply for the areas investigated here.

R: We noted that the AOD can represent the dry aerosol deposition and its transport pathway, which could provide useful information about the possible sources of the ILAPs in the TP glaciers. As a result, we moved this sentence to the result section and reconstructed as a separate section 3.1 of "Aerosol optical depth (AOD)".

15

20 Comment 11: p. 5 lines 10-12: Qiyi glacier, which classification scheme uses bucketvalley glacier? What does it actually mean with subcontinental according to physical characteristics? Please clarify what you mean 'with an elevation of 6178 m', highest point of glacier or lowest?

R: We have modified the description of Qivi glacier (also see Fig. 1) and reconstructed

- 25 this paragraph as "Samples 1 to 19 were collected from 2013 to 2015 during the monsoon season in the center of the Qiyi glacier (QY, 39°14' N, 97°45' E) (Fig. 1a). The QY glacier is a small valley glacier, with the area of 2.98 km2 and the length of 3.8 km. It is located in the Qilian Mountains on the north border of the TP regions. This glacier is recognized as a typical "wet island" in arid region due to its multi-land types
- 30 (e.g. forests, bushes, steppes and meadows).". More information about the other glaciers could also be found in section 2.1 in the revised manuscript.
   Comment 12: p. 5 lines 13-16: Xiaodongkemadi glacier, how do you know the average snowline and mean temperature there? Your own measurements or reference?

R: See comment 12, or section 2.1 in the revised manuscript.

Comment 13: p. 5 lines 16-17: Yuzhufeng glacier, is the glacier actually on top of the mountain peak? Or does it start from the peak flowing down?

R: See comment 12, or section 2.1 in the revised manuscript. The topography of the seven glaciers could be seen clearly in Fig. 1.

Comment 14: p. 5 line 18: Meikuang and Qiumianleiketage are located 'in' Kunlun Mountains not 'over'.

R: Changed as suggested.

5

Comment 15: p. 5 lines 25-26: Hariqin and Meikuang have similar altitudes but are

10 from different mountains? This sentence needs to be reworked. Which are the mountains?

R: See comment 12, or section 2.1 in the revised manuscript.

Comment 16: p. 5 lines 26-28: Elevation range for Gurenhekou glacier?

R: See comment 12, or section 2.1 in the revised manuscript. The elevation ranges for

- 15 Gurenhekou glacier could also be found in Fig. 1.
  - Comment 17: p. 5 lines 28-30: I would argue that this sentence is added to the beginning of section 2.1 as it is an introductory statement. And is it actually enrichment of ILAPs that was studied? I would not argue that considering the results presented. How were the ice samples collected? Through drilling or?
- 20 R: We have moved this sentence to section 2.1, and the detailed procedure on collecting ice samples in the TP glaciers is described in section 2.1 (also see Fig. 2 and Fig. S2) as follows:

Wang et al. (2015) pointed out that the annual accumulation of snow/ice at the drilling site over the TP glaciers was around 2 m on average. Therefore, a 1.2-m pure clean

- 25 plastic bag with a diameter of 20 cm was put into a vertical tube to collect the ice samples via wet and dry deposition during monsoon and non-monsoon seasons in each sample location from 2013 to 2015 (Fig. 2). Due to the high altitudes of these glaciers, the wet deposition in these areas were predominant by new fallen snow, while much less formed by precipitation. However, most of the samples were gathered by column
- 30 ice due to the multi-melting processes. Then, the column ice samples were kept frozen under -20 °C and transported to laboratory facilities at the State Key Laboratory of Cryospheric Sciences, Cold and Arid Regions Environmental and Engineering Research Institute in Lanzhou. Firstly, each sample was cut vertically into four pieces

from the top to the bottom as shown in Fig. S2, and only one of the vertical samples was cut at 10 cm resolution following clean protocols, resulting in a total of 189 samples used in this study. It should be noted that if there is a significant dirty layer inside, then, this layer will be cut and analyzed separately. Another key issue is that some of the ice

- 5 samples in the top layer is not uniform due to the multi-melting processes. Therefore, several samples were cut longer or shorter than the other samples (e.g. sites 13 and 26). To minimize the losses of ILAPs to the container walls, each sample was put into a clean glass beaker and melted quickly in a microwave oven. The melted water then immediately filtered through Nuclepore filters with a pore size of 0.2-µm, as were used
- by Doherty et al. (2010). Further details for filtrate processing can be found in Wang et al. (2013) and Doherty et al. (2014).

Comment 18: p. 6 lines 11-14: I do not think my comments on the filters and filtering procedure in the last review (see major comments in last review) was adequately addressed in the revised manuscript section on the filtering. For example, how were the

15 samples melted? 0.2 µm refers to what on the filter? (pore size I assume?). It is good to have the reference to previous works on how the filtering was done, but I still feel like some essential information on the filtering should be included here as suggested above. R: See comment 18.

Comment 19: p. 7 lines 2-9: This sentence is very long, I suggest that it is shortened and it will also become more clearly what the authors did.

R: We have reconstructed this sentence as several concise sentences.

Comment 20: p. 7 lines 12-13: The definition for CBCest is not clear, please revise.

R: We have revised the definition for  $C_{BC}^{est}$ .

20

Comment 21: p. 8 lines 8-9: Please write out how the calculations were done, even

25 though it is in Doherty et al. 2014, I think it would be valuable for the reader to see it how you have done it.

R: We have added the relative equation in section 2.2 as suggested.

Comment 22: p. 8 lines 9-11: Are these statements part of the optical analysis? Please place them in the appropriate section.

R: We have deleted this sentence.
 Comment 23: p. 8 lines 13-14: consists of instead of 'derived from'
 R: Changed as suggested

Comment 24: p. 8 line 15: liquid instead of 'liquor'.

R: We have deleted the relative statements of the WSOC throughout the manuscript based on two reasons. The most important novelty in this study is to investigate the mixing ratios of ILAPs and their source attributions in the TP glaciers. Another reason is that although we have confidence that our procedure on measuring WSOC in ice

5 sample is accurate, the values of WSOC are still unexpectedly higher than those reported for the similar regions, and we can't find the possible reason yet.
Comment 25: p. 8 line 25-28: Please clarify what you are saying in these sentences, it is not clear what this means.

R: The sentence is irrelevant with this section, and we have deleted this sentence.

10 Comment 26: p. 8 line 30: With it mentioned here, it is unclear if the samples for ICP-MS are filtered ice samples or not?

R: We have modified this sentence more clearly as "Briefly, we acidified all melted samples directly to pH<2 with ultra-pure HNO3, then let settle for 48h."

Comment 27: p. 9 line 16: What did the straightforward method entail? Would be good

15 to provide more information on this.

R: Due to the the values are not commonly used in the seawater globally, we have deleted this sentence.

Comment 28: p. 9 line 28: What are the anthropogenic sources? You give examples of natural, so please provide it for anthropogenic also.

20 R: We have provided the relative anthropogenic sources such as fossil fuels and vehicle exhaust.

Comment 29: p. 9 line 28: Write out the EFc abbreviation.

R: We have corrected this mistake, and revised "EFc" as "EF".

Comment 30: p. 10 line 29- p. 11 line 2: You provide a higher range for glaciers in the

northern TP, but not for the southern glaciers (and the lower range). Please be consistent.R: Modified as suggested.

Comment 31: p. 11 line 3-5: This statement is not part of the results, please remove and place it elsewhere if necessary.

R: The sentences have been removed.

30 Comment 32: p. 11 lines 12-15: The way the sentence reads now, it could be interpreted that the previous studies have found that ILAPs originated from combustion sources for this glacier, whereas I believe you want to say that due to your Å it indicates that the ILAPs come from combustion sources (and like others have indicated through their studies on Å).

R: Because the parameter of Å is not an appropriate tracer to provide the general sources of ILAPs, we prefer to deleted this sentence, and reconstructed this paragraph to illustrate the changes of the optical parameters measured by ISSW spectrophotometer.

5 Comment 33: p. 11 lines 25-29: I do not see how these statements connect with the previous sentence and how they are relevant for the results discussion.

R: We agreed with the reviewer that these sentences are irrelevant with this study, and we have deleted these sentences.

Comment 34: p. 11 line 30: Change 'feather' to feature.

10 R: We have corrected this mistake.

Comment 35: p. 12 lines 4-7: This statement would be more useful in the beginning of the results section, since you then talk about difference between north and south. R: Changed as suggested

Comment 36: p. 12 lines 9-11: This has already been mentioned in the beginning of the results section, please add it to that.

R: Changed as suggested.

15

Comment 37: p. 12 lines 13-15: Is it actually local? How is that? What is the explanation for that?

R: We realized that this sentence is not reasonable, and we deleted this sentence.

20 Comment 38: p. 13 lines: 10-22: I do not see how this explains the non-existing difference between monsoon and non-monsoon seasons. Please reformulate.

R: We have reanalyzed the datasets of the BC, OC, and Fe in the glaciers during monsoon and non-monsoon seasons, and we found that there are actually significant differences in three glaciers. Therefore, we have reconstructed this paragraph as follows:

- 25 "Figure 6 shows the regional variations of BC, OC, and Fe concentration in each glacier during monsoon and non-monsoon seasons. Although there were significant differences between the median and average values of the ILAPs concentration in each glacier, we found that all kinds of ILAPs exhibited a similar variation from the northern QY glacier to southern GR glacier. In addition, we collected the ice samples during both monsoon
- 30 and non-monsoon seasons in five glaciers, only except the QY and QM glaciers. On average, the BC and OC concentrations in the HRQ, XD, and GR glacier during non-monsoon season were several orders of magnitude higher than those in monsoon seasons. The result was highly consistent with the previous study by Cong et al. (2015),

who found that although the transport pathways of air masses arriving the middle Himalayas during monsoon and non-monsoon were similar, a distinctly higher carbonaceous aerosol level was found only in the non-monsoon season. Lüthi et al. (2015) also exhibited that the atmospheric brown cloud over South Asia can climb

- 5 across the Himalayan and transport of polluted air mass, which may have serious implications of the cryosphere in the TP regions. However, there appeared to be no apparent difference in the mixing ratios of ILAPs between monsoon and non-monsoon seasons in two adjacent (MK and YZF) glaciers. This can be mainly explained that, except the long-range transport of ILAPs, local air pollutants could also affect the
- 10 ILAPs in the central TP regions. For instance, Huang et al. (2018) investigated that the air masses across the MK and YZF glaciers were originated from the arid western TP and Taklimakan desert regions, and the concentration of trace elements in the YZF glacier was closer to the dust sources indicating that YZF glacier was less influenced by human activities. The median values of the  $C_{BC}^{est}$  and  $C_{OC}$  (referred as the mass
- 15 concentration of OC) obviously showed a slightly decreasing trend from the northern TP to the southern TP. The mass concentration of BC in northern TP glaciers was higher than that in southern TP glaciers, which showed a good agreement with Ming et al. (2013). "

Comment 39: p. 13 lines 28-30: This is valuable information that these samples were

- 20 collected from the monsoon season, but I'm left wondering what it means compared to with other TP glaciers? Does it mean that your samples should be considered as high concentrations since they were collected during the monsoon? (although you have previously stated that you found no difference between seasons) Please elaborate on this for clarity.
- 25 R: See comment 39, and we also summarized the ILAPs in snow and ice in the TP glaciers by recent investigations (see Table 2 in the revised manuscript), which could provide comparable results with this study.

Comment 40: p. 14 lines 2-3: I commented on this previously. I find it confusing the way it is written now, if concentrations are increasing from the top to the bottom, or

vice versa. I believe Doherty et al. (2013) found that the concentrations were highest at the surface. Please carefully review and restructure this.
 R: We have revised this sentence as "This result seemed inconsistent with a previous

study by Doherty et al. (2013). However, Xu et al. (2012) observed that the

concentrations of BC were higher not only at the snow surface, but also found at the bottom due to the percolation time of meltwater and superimposed ice by the temperature decline in the snowpack."

Comment 41: p. 14 lines 4-6: How were the profiles more complicated compared to

5 other profiles? In the following sentences, I do not find an explanation on this. Please add the details or take it away.

R: We have revised "complicated" to "variable", this sentence is modified as "Fig. S7 shows that the vertical profiles of the mass mixing ratios of BC, OC, and Fe for the ice samples in the XD glacier were more variable than those for the other regions."

- 10 Comment 42: p. 14 lines 11-13: This was commented on earlier and your response was sufficient. However, you have not changed this in the revised manuscript as it still reads that ILAPS are scavenged with meltwater and that leads to higher concentrations at the surface. If ILAPs are 'washed out' from the surface layer, it would lead to lower concentrations in the surface snow. Please review and clarify.
- 15 R: We have added a new section 3.3 of "Scavenging and washing efficiencies" as the reviewer suggested.

Comment 43: p. 14 lines 17-22: This discussion is interesting and is touched upon earlier in the manuscript (the argument from the previous studies for example could be introduced earlier in the manuscript. To me, it seems like there are profiles in this study

- 20 where the surface layer is enriched with ILAPs and then times where this is not observed. Could this information that is scattered throughout the results be collected and made into a section that is thoroughly discussed and reviewed? (possibly its own section 3.2 or something similar). I think this would only make your data stand out more and would be easier interpreted by future readers.
- R: We agreed with the reviewer, and try to divided this discussion into a separate section
  3.3 as "Scavenging and washing efficiencies". (See section 3.3)
  Comment 44: p. 14 lines 26-28: What types of samples is this based on? Please clarify.
  R: We have removed the WSOC section.

Comment 45: p. 14 lines 28-30: The first part of the sentence almost sounds like it is part of your own results. Please rephrase this statement.

**R: See comment 45.**

30

Comment 46: p. 14 line 30- p. 15 lines 1-2: You are only referring to one study, so it is not 'previous studies'.

R: See comment 45.

Comment 47: p. 15 lines 4-6: How do you interpret the ratios that you are presenting? Please add in the manuscript.

R: See comment 45.

5 Comment 48: p. 15 lines 7-10: I do not understand how the previous sentences lead you to this last statement in this section (3.2). Please explain this further. How are you indicating this? The last part of the statement feels more like a claim to be put in the conclusions of the manuscript.

R: See comment 45.

10 Comment 49: p. 15 lines 13-16: This was supposed to removed and placed in your introduction according to your author's response.

R: We have modified this sentence, and then moved to the introduction section.

Comment 50: p. 15 lines 16-18: I do not see the point of this sentence in this section. Please remove.

15 R: Changed as suggested

Comment 51: p. 15 lines 18-20: I would argue that fig. S7 is more informative and sums up the results better than fig. 7. Consider changing fig. S7 to the manuscript and putting the current fig. 7 as fig. S7. If this is done

R: We have moved Fig. S7 as Fig. 7 in the revised manuscript.

20 Comment 52: please revise section 3.3 by referring to the median numbers presented in current fig. S7. Of course, specific numbers could still be highlighted in the text and then referenced to in fig. S7.

R: Changed as suggested

Comment 53: p. 16 lines 22-23: This statement should be moved to conclusions.

25 R: Changed as suggested

Comment 54: p. 18 lines 4-8: How do these statements not contradict the arguments made in the 3.3 section, where dusts role is downplayed and BC is lifted up as the major light absorber? Please clarify these statements and synchronize the results.

R: We noted the statements here are not contradict with the previous discussion in

30 section 3.3. The results in section 3.3 show the total light absorption of ILAPs in ice samples due to BC, OC, and MD, while the findings in section 3.5 exhibited the source attributions of the ILAPs based on the PMF receptor model.

Comment 55: Figure 1. The second half of the first sentence is redundant. Should not

the wavelength information (I assume it is 500 nm?) also be included?

R: We have deleted the second half of the first sentence, and modified the caption as "Spatial distribution of the averaged AOD retrieved from Aqua-MODIS at 500 nm over Tibetan Plateau from 2013 to 2015."

5 Comment 56: Figure 2. Pictures are always good, and a nice addition to the figures. But, as the text now mentions it is 'ice sampling locations' how about indicating in the pictures where the samples from each glacier were taken? These pictures could be considered to be placed in the supplementary materials.

R: We have moved Fig. 1 as Fig. S1 in this revised manuscript as suggested, and a new

10 Fig. 1 is given to show the geographical maps of the seven glaciers, and their sampling locations.

15

20

25

Response to reviewer 2

We are very grateful for the reviewer's critical comments and suggestions, which have helped us improve the paper quality substantially. We have addressed all of the comments carefully as detailed below in our point-by-point responses. Our responses

5 start with "R:".

Comments

After the first revision the paper has considerably improved but need more explanation before it could be accepted for publication in the journal. Some specific points have been listed below:

- 10 Comment 1: Still for the sampling which I am confused. The author collected "ice samples" (is it ice core?), I am not sure the authors mean supraglacial ice, granular ice or snowpits (which depends on where ice core retrieved, above ELA or ablation zone)? As in the supplementary information, depth of ice sample is more than 50 cm. For a snow pit, it is easy to dig out with 50 cm depth, but ice, you have to drill down to 50
- 15 cm depth manually. As my experience, drilling down to get an ice core of 50cm depth is quite hard work manually. Please clarify.

R: The equipment for collecting new snow samples was shown as follows, and a clean 1.2-m plastic bag with a diameter of 20 cm was put into the tube to collect the snow and ice samples shown in Fig. 2 in the revised manuscript. However, due to the multi-

20 melting processes in the TP regions, most of the samples were gathered as column ice samples during these field campaigns. Therefore, we prefer to use ice surveys in this study.

Figure 2. The equipment for collecting new snow samples in seven TP glaciers.

Comment 2: In the supplementary Table S1, for examples, in the regions 1 Qiyi glacier, are the sites from 1 to 19 located at the same latitude and longitude? The author only showed one latitude and longitude coordinates.

R: We feel sorry for the misleading. In order to avoid the local impacts due to human
activities, we collected several snow samples at each site to better represent the regional
characteristics of the ILAPs in the TP glaciers, and each of the equipment was set up
for 100-300 m away from each other. As a result, the snow samples at sites 1-19 in Qiyi
glacier were collected near the sample location as shown in Fig. 2a.

Comment 3: Page 6 line 4, "Then each ice sample was cut vertically into small pieces from the surface to the bottom". This sentence is confused. Each ice sample means each ice core? The resolution of each pieces is not the same as shown in Table S1. Especially for the surface sample. Why you choose the surface layer is 22 cm for site 10, and 12.5cm for site 13?

R: We have revised the description as "Wang et al. (2015) pointed out that the annual

- 15 accumulation of snow/ice at the drilling site over the TP glaciers was around 2 m on average. Therefore, a 1.2-m pure clean plastic bag with a diameter of 20 cm was put into a vertical tube to collect the ice samples via wet and dry deposition during monsoon and non-monsoon seasons in each sample location from 2013 to 2015 (Fig. 2). Due to the high altitudes of these glaciers, the wet deposition in these areas were predominant
- 20 by new fallen snow, while much less formed by precipitation. However, most of the samples were gathered by column ice due to the multi-melting processes. Then, the column ice samples were kept frozen under -20 °C and transported to laboratory facilities at the State Key Laboratory of Cryospheric Sciences, Cold and Arid Regions Environmental and Engineering Research Institute in Lanzhou. Firstly, each sample
- 25 was cut vertically into four pieces from the top to the bottom as shown in Fig. S2, and only one of the vertical samples was cut at 10 cm resolution following clean protocols, resulting in a total of 189 samples used in this study. It should be noted that if there is a significant dirty layer inside, then, this layer will be cut and analyzed separately. Another key issue is that some of the ice samples in the top layer is not uniform due to
- 30 the multi-melting processes. Therefore, several samples were cut longer or shorter than the other samples (e.g. sites 13 and 26). To minimize the losses of ILAPs to the container walls, each sample was put into a clean glass beaker and melted quickly in a microwave oven. The melted water then immediately filtered through Nuclepore filters

with a pore size of 0.2- $\mu$ m, as were used by Doherty et al. (2010). Further details for filtrate processing can be found in Wang et al. (2013) and Doherty et al. (2014)."

Figure S2. The cutting processes of the column ice samples collected in each glacier

**5 following clean protocols.**

Comment 4: WSOC measurements: In the previous studies of WSOC from the glaciers, precipitation and river waters in the Tibetan Plateau, e.g., Li Xiangying et al. (2018), Hu Zhaofu et al. (20187), Liu Yanmei et al. (2016), Li Chaoliu et al. (2018), Qu Bin et al. (2017), the concentrations of WSOC is much lower than the data the author

- 10 measured. What kind of bottles (or vials) did the authors use to collect ice samples in this study? Any pretreatment? Any blanks of WSOC measurements? When the author cut the ice pieces, any pretreatment to shaving the outer layer (avoiding contamination) where contacted with the plastic bags? Dose the author mean the pretreatment as same as that for ice core? Any explanation for the very higher WSOC concentrations in this
- 15 study?

R: Due to the dataset the WSOC concentrations is unexpectedly higher than the previous studies, and this scope mainly focuses on the ILAPs in the TP glaciers, we decide to delete this section of WSOC in this study. However, although we delete the dataset of WSOC concentration in this study, we have fully confidence that the

20 procedure on measuring WSOC in ice samples is accurate. Comment 5: The author mentioned "10 ml refers to the amount of sample solution after filtration that used to measure the WSOC concentration," what kind of filters used to filter the WSOC samples?

R: See comments 4.

Comment 6: For Fe analysis, dose the author use the bulk sample or the filtered sample?How many days you used for acidifying the samples?

R: We use the bulk sample to analyze the concentration of Fe, and at least 48 hours were used for acidifying the samples.

Comment 7: In the response to Reviewer 2, the author declared that "we referred to attribute the major industrial pollution and biomass burning sources to the anthropogenic emission sources." As an important biomass burning, how does the

author to eliminate the impact of forest fire (natural source)?

R: Agreed, we can't separate the forest fire into natural and anthropogenic sources by using the PMF receptor model in this study. Therefore, we have revised the relative statements in this revised manuscript.

- 10 Comment 8: For the MD data (average concentration is 241±452 ng g-1 on TP glaciers in the abstract) in this study, I can't see the Al concentrations, but I think the unit should be ppm rather than ppb as compared with previous studies from glaciers in the Tibetan Plateau. Al concentrations may be much higher than that of Fe in glaciers, thus calculation should be larger the current data.
- 15 R: We have added the dataset of AI in Table S1, and convert the unit of ng g-1 to ppm for AI and Fe. We also noted the concentration of MD in this study is calculated by using the AI concentration based on equation 5 in the revised manuscript.

Comment 9: In the conclusion, "The lower absorption Ångström exponent (Åtot <2) suggested that the sites 30, 51, and 56-58, 65 were primarily influenced by fossil fuel

- 20 emission, whereas the rest of the sites were heavily influenced by mineral dust and biomass burning." Sites 30, 5a, and 56-58, 65, were corresponding to which glacier and which season? Why for the same glacier, the influence of fossil fuel emission, mineral dust and biomass burning are difference? Any reasons? In the conclusion, I prefer to see the clear summary of region rather than different sites.
- 25 R: We have reconstructed the conclusion section for the regional averages rather than different sites.

30

Reference

[revised manuscript text omitted]

- chemical analysis by assuming that the light absorption of mineral dust is due to iron oxide. The results indicated that the mass mixing ratios of BC, OC, and MD showed a large variation of 10-3100 ng g-1, 10-17000 ng g-1, 10-3500 ng g-1, with mean values of  $220\pm400$ ,
- 10 ng g-1, 1360+2420 ng g-1, 240+450 ng g-1 on TP glaciers during the entire ice field campaign, respectively. Although the mineral dust was assumed to be the highest contributor to the mass loading of ILAPs, we noted that the averaged light absorption of BC (50.7%) and OC (33.2%) was largely responsible for the measured light absorption in the TP glaciers at the wavelengths of 450-600 nm. The chemical elements and the selected
- 15 carbonaceous particles were also analyzed for the source attributions of the particulate light absorption based on a positive matrix factorization (PMF) receptor model. On average, the industrial pollution (33.1%), biomass/biofuel burning (29.4%), and mineral dust (37.5%) were the major sources of the ILAPs in TP glaciers.

20

25

30

|                                                                                                                                                                                                                                                                                                                                                                                                                                                                                                                                                                                                                                                                                                                                                                                                                                                                                                                                                                                                                                                                                                                                                                                                                                                                                                                                                                                                                                                                                                                                                                                                                                                                                                                                                                                                                                                                                                                                                                                                                                                                                                                                | 删除的内容: IS                                          |
|--------------------------------------------------------------------------------------------------------------------------------------------------------------------------------------------------------------------------------------------------------------------------------------------------------------------------------------------------------------------------------------------------------------------------------------------------------------------------------------------------------------------------------------------------------------------------------------------------------------------------------------------------------------------------------------------------------------------------------------------------------------------------------------------------------------------------------------------------------------------------------------------------------------------------------------------------------------------------------------------------------------------------------------------------------------------------------------------------------------------------------------------------------------------------------------------------------------------------------------------------------------------------------------------------------------------------------------------------------------------------------------------------------------------------------------------------------------------------------------------------------------------------------------------------------------------------------------------------------------------------------------------------------------------------------------------------------------------------------------------------------------------------------------------------------------------------------------------------------------------------------------------------------------------------------------------------------------------------------------------------------------------------------------------------------------------------------------------------------------------------------|----------------------------------------------------|
|                                                                                                                                                                                                                                                                                                                                                                                                                                                                                                                                                                                                                                                                                                                                                                                                                                                                                                                                                                                                                                                                                                                                                                                                                                                                                                                                                                                                                                                                                                                                                                                                                                                                                                                                                                                                                                                                                                                                                                                                                                                                                                                                | 删除的内容: a                                           |
|                                                                                                                                                                                                                                                                                                                                                                                                                                                                                                                                                                                                                                                                                                                                                                                                                                                                                                                                                                                                                                                                                                                                                                                                                                                                                                                                                                                                                                                                                                                                                                                                                                                                                                                                                                                                                                                                                                                                                                                                                                                                                                                                | 删除的内容: 218                                         |
|                                                                                                                                                                                                                                                                                                                                                                                                                                                                                                                                                                                                                                                                                                                                                                                                                                                                                                                                                                                                                                                                                                                                                                                                                                                                                                                                                                                                                                                                                                                                                                                                                                                                                                                                                                                                                                                                                                                                                                                                                                                                                                                                | 删除的内容: 397                                         |
|                                                                                                                                                                                                                                                                                                                                                                                                                                                                                                                                                                                                                                                                                                                                                                                                                                                                                                                                                                                                                                                                                                                                                                                                                                                                                                                                                                                                                                                                                                                                                                                                                                                                                                                                                                                                                                                                                                                                                                                                                                                                                                                                | 删除的内容: 1357                                        |
|                                                                                                                                                                                                                                                                                                                                                                                                                                                                                                                                                                                                                                                                                                                                                                                                                                                                                                                                                                                                                                                                                                                                                                                                                                                                                                                                                                                                                                                                                                                                                                                                                                                                                                                                                                                                                                                                                                                                                                                                                                                                                                                                | 删除的内容: 2417                                        |
| And a state of the | 删除的内容: 241                                         |
|                                                                                                                                                                                                                                                                                                                                                                                                                                                                                                                                                                                                                                                                                                                                                                                                                                                                                                                                                                                                                                                                                                                                                                                                                                                                                                                                                                                                                                                                                                                                                                                                                                                                                                                                                                                                                                                                                                                                                                                                                                                                                                                                | 删除的内容: 452                                         |
|                                                                                                                                                                                                                                                                                                                                                                                                                                                                                                                                                                                                                                                                                                                                                                                                                                                                                                                                                                                                                                                                                                                                                                                                                                                                                                                                                                                                                                                                                                                                                                                                                                                                                                                                                                                                                                                                                                                                                                                                                                                                                                                                | 删除的内容: of                                          |
|                                                                                                                                                                                                                                                                                                                                                                                                                                                                                                                                                                                                                                                                                                                                                                                                                                                                                                                                                                                                                                                                                                                                                                                                                                                                                                                                                                                                                                                                                                                                                                                                                                                                                                                                                                                                                                                                                                                                                                                                                                                                                                                                | 删除的内容: Although the mineral dust assumed to be the |

highest contributor to the mass loading of ILAPs, we noted that the averaged light absorption of BC (50.7%) and ISOC (33.2%) was largely responsible for the measured light absorption in the high mountain glaciers at the wavelengths

of 450-600 nm.

**1 Introduction**

30

Ample evidence indicated that the snow albedo at visible wavelengths is largely dominant by black carbon (BC) (Warren and Wiscombe, 1980, 1985; Brandt et al., 2011; Hadley and Kirchstetter, 2012), For instance, a mixing ratio of 10 ng g-1 of BC in snow can reduce

- 5 snow albedo by 1%, which has a similar effect to that of 500 ng g-1 of dust at 500 nm (Warren and Wiscombe, 1980; Warren, 1982; Wang et al., 2017). Chylek et al. (1984) indicated that the absorbing efficiency of BC is higher in snow than in the atmosphere due to more sunlight scattering in snow. Conway et al. (1996) measured a snow albedo reduction of 0.21 and a 50% increase in the ablation rate of natural snow attributed to 500
- 10 ng g-1 BC contamination. Liou et al. (2011) developed a geometric-optics surface-wave approach to demonstrate the snow albedo reduction by as much as ~5–10% due to small amounts of BC internally mixed with snow grains. Totally, BC accounts for 85% of absorption by all insoluble light-absorbing impurities (ILAPs) in snow at the wavelength of 400-700 nm (Bond et al., 2013). Due to the impact of BC on snow and ice albedos, the
- 15 "efficacy" of this BC-snow forcing is twice as effective as CO2, and may have contributed to global warming of the past century in the Northern Hemisphere (Hansen and Nazarenko, 2004).

The Tibetan Plateau (TP), known as the highest plateau in the world and its surrounding areas, contains the largest store of snow and ice outside the polar regions (Qin et al., 2006).

- 20 However, ~82% of the plateau's glaciers have retreated, and 10% of its permafrost has degraded in the past decade (Qiu, 2008; Yao et al., 2012). Xu et al. (2009a, b) indicated that the BC deposited in snow and ice potentially lead the melting seasons earlier, and the large retreat of these glaciers across the TP regions may affect the atmospheric circulation and ecosystem at regional and global scales in multiple ways (Qian et al., 2011; Skiles et al., 2012; Skiles et al., 2012; Skiles et al., 2012; Skiles et al., 2011; Sk
- 25 al., 2012; Sand et al., 2013). Therefore, the BC content is considered one of the major absorbers to lead great decrease in length and area of TP glaciers (Xu et al., 2006, 2009a; Qian et al., 2015; Li et al., 2016).
  In addition to BC, organic carbon (OC) and mineral dust (MD) recognized as the other

types of ILAPs that substantially contribute to springtime snowmelt and surface warming through the snow darkening effects (Painter et al., 2010, 2012; Huang et al., 2011; Kaspari

已下移 [1]: The Tibetan Plateau (TP), known as the highest plateau in the world, and its surrounding areas contain the largest snow and ice mass outside the polar regions (Qin et al., 2006). 删除的内容: The Tibetan Plateau (TP), known as the highest [1] 删除的内容: evidence has indicated 删除的内容: the deposition of 删除的内容: insoluble light-absorbing particles (ILAPs) [2] 删除的内容: The unusual increase in temperature over the [3] 删除的内容: example 删除的内容: mineral 删除的内容: on the albedo of snow and ice 删除的内容: wavelength 带格式的:字体:小四,字体颜色:文字1 删除的内容: 带格式的:字体:小四,字体颜色:文字1 带格式的:字体:小四,字体颜色:文字1 带格式的:字体:小四,字体颜色:文字1 带格式的:字体:小四,字体颜色:文字1 带格式的:字体:小四,字体颜色:文字1 带格式的:字体:小四,字体颜色:文字1 带格式的:字体:小四,字体颜色:文字 1,下标 带格式的:字体:小四,字体颜色:文字1 带格式的:字体:小四,字体颜色:文字1 带格式的:字体:小四,字体颜色:文字1 带格式的 [4] 带格式的:字体: (默认) Times, 字体颜色:黑色 带格式的: 两端对齐, 缩进: 首行缩进: 0 字符, 无孤行控制 删除的内容: also 删除的内容: Yasunari et al., 2015;

et al., 2014; Wang et al., 2013, 2014; Yasunari et al., 2015). However, the optical properties of OC in snow are still absent due to limited small-scale field campaigns and technical limitations. For instance, the OC concentrations extracted at Antarctic sites are unexpectedly higher ranging from 80 to 360 ng g-1 than those reported for Greenland (10–

- 5 40 ng g-1) and Alpine (45–98 ng g-1) for pre-industrial ice (Federer et al., 2008; Preunkert et al., 2011). Furthermore, there are still significant uncertainties in estimating the light absorption by different types of OC associated with both the chemical and optical analyses from snow samples across western North America (Dang et al., 2014). Although the contribution of OC to the global warming is generally lower than BC, but still significant
- 10 mainly over southeastern Siberia, northeastern East Asia, and western Canada (Yasunari et al., 2015). As summarized by Flanner et al. (2009), consideration of OC in snow is a key approach for better estimating the climate effects in global models due to the absorption of solar radiation by other ILAPs from the ultraviolet to visible wavelengths. It is well known that the light absorption capacity of MD mainly depends on the iron oxides
- 15 (hereafter referred to Fe) (Alfaro et al., 2004; Lafon et al., 2004, 2006; Moosmuller et al., 2012). Fe, (primarily hematite and goethite) imparted a yellow-red color is a major component, which affects the ability of mineral dust to absorb sunlight at short wavelengths, then alters the dust's radiative properties and may influence the climate (Takahashi et al., 2011; Jeong et al., 2012; Zhou et al., 2017), Cong et al. (2018) indicated that the goethite
- 20 was predominant form of Fe (81% to 98 % in mass fraction) among the glaciers in the TP regions. Painter et al. (2007) pointed out that snow cover duration in a seasonally snow-covered mountain was shortened by 18 to 35 days due to the deposition of disturbed desert dust. Wang et al. (2013) revealed that the light absorption was major dominated by OC across the grassland of Inner Mongolia across northern China, while the snow particulate
- light absorption was mainly contributed by local soil and desert dust at the northern boundary of the TP regions.
   Due to the importance of the climate effects by ILAPs, numerous snow surveys have been conducted to investigate the light absorption of ILAPs (Xu et al., 2009a, b; Doherty et al., 2010; Huang et a., 2011; Wang et al., 2013; Dang et al., 2014), and their potential source
- 30 attribution in snow and ice (Hegg et al., 2010; Zhang et al., 2013a; Doherty et al., 2014

|---------------------------|---------------------------------------------------------------------|
| /γ                        | ( 带格式的: 字体: 小四                                                      |
| $\langle \rangle$         | ( 带格式的: 字体: 小四                                                      |
| $\langle \rangle \rangle$ |                                                                     |
|                           | 〒11111                                                              |
|                           | 带恰式的: 子'种:小四,子'体颜巴: 义子 I                                     |
|                           |                                                                     |
|                           | 市田山山, 于平. 小四, 于平颜巴. 又于 1                                            |
| Y                         | ( 带格式的: 字体颜色:自动设置                                            |
|                           | highest plateau in the world, and its surrounding areas contain [6] |

(已移动(插入)[1]

Jenkins et al., 2016; Li et al., 2016; Pu et al., 2017), Hegg et al. (2009) indicated that the light absorption by ILAPs in Arctic snow is mainly originated from biomass burning, pollution, and marine sources based on a positive matrix factorization (PMF) receptor model. Doherty et al. (2014) found that the source attribution of particulate light absorption

- 5 in seasonal snow is dominated by biomass/biofuel burning, soil dust, and fossil fuel pollution based on the chemical and optical data from 67 North American sites, Up to now, the light absorption and emission sources of ILAPs remain poorly understand.
  Increasing the in-situ measurements of ILAPs in snow and ice is the most urgent task to explore the glacier retreat, especially in the TP regions. Here, we performed a large survey
- 10 on collecting column ice samples on seven glaciers in the TPvregions during the monsoon and non-monsoon seasons from 2013-2015. By using an integrating sphere/integrating sandwich spectrophotometer (ISSW) system associated with the chemical analysis, the particulate light absorption by BC, OC, and MD in TP glaciers was evaluated. Finally, the relative contributions of their emission sources in these glaciers was explored based on a
- 15 PMF receptor model.
  - **2** Site description and methods

**2.1 Site description and sample collection**

According to the second Chinese glacier inventory dataset, Fig. 1 exhibits the topographical maps in each glacier associated with the sampling locations (Liu et al., 2014). Fig. S1

- 20 shows the pictures of the sampling locations in all seven glaciers, and all these glaciers are arranged from north to south according to their latitude and longitude in this study. Basically, the sampling locations are selected to be at least 50 km apart from the main road and the cities to minimize the effects of local sources. ~67 column ice samples were gathered during monsoon and non-monsoon seasons along a south-north transect over the
- 25 TP regions from 2013-2015. It is worth noting that the seven glaciers can represent of different climate and land surface types gradually from the dry area to wet area along the northern to the southern over the TP regions.
  Samples 1 to 19 were collected from 2013 to 2015 during the monsoon season in the center of the Qiyi glacier (QY, 39°14' N, 97°45' E) (Fig. 1a). The QY glacier is a small valley
- 30 glacier, with the area of 2.98 km2 and the length of 3.8 km. It is located in the Qilian

|   | Recently, Li et al. (2016) exhibited that similar contributions  |
|---|------------------------------------------------------------------|
|   | from fossil fuel (46±11%) and biomass (54±11%)                   |
|   | combustion of the BC sources based on the dual-carbon            |
|   | isotopes technique from aerosol and snowpit samples in the       |
|   | TP regions. Bond et al. (2013) indicated that the best estimate  |
|   | of climate forcing from BC deposition on snow and sea ice in     |
|   | 巴移动(插入)[2]                                                       |
|   |                                                                  |
| V | 已上移 [2]: Due to the importance of the climate effects by         |
| l |                                                                  |
| 7 |                                                                  |
|   | ( 带格式的: 字体: (默认) Times New Roman, 字体颜色: 自
动设置                 |
体, 字体颜色: 文字 1 |
|   | 已移动(插入) [5]                                                      |
|   | 已下移 [6]: As shown in Fig. 1, the spatial distribution of         |
| 1 |                                                                  |
| 1 |                                                                  |
| Ι |                                                                  |
[15]                        |

Mountains on the north border of the TP regions, This glacier is recognized as a typical "wet island" in arid region due to its multi-land types (e.g. forests, bushes, steppes and meadows).

- Samples 20 to 22 were collected during the non-monsoon season in the southeast
- Qiumianleiketage glacier (QM, 36°70' N, 90°73' E), which is originated from the Kunlun Mountains of the Qinghai-Tibet Plateau (Fig. 1b). The length of the QM glacier is 2.6 km, and the area is 1.73 km2.
   Samples 23-32 were collected in the northern Meikuang glacier during both monsoon and

non-monsoon season (MK, 35°42' N, 94°12' E). The MK glacier is located in the eastern

10 Kunlun Mountains, where is characterized by alluvial deposits and sand dunes. The MK glacier is 1.8 km in length with an area of 1.1 km2 (Fig. 1c).

As shown in Fig. 1d, samples 33-44 were collected in the southwest Yuzhufeng glacier (YZF, 35°38' N, 94°13' E). The YZF glacier is adjacent to MK glacier with the highest peak of 6178 m across the eastern Kunlun Mountains at the northern margin of the TP

- 15 regions. The glacier is surrounded by a small quantity of ferns, forests and some bushes due to the high altitude as well as the cold and arid climate.
  Samples 45-49 were collected in the center of Hariqin glacier (HRQ, 33°14' N, 92°09' E), which is located at the headwaters of the Dongkemadi river on the northern slope of the Tanggula Mountains in the central region of the Qinghai-Tibetan Plateau (Fig. 1e), The
- 20 HRQ glacier face north, with a mountain peak of 5820 m a.s.l. to its terminus of 5400 m a.s.l.

Samples 50-60 were collected in the southern Xiaodongkemadi glacier (XD, 33°04' N, 92°04' E). The XD glacier is adjacent to HRQ glacier, with an area of 1.767 km2 and 2.8 km in length (Fig. 1f). The elevations of the glacier from the peak to its terminus are 5900

- 25 and 5500 m a.s.l., respectively. It has a cold steppe landscape\_mainly surrounded by tundra. Samples 61-67 were collected in the eastern Gurenhekou glacier (GR, 30°19' N, 90°46' E). The GR glacier is relatively small and cold alpine-type valley glacier in the central part of the southern TP, which is seated about 90 km northwest of Lhasa, the capital city of Tibet (Fig. 1g), The glacier area is 1.4 km2, with a length and width of 2.5 km and 0.6 km, and
- 30 the elevation is in the range of 5600 and 6000 m a.s.l. Kang et al. (2009) and Bolch et al.

|-------------------|------------------------------------------------------------------|
|                   | bucket-valley glaciers according to its shape, and is classified |
|                   | subcontinental glacier according to the physical                 |
| $\langle \rangle$ | characteristics of glacier.                                      |
|                   |                                                                  |
|                   | …[22]
|                   |                                                                  |
|                   | 删际的内容: in the central Qinghai-Tibetan Plateau It is              |
| $\ $              | 2.8 km in length and the average snowline is 5560 m              |
|                   | a.s.lThe elevations of the glacier from the peak to its          |
|                   | terminus are 5900 and 5500 m a.s.l., respectively. The annual    |
| 7                 |                                                                  |
| -                 |                                                                  |

(2010) indicated that the Gurenhekou glacier is mainly influenced by both the continental climate of central Asia and the Indian monsoon system.

Wang et al. (2015) pointed out that the annual accumulation of snow/ice at the drilling site over the TP glaciers was around 2 m on average. Therefore, a 1.2-m pure clean plastic bag

- 5 with a diameter of 20 cm was put into a vertical tube to collect the ice samples via wet and dry deposition during monsoon and non-monsoon seasons in each sample location from 2013 to 2015 (Fig. 2). Due to the high altitudes of these glaciers, the wet deposition in these areas were predominant by new fallen snow, while much less formed by precipitation. However, most of the samples were gathered by column ice due to the multi-melting
- 10 processes. Then, the column ice samples were kept frozen under -20 °C and transported to laboratory facilities at the State Key Laboratory of Cryospheric Sciences, Cold and Arid Regions Environmental and Engineering Research Institute in Lanzhou. Firstly, each sample was cut vertically into four pieces from the top to the bottom as shown in Fig. S2, and only one of the vertical samples was cut at 10 cm resolution following clean protocols,
- 15 resulting in a total of 189 samples used in this study. It should be noted that if there is a significant dirty layer inside, then, this layer will be cut and analyzed separately. Another key issue is that some of the ice samples in the top layer is not uniform due to the multi-melting processes. Therefore, several samples were cut longer or shorter than the other samples (e.g. sites 13 and 26). To minimize the losses of ILAPs to the container walls, each
- 20 sample, was put into a clean glass beaker and melted quickly in a microwave oven. The melted water then immediately filtered through Nuclepore filters with a pore size of 0.2μm, as were used by Doherty et al. (2010). Further details for filtrate processing can be found in Wang et al. (2013) and Doherty et al. (2014).

**2.2 Optical analysis**

- An updated integrating sphere/integrating sandwich spectrophotometer (ISSW) was used to calculate the mass mixing ratio of BC in the ice samples, which is similar with the instrument developed by Grenfell et al. (2011). Compared with the ISSW spectrophotometer developed by Grenfell et al. (2011), the major difference is that we used two integrating spheres instead of the integrating sandwich diffuser to reduce the diffuse
- 30 radiation during the measuring process. This ISSW spectrophotometer measures the light

已上移 [5]: To investigate the enrichment of ILAPs via wet and dry deposition on glaciers, we collected 67 ice samples in seven glaciers on the Tibetan Plateau from May 2013 to October 2015 (Fig. 2).

| ł | 带恰式的 [32]                                                        |
|---|------------------------------------------------------------------|
|   | under -20 °C and transported to laboratory facilities at the     |
|   | State Key Laboratory of Cryospheric Sciences, Cold and Arid      |
| 1 | 已上移 [4]: Therefore, the ice samples numbered in                  |
|   | chronological order from 1 to 19 was in the Qiyi glacier,        |
|   | while sites 20-22, 23-32, 33-44, 45-49, 50-60, and 61-67 in      |
|   | in Lanzhou University. Theampleswasereput into a                 |
|   | clean glass beaker and quicklyelted quickly in a [34]            |
|   | processing the ice filtration processeshavean be found           |
|   | been previously reported (n Wang et al. (2013)Doherty et
[35] |
| Ì | ( 带格式的: 两端对齐                                              |
|   | of BC in the ice samplessnow which is similar with the           |

instrument developed by Grenfell et al. (2011). Compared ... [36]

attenuation spectrum from 400 to 700 nm. The total light attenuation spectrum is extended over the full spectral range by linear extrapolation from 400 to 300 and from 700 to 750 nm. Light attenuation is nominally only sensitive to ILAPs on the filter because of the diffuse radiation field and the sandwich structure of two integrated spheres in the ISSW

5 (Doherty et al., 2014). Briefly, the transmitted light detected by the system for an ice sample, S( $\lambda$ ), is compared with the signal detected for a blank filter, S0( $\lambda$ ), and the relative attenuation (Atn) is expressed as:

Atn=ln[S0( $\lambda$ )/S( $\lambda$ )]

20

25

(1)

- The MACs and the absorption Ångström exponents (Å) for BC, OC, and Fe used in this study could be found in Wang et al. (2013). By using this technique, we can estimate the following parameters included equivalent BC ( $C_{BC}^{equiv}$ ), maximum BC ( $C_{BC}^{max}$ ), estimated BC ( $C_{BC}^{est}$ ), fraction of light absorption by non-BC ILAPs ( $f_{non-BC}^{est}$ ), the absorption Ångström exponent of non-BC ILAPs ( $\dot{A}_{non-BC}$ ) and the total absorption Ångström exponent ( $\dot{A}_{tot}$ ). These parameters are defined as follows:
- 15 **1.**  $C_{BC}^{max}$  (ng g-1): maximum BC is the maximum possible BC mixing ratio in snow by assuming that all light absorption is due to BC at the wavelengths of 650-700 nm.

2.  $C_{BC}^{est}$  (ng g-1): *estimated* BC is the estimated true mass of BC in snow derived by separating the spectrally resolved total light absorption and non-BC fractions.

3.  $C_{BC}^{equiv}$  (ng g-1): equivalent BC is the amount of BC that would be needed to produce absorption of solar energy by all insoluble particles in snow for the wavelength-integrated from 300-750 nm.

4.  $Å_{tot}$ : absorption Ångström exponent is calculated for all insoluble particles deposited on the filter between 450 and 600 nm.

5.  $Å_{non-BC}$ : non-BC absorption Ångström exponent is derived from the light absorption by non-BC components of the insoluble particles in snow between 450-600 nm.

6.  $f_{non-BC}^{est}$  (%): fraction of light absorption by non-BC light absorbing particles is the integrated absorption due to non-BC light absorbing particles, which is weighted by the down-welling solar flux at the wavelengths of 300-750 nm.

It is well known that the aerosol composition and the size distribution are key parameters.

30 that affect the absorption Ångström exponent. Doherty et al. (2010) reported that the value

**带格式的:** 字体: 小四, 字体颜色: 文字 1 **带格式的:** 字体: 小四, 字体颜色: 文字 1 删除的内容: The following measured parameters included equivalent BC ( $C_{BC}^{equiv}$ ), maximum BC ( $C_{BC}^{max}$ ), estimated BC ( $C_{BC}^{est}$ ), fraction of light absorption by non-BC ILAPs ( $f_{non-BC}^{est}$ ), the non-BC absorption Ångström exponent ( $A_{non}$ . BC) and the absorption Ångström exponent of all ILAPs ( $\hat{A}_{tot}$ ), were calculated by using the wavelength dependence of the measured spectral light absorption and by assuming that the MACs of the BC, OC, and Fe are 6.3, 0.3, and 0.9 m2 g-1, respectively, at 550 nm and that the absorption Ångström exponents (Å or AAE) for BC, OC, and Fe are 1.1, 6, and 3, respectively (Doherty et al., 2010, 2014; Grenfell et al., 2011; Wang et al., 2013). These parameters are defined as follows:

带格式的:两端对齐,定义网格后自动调整右缩进,调整中 文与西文文字的间距,调整中文与数字的间距 of the absorption Ångström exponent of OC was close to 5, which is consistent with previous studies with values ranging from 4-6 (Kirchstetter et al., 2004). Several studies indicated that the absorption Ångström exponent of mineral dust ranged from 2 to 5 (Fialho et al., 2005; Lafon et al., 2006; Zhou et al., 2017; Cong et al., 2018). The variation of the

- absorption Ångström exponents for urban and industrial fossil fuel emissions is typically in the range of 1.0-1.5 (Millikan, 1961; Bergstrom et al., 2007), which is slightly lower than that of biomass burning aerosols, which primarily falls in the range of 1.5-2.5 (Kirchstetter et al., 2004; Bergstrom et al., 2007). In this study, we noted that the absorption Ångström exponent ( $\hat{A}_{tot}$ ) is due to the mix state of BC and non-BC impurities on the filters,
- 10 and the calculations of  $\hat{A}_{tot}$  and  $\hat{A}_{non-BC}$  could be found in the study of Doherty et al. (2014). The  $\hat{A}_{non-BC}$  is calculated as a linear combination of contributions to light absorption due to OC and Fe, and the equation is listed as follows:

**$\underline{\mathring{A}_{non-BC}} = \underline{F_{OC}} \times \underline{\mathring{A}_{OC}} + \underline{F_{Fe}} \times \underline{\mathring{A}_{Fe}}$**

**2.3 Chemical analysis**

- 15 The major metallic elements (Al, Cr, Mn, Fe, Ni, Cu, Zn, Cd, Pb) were analyzed by an inductively coupled plasma-mass spectrometry (ICP-MS, X-7 Thermo Elemental) at the Institute of Tibetan Plateau Research in Beijing. The detection limits are Al, 0.238 ng ml-1; Cr, 0.075 ng ml-1; Mn, 0.006 ng ml-1; Fe, 4.146 ng ml-1; Ni, 0.049 ng ml-1; Cu, 0.054 ng ml-1; Zn, 0.049 ng ml-1; Cd, 0.002 ng ml-1; Pb, 0.002 ng ml-1. Briefly, we acidified all
- 20 melted samples directly to pH<2 with ultra-pure HNO3e then let settled for 48hv,The relative deviation between most of the measured values and the standard reference values is within 10%. Details on these procedures are given in Li et al. (2009) and Cong et al. (2010).
- Meanwhile, for the filtrated water samples, we measured the major anions (Cl\*, NO2\*, NO3\*, SO42\*) and cations (Na+, NH4+, K+, Mg2+, Ca2+) with an ion chromatograph (Dionex 320; Dionex, Sunnyvale, CA) using a CS12 column for cations and an AS11 column for anions at the Institute of Tibetan Plateau Research in Beijing. All the detection limit of the ions was 1\_µg · l-1. In addition, except for the anions and cations and trace elements, CI salt, MD, and biosmoke K (KBiosmoke) were determined to assess the mass
- 30 contributions of the major components in the ice samples. CI salt was estimated as follows

|   | // 删除的内容: inf the absorption Ångström exponents for                                  |
|---|--------------------------------------------------------------------------------------|
|   | urban and industrial fossil fuel emissions is typically in the                       |
|   | range of 1.0-1.5 (Millikan, 1961; Bergstrom et al., 2007),                           |
|   | which is slightly lower than that of biomass burning aerosols,                       |
|   | which primarily falls in the range of 1.5-2.5 (Kirchstetter et                       |
|   | al., 2004; Bergstrom et al., 2007). Although the source                              |
|   | attribution of the insoluble light-absorbing particles in the                        |
|   | samples is not a dominant determinant of the value of the                            |
|   | absorption Ångström exponent, fossil fuel burning may have                           |
|   | a lower absorption Ångström exponent (<2) than 2-5                                   |
|   | (Millikan, 1961; Fialho et al., 2005)n this study, we noted                          |
|   | that the absorption Ångström exponent $(\hat{A}_{tot})$ is due to a he               |
|   | mix state of BC and non-BC impurities on our he filters,                             |
|   | and the calculations of $\hat{A}_{tot}$ and $\hat{A}_{non-BC}$ could be found in the |
|   | study of Doherty et al. (2014). The OC mixing ratio was also                         |
|   | /////////////////////////////////////                                                |
|   |                                                                                      |
|   |                                                                                      |
|   | 带恰式的:子体颜巴:目刻设直                                                                       |
| 1 | →                                                                                    |
| 1 | ────────────────────────────────────                                                 |
| 1 | →                                                                                    |

(2)

|    | in accordance with Pio et al. (2007), by adding to sodium, chloride, and se                                                                                                                                                                                                                                                                                                                                                                                                                                                                                                                                                                                                                                                                                                                                                                                                                                                                                                                                                                                                                                                                                                                                                                                                                                                                                                                                                                                                                                                                                                                                                                                                                                                                                                                                                                                                                                                                                                                                                                                                                                                  | ea-salt      |                                                             |
|----|------------------------------------------------------------------------------------------------------------------------------------------------------------------------------------------------------------------------------------------------------------------------------------------------------------------------------------------------------------------------------------------------------------------------------------------------------------------------------------------------------------------------------------------------------------------------------------------------------------------------------------------------------------------------------------------------------------------------------------------------------------------------------------------------------------------------------------------------------------------------------------------------------------------------------------------------------------------------------------------------------------------------------------------------------------------------------------------------------------------------------------------------------------------------------------------------------------------------------------------------------------------------------------------------------------------------------------------------------------------------------------------------------------------------------------------------------------------------------------------------------------------------------------------------------------------------------------------------------------------------------------------------------------------------------------------------------------------------------------------------------------------------------------------------------------------------------------------------------------------------------------------------------------------------------------------------------------------------------------------------------------------------------------------------------------------------------------------------------------------------------|--------------|-------------------------------------------------------------|
|    | contributions of sodium, magnesium, calcium, potassium, and sulfate, as follows:                                                                                                                                                                                                                                                                                                                                                                                                                                                                                                                                                                                                                                                                                                                                                                                                                                                                                                                                                                                                                                                                                                                                                                                                                                                                                                                                                                                                                                                                                                                                                                                                                                                                                                                                                                                                                                                                                                                                                                                                                                             |              |                                                             |
|    | $CL_{salt} = Na_{Ss}^{+} + Cl^{-} + Mg_{S_{s}}^{2+} + Ca_{S_{s}}^{2+} + K_{S_{s}}^{+} + SO_{4S_{s}}^{2-}$                                                                                                                                                                                                                                                                                                                                                                                                                                                                                                                                                                                                                                                                                                                                                                                                                                                                                                                                                                                                                                                                                                                                                                                                                                                                                                                                                                                                                                                                                                                                                                                                                                                                                                                                                                                                                                                                                                                                                                                                                    |              |                                                             |
|    | $= Na_{Ss}^{+} + Cl^{-} + 0.12Na_{Ss}^{+} + 0.038Na_{Ss}^{+} + 0.038Na_{Ss}^{+} + 0.25Na_{Ss}^{+}$                                                                                                                                                                                                                                                                                                                                                                                                                                                                                                                                                                                                                                                                                                                                                                                                                                                                                                                                                                                                                                                                                                                                                                                                                                                                                                                                                                                                                                                                                                                                                                                                                                                                                                                                                                                                                                                                                                                                                                                                                           | (3)          | 删除的内容: 2                                                    |
| 5  | $Na_{SS} = Na_{Total} - Al \cdot (Na/Al)_{Crust}$                                                                                                                                                                                                                                                                                                                                                                                                                                                                                                                                                                                                                                                                                                                                                                                                                                                                                                                                                                                                                                                                                                                                                                                                                                                                                                                                                                                                                                                                                                                                                                                                                                                                                                                                                                                                                                                                                                                                                                                                                                                                            | (4 )  | 删除的内容: 3                                                    |
|    | Where $(Na/Al)_{Crust} = 0.33$ , and represents the Na/Al ratio in the dust materials (Wede                                                                                                                                                                                                                                                                                                                                                                                                                                                                                                                                                                                                                                                                                                                                                                                                                                                                                                                                                                                                                                                                                                                                                                                                                                                                                                                                                                                                                                                                                                                                                                                                                                                                                                                                                                                                                                                                                                                                                                                                                                  | pohl, 🔸      |                                                             |
|    | 1995).                                                                                                                                                                                                                                                                                                                                                                                                                                                                                                                                                                                                                                                                                                                                                                                                                                                                                                                                                                                                                                                                                                                                                                                                                                                                                                                                                                                                                                                                                                                                                                                                                                                                                                                                                                                                                                                                                                                                                                                                                                                                                                                       |              | 删除的内容:                                                      |
|    | The MD content was calculated by a straightforward method, and the Al concentration                                                                                                                                                                                                                                                                                                                                                                                                                                                                                                                                                                                                                                                                                                                                                                                                                                                                                                                                                                                                                                                                                                                                                                                                                                                                                                                                                                                                                                                                                                                                                                                                                                                                                                                                                                                                                                                                                                                                                                                                                                          | ion in       | (带格式的:两端对齐,段落间距段后:0磅                                        |
|    | dust was estimated as 7% (Zhang et al., 2013b):                                                                                                                                                                                                                                                                                                                                                                                                                                                                                                                                                                                                                                                                                                                                                                                                                                                                                                                                                                                                                                                                                                                                                                                                                                                                                                                                                                                                                                                                                                                                                                                                                                                                                                                                                                                                                                                                                                                                                                                                                                                                              |              | 删除的内容: With 0.12, 0.038, 0.038, and 0.25 being the          |
| 10 | MD=A1/0.07                                                                                                                                                                                                                                                                                                                                                                                                                                                                                                                                                                                                                                                                                                                                                                                                                                                                                                                                                                                                                                                                                                                                                                                                                                                                                                                                                                                                                                                                                                                                                                                                                                                                                                                                                                                                                                                                                                                                                                                                                                                                                                                   | (5)          | mass rations in seawater of magnesium to sodium, calcium to |
|    | We determined K Biosmoke as follows (Pu et al., 2017):                                                                                                                                                                                                                                                                                                                                                                                                                                                                                                                                                                                                                                                                                                                                                                                                                                                                                                                                                                                                                                                                                                                                                                                                                                                                                                                                                                                                                                                                                                                                                                                                                                                                                                                                                                                                                                                                                                                                                                                                                                                            |              | sodium, as well as potassium to sodium and sulfate to       |
|    | $K_{Biosmoke} = K_{Total} - K_{Dust} - K_{Ss}$                                                                                                                                                                                                                                                                                                                                                                                                                                                                                                                                                                                                                                                                                                                                                                                                                                                                                                                                                                                                                                                                                                                                                                                                                                                                                                                                                                                                                                                                                                                                                                                                                                                                                                                                                                                                                                                                                                                                                                                                                                                                               | (j)          | sodium, respectively.                                       |
|    | $K_{Dust} = Al \cdot (K/Al)_{Crust}$                                                                                                                                                                                                                                                                                                                                                                                                                                                                                                                                                                                                                                                                                                                                                                                                                                                                                                                                                                                                                                                                                                                                                                                                                                                                                                                                                                                                                                                                                                                                                                                                                                                                                                                                                                                                                                                                                                                                                                                                                                                                                         | (7)          | 删除的内容: t                                                    |
|    | $K_{Ss} = Na_{Ss} \cdot 0.038$                                                                                                                                                                                                                                                                                                                                                                                                                                                                                                                                                                                                                                                                                                                                                                                                                                                                                                                                                                                                                                                                                                                                                                                                                                                                                                                                                                                                                                                                                                                                                                                                                                                                                                                                                                                                                                                                                                                                                                                                                                                                                               | ( 8 ) | 删除的内容: 4                                                    |
| 15 | Where $(K/Al)_{Crust}$ is 0.37, which represents the K/Al ratio in the dust materials (Weather the second | depohl,      | 带格式的:两端对齐                                                   |
|    | 1995) and $Na_{Ss}$ is estimated by Eq. (4),                                                                                                                                                                                                                                                                                                                                                                                                                                                                                                                                                                                                                                                                                                                                                                                                                                                                                                                                                                                                                                                                                                                                                                                                                                                                                                                                                                                                                                                                                                                                                                                                                                                                                                                                                                                                                                                                                                                                                                                                                                                                                 |              | 删除的内容:5                                                     |
|    | 2.4 Enrichment factor (EF)                                                                                                                                                                                                                                                                                                                                                                                                                                                                                                                                                                                                                                                                                                                                                                                                                                                                                                                                                                                                                                                                                                                                                                                                                                                                                                                                                                                                                                                                                                                                                                                                                                                                                                                                                                                                                                                                                                                                                                                                                                                                                                   |              | 删除的内容:6                                                     |
|    | To evaluate the relative contributions of trace elements from natural (e.g., mineral and                                                                                                                                                                                                                                                                                                                                                                                                                                                                                                                                                                                                                                                                                                                                                                                                                                                                                                                                                                                                                                                                                                                                                                                                                                                                                                                                                                                                                                                                                                                                                                                                                                                                                                                                                                                                                                                                                                                                                                                                                                     | nd soil      | אראנטאווש. ס                                                |
|    | dust) versus anthropogenic sources (e.g., fossil fuels and vehicle exhaust), an inter-                                                                                                                                                                                                                                                                                                                                                                                                                                                                                                                                                                                                                                                                                                                                                                                                                                                                                                                                                                                                                                                                                                                                                                                                                                                                                                                                                                                                                                                                                                                                                                                                                                                                                                                                                                                                                                                                                                                                                                                                                                       | annual       | 删除的内容: 7                                                    |
| 20 | comparison of EF values, which represent the enrichment of a given element relative

---

## Author Response (AR3)

Stallard Scientific Limited 56 Brougham Street Nelson 7010 New Zealand Tel: +64 3 5489108 Fax: +64 3 5489106 Email: info@stallardediting.com Web: www.stallardediting.com

**Invoice**

| Date:       | 28 December 2018                                                                                                                          |
|-------------|-------------------------------------------------------------------------------------------------------------------------------------------|
| Job number: | 18492                                                                                                                                     |
| Title:      | Quantifying light absorption and its source attribution of insoluble light-absorbing particles in Tibetan Plateau glaciers from 2013-2015 |
| Client:     | Prof. Xin Wang
Lanzhou University, Lanzhou, Gansu
China                                                                             |

| Description                           |      | Cost                 | Total    |
|---------------------------------------|------|----------------------|----------|
| Scientific editing                    |      |                      |          |
| Word count:                           | 7593 | US\$22 per 250 words | 668      |
|                                       |      |                      |          |
|                                       |      |                      |          |
|                                       |      |                      |          |
| Intensive edit surcharge (+1          | 0%)  |                      | 67       |
| Discount for first-time client (–10%) |      |                      | -67      |
|                                       |      |                      |          |
|                                       |      | Total due:           | US\$ 668 |

Date job returned: 28 December 2018

Thank you for choosing Stallard Scientific Editing. We look forward to working with you again in the future.

Yours sincerely,

Carry Sallard.

Aaron Stallard Managing Editor Stallard Scientific Editing

Bank details Bank name: ANZ Branch name: Trafalgar Street Branch address: 248 Trafalgar Street, Nelson 7010, New Zealand Account name: Stallard Scientific Limited Swift code: ANZBNZ22 Account number: 06-0665-0199616-00 Note: All bank charges should be for your account. Please mention the job number in the payment details. Response to editor:

Dear Dr. Wang et al

Comment 1: Thank you for submitting a revised draft of your manuscript. You have adequately addressed the major science concerns that were raised by reviewers, but before the manuscript can be accepted for publication in The Cryosphere it will need careful editing for English grammar and consistency.

R: This manuscript has been edited by the Stallard Scientific Editing company (https://www.stallardediting.com). Although we have paid for this manuscript, we only

10 get the invoice of this manuscript instead of the statement due to the new year's vacation (See Author's Response in Page 1).

Comment 2: The abstract states that "Although the mineral dust was assumed to be the highest contributor to the mass loading of ILAPs...", and yet the prior sentence reports

15 larger mean mass concentrations for OC than for mineral dust. So why is it assumed that mineral dust is the highest contributor to mass loadings? These two sentences seem inconsistent. This is one example of why the manuscript needs careful examination from multiple people to ensure that the results are reported consistently.

R: Corrected as suggested.

20

30

Comment 3: Table 1 is cut off at the edge of the page and cannot even be read in its entirety.

R: Corrected as suggested.

25 Comment 4: English grammar:R: See comment 1.

Comment 5: The paper needs careful editing by one or more people who are fluent in English. Readers will discover grammatical errors starting with the first sentence of the Introduction.

R: See comment 1.

Comment 6: Data sharing policy: Please read the journal's data policy at: https://www.the-cryosphere.net/about/data\_policy.html. Prior to publication, your data should be posted in a publicly-accessible repository, and a link to the data should be provided in the Data Availability section of the manuscript. This is particularly

5 important for new measurements such as those reported here.R: We have provided the data sharing contents as suggested.

10

I do not plan to send out your manuscript again for additional peer review. After your manuscript is \*carefully\* cleaned up, I expect we will be able to publish it in The Cryosphere.

|    |                                                                                                                                                                                     | 【样式定义 [1]                                                                          |
|----|-------------------------------------------------------------------------------------------------------------------------------------------------------------------------------------|------------------------------------------------------------------------------------|
|    | Quantifying the light absorption and source attribution of                                                                                                                          | 带格式的: 左侧: 3.17 厘米, 右侧: 3.17 厘米                                              |
|    | insoluble light-absorbing particles on Tibetan Plateau glaciers                                                                                                                     | 带格式的:字体颜色:黑色,英语(英国)                                                                |
|    | between 2013 and 2015                                                                                                                                                               | 带格式的:字体颜色:黑色,英语(英国)                                                                |
|    | Xin Wang 1 , Hailun Wei 1 , Jun Liu 1 , Baiqing Xu 1,2 , Mo Wang 2 , Mingxia Ji 1 , and Hongchun Jin 3 | 带格式的:字体颜色:黑色,英语(英国)                                                                |
| E  | Kay Jaharatary for Sami Arid Climate Change of the Ministry of Education, Collage of Atmospheric                                                                                    | 删除的内容: from                                                                        |
| 5  | Key Laboratory for Senii-Arid Chinate Change of the Ministry of Education, Conege of Autospheric                                                                                    | 带格式的:字体颜色:黑色,英语(英国)                                                                |
|    | Sciences, Lanzhou University, Lanzhou, 730000, China
2 Key Laboratory of Tibetan Environment Changes and Land Surface Processes. Institute of Tibetane                | 删除的内容:-                                                                            |
|    | Rey Eaboratory of Thetan Environment changes and Eand Surface Treesses, institute of Thetan                                                                                         | 带格式的:字体颜色:黑色,英语(英国)                                                                |
|    | Plateau Research, Chinese Academy of Sciences, Beijing 100085, China                                                                                                                | 带格式的:字体:小四,字体颜色:黑色,英语(英                                                            |
|    | 3 KuWeather Science and Technology, Haidian, Beijing, 100085, China                                                                                                      |                                                                                    |
|    | Corresponding outports V. Wang (unit @law edu en) and P. Yu (baising@itness.ac.en)                                                                                                  | (带格式的 [2])                                                                         |
|    | corresponding autions; A. wang (wxin@izu.edu.cr) and B. Xu (baiqing@itpcas.ac.cn)                                                                                                   | (带格式的:字体颜色:黑色,英语(英国)                                                               |
|    |                                                                                                                                                                                     | (新格式的: 字体: (默认) Times New Roman, (中
文) +西文正文 (Calibri), 五号, 字体颜色: 黑色,
英语(英国) |
体颜色: 黑色, 英语(英国)                               |
(英国)                                                     |

[revised manuscript text omitted]

**1** Introduction**

[revised manuscript text omitted]

| /删      | 除的内容: is mainly originated           |                 |
|---------|--------------------------------------|-----------------|
| 带       | 格式的                                  | [156]           |
| /删      | 除的内容: based on a positive matrix     | [158]           |
| 带       | 格式的                                  | [157]           |
| 带       | 格式的                                  | [159]           |
| 删       | 除的内容: found                          |                 |
| 带       | 格式的                                  | [160]           |
| 删       | 除的内容: based on the chemical and c    | ptical
[161] |
| 带       | 格式的                                  | [162]           |
| 删       | 除的内容: Up to                          |                 |
| 一册      | 除的内容: understand.                    |                 |
| 带       | 格式的                                  | [163]           |
| ∖│删     | 除的内容: the                            |                 |
| 带       | 格式的                                  | [164]           |
| ∭删      | 除的内容: of ILAPs remain                |                 |
| 带       | 格式的                                  | [165]           |
| 删       | 除的内容: [Please check that this is you | ur
[166]     |
| 带       | 格式的                                  | [167]           |
| 删       | 除的内容: -                              |                 |
| 带       | 格式的                                  | [168]           |
| 删       | 除的内容: measurements                   |                 |
| 带       | 格式的                                  | [169]           |
| 〔删      | 除的内容: most urgent task to explore t  | he              |
| 带       | 格式的                                  | [170]           |
| 〔删      | 除的内容:, especially in the TP regions  | s. Here         |
| 带       | 格式的                                  | [171]           |
| 删       | 除的内容: performed a large survey on    | [172]           |
| 带       | 格式的                                  | [173]           |
| 删       | 除的内容: on                             |                 |
| 带       | 格式的                                  | [174]           |
| 删       | 除的内容: in the TP regions              |                 |
| 删       | 除的内容: monsoon                        |                 |
| 带       | 格式的                                  | [176]           |
| 删       | 除的内容: non-monsoon                    |                 |
| 带       | 格式的                                  | [175]           |
| 带       | 格式的                                  | [177]           |
| 删       | 除的内容: from 2013-2015.                |                 |
| 带       | 格式的                                  | [178]           |
| 删       | 除的内容:/                               |                 |
| 删       | 除的内容:                                | [179]           |
| 带       | 格式的                                  | [180]           |
| الملك 📗 | 除的内容: associated                     |                 |

Kunlun Mountains of the TP (Fig. 1b), OM Glacier has a length of 2.6 km and an area of 1.73 km2.

Samples 23-32 were collected from the northern part of Meikuang (MK) Glacier (35°42'N,

5 94°12′E), located in the eastern Kunlun Mountains, during both the wet and the dry seasons. This region is characterised by alluvial deposits and sand dunes. MK Glacier is 1.8 km Jong and 1.1 km2 in area (Fig. 1c), Immediately east of MK Glacier, samples 33–44 were collected from the southwestern reaches of Yuzhufeng (YZE) Glacier (35°38′N, 94°J3′E), located on the highest peak (6178 m) of the eastern Kunlun Mountains, This high altitude region is characterised by a

10 cold, arid climate and by fern, forest, and scrubby vegetation.

Samples 45–49 were obtained from the centre of Hariqin (HRQ) Glacier (33°14'N, 92°09'E), a north-facing system located on the northern flank of the Tanggula Mountains, central Qinghai– Tibetan Plateau (Fig. 1e). HRQ Glacier drops from an elevation of 5820 m a.s.l. to its terminus at

15 5400 m, where it forms the headwaters of the Dongkemadi River. To the southwest of HRQ Glacier, the 2.8-km-long Xiaodongkemadi (XD) Glacier (33°04'N, 92°04'E) covers an area of 1.77, km2 and descends from 5900 m elevation to its terminus at 5500 m (Fig. 1f). The surrounding landscape is predominantly cold steppe and tundra. Samples 50–60 were collected from the southern reaches of XD glacier.

20

25

30

Gurenhekou (GR) Glacier (30°19'N, 90°46'E) is a relatively small (area: 1.4 km2, length, 2.5 km; width; 0.6 km) cold-based alpine glacier located approximately 90 km north of Lhasa in southern Tibet (Fig. 1g). The glacier ranges in elevation from 6000 m to its terminus at 5600 m. Both Kang et al. (2009) and Bolch et al. (2010) suggested that GR is influenced by both the continental climate of central Asia and the Indian monsoon system. Samples 61–67 were collected from the eastern part of the glacier.

According to Wang et al. (2015), the mean annual accumulation of snow/ice at our TP drilling sites is approximately 2 m. Therefore, for each glacier sampled between 2013 and 2015, we used a 1.2-m-long vertical tube lined with a clean, 20-cm-diameter plastic bag to collect ice deposited via

| 带格式的                                                 | [238]              |
|------------------------------------------------------|--------------------|
| 删除的内容: Qinghai-Tibet Plateau (Fig                    | . 1b) The
[239] |
| ≻
删除的内容: <mark>[In other cases, you use 'T</mark> | ibetan
2411     |
| 带格式的                                                 | [242]              |
| 带格式的                                                 | [240]              |
| 带格式的                                                 | [243]              |
| 带格式的                                                 | [244]              |
| 带格式的                                                 | [245]              |
| 带格式的                                                 | [246]              |
| 带俗九 り
一
删除的内容:-                         | [247]              |
| ₩除的内突: in                                            |                    |
| 一————————————————————————————————————                | [249]              |
| 删除的内容: glacier during both monso                     | on and $2501$      |
| 一—————————————————————————————————————               | [230]              |
| 带格式的                                                 | [251]              |
| 带格式的                                                 | [252]              |
| 带格式的                                                 | [253]              |
| 带格式的                                                 | [254]              |
| 带格式的                                                 | [255]              |
| 带格式的                                                 | [256]              |
| 带格式的                                                 | [257]              |
| 带格式的                                                 | [258]              |
| 带格式的                                                 | [259]              |
| 删除的内容:                                               | [260]              |
| 带格式的                                                 | [261]              |
| 带格式的                                                 | [262]              |

... [263]

both wet and dry deposition (Fig. 2). Owing to their relatively high altitude, wet deposition over, these glaciers is dominated by fresh snowfall, with considerably less derived from rainfall. Nonetheless, the majority of samples consist of ice rather than snow, reflecting the prevalence of multiple melting processes. Following collection, ice samples were maintained at a temperature

5 of -20 °C during transportation to the State Key Laboratory of Cryospheric Sciences, Cold and Arid Regions Environmental and Engineering Research Institute in Lanzhou, China.

In the laboratory, samples were cut vertically into four pieces following established cleansampling protocols (Fig. S2), after which one of the four pieces was cut at 10-cm resolution. Where

- 10 multiple melting events have produced a non-uniform surface layer (e.g., sites 13 and 26), we cut samples to be longer or shorter than the average. Any dirty layers were cut and analysed separately. A total of 189 samples were used in this study. To minimise the loss of ILAPs to the container walls, each sample was placed in a clean glass beaker and melted quickly in a microwave oven, immediately after which the water was filtered through Nuclepore filters (pore size 0.2, µm)
- 15 following the procedure reported in Doherty et al. (2010). Further details of the filtration process are given in Wang et al. (2013) and Doherty et al. (2014),

**2.2 Optical analysis**

To calculate the mass-mixing ratio of BC in our samples, we employed an updated

- 20 integrating sphere/integrating sandwich spectrophotometer (ISSW). Although this instrument is similar to that developed by Grenfell et al. (2011), a chief difference is that we used two integrating spheres to reduce diffuse radiation during measurement instead of the integrating sandwich diffuser employed by those authors. The ISSW spectrophotometer measures the light-attenuation spectrum from 400 to 700 nm, with the total light-attenuation spectrum being, extended by linear
- 25 extrapolation to cover the full spectral range (300-750 nm). Nominally, light attenuation is sensitive solely to ILAPs trapped on the filter as a result of the diffuse radiation field and the sandwich structure of the two integrated spheres in the ISSW (Doherty et al., 2014). Specifically, the system detects the light transmitted by an ice sample,  $S(\lambda)$ , and compares this value to that transmitted by a blank filter,  $S_0(\lambda)$ . The relative attenuation (Atn) is then expressed as;

7

|----|------------------------------------------------------|
| 1  | /
|    | 带借政的 [343]                                           |
|    | - 一方方子 areas were predominant                        |
|    |                                                      |
|    | 删陈的闪容: new fallen snow, while much                   |
[353]      |
[362] |

**$Atn = \ln[S_0(\lambda)/S(\lambda)]$**

(1)

|    | The mass absorption efficiency (MAE), and absorption Ångström exponents (Å) employed here                                                                                             |
|----|---------------------------------------------------------------------------------------------------------------------------------------------------------------------------------------|
|    | for BC, OC, and Fe are described in detail by Wang et al. (2013). Using this technique, we are                                                                                        |
| 5  | able to estimate the following parameters, equivalent BC ( $C_{BC}^{equiv}$ ), maximum BC ( $C_{BC}^{max}$ ), estimated                                                               |
|    | BC ( $C_{BC}^{est}$ ), the fraction of light absorption by non-BC ILAPs ( $f_{non-BC}^{est}$ ), the absorption Ångström                                                               |
|    | exponent of non-BC ILAPs $(\hat{A}_{non-BC})$ , and the total absorption Angström exponent $(\hat{A}_{tot})$ . These                                                                  |
|    | parameters are defined as follows:                                                                                                                                                    |
|    | 1. <math>C_{BC}^{equiv}</math> (ng g-1): equivalent BC is the amount of BC that would be needed to produce                                                          |
| 10 | absorption by all insoluble particles in snow for wavelengths of 300-750 nm.                                                                                                          |
|    | $2_{\rm r} C_{BC_{\star}}^{max}$ (ng $g_{\star}^{-1}$ ): maximum BC is the maximum possible BC mixing ratio in snow, assuming that                                                    |
|    | all light absorption is due to BC at wavelengths of 650, 700 nm.                                                                                                                      |
|    | $\underline{3}_{\mathfrak{m}} C_{BC_{\mathfrak{m}}}^{est}$ (ng $\underline{g}_{\mathfrak{m}}^{-1}$ ): estimated BC is the estimated true mass of BC in snow derived by separating the |
|    | spectrally resolved total light absorption and non-BC fractions,                                                                                                                      |
| 15 | $4.f_{non-BC}^{est}$ (%): the fraction of light absorption by non-BC light-absorbing particles is the                                                                                 |
|    | integrated absorption due to non-BC light-absorbing particles. This value is weighted by                                                                                              |
|    | the down-welling solar flux at wavelengths of 300-750 nm.                                                                                                                             |
|    | 5. Ånon-BC: non-BC absorption Ångström exponent, derived from the light absorption by                                                                                                 |
|    | non-BC components for wavelengths of 450-600 nm.                                                                                                                                      |
| 20 | $\int_{\Omega} A_{tot}$ : absorption Angström exponent calculated for all insoluble particles deposited on the filter                                                                 |
|    | between 450 and 600 nm.                                                                                                                                                               |
|    |                                                                                                                                                                                       |
|    | Both the composition and the size distribution of aerosols are well-known parameters influencing                                                                                      |
|    | the absorption Ångström exponent, Doherty et al. (2010) reported that the absorption                                                                                                  |
| 25 | Ångström exponent of OC is close to 5, consistent with the previously reported range of 4-6                                                                                           |
|    | (Kirchstetter et al., 2004), and several studies have included absorption Ångström exponents of 2-                                                                                    |
|    | 5 for MD (Fialho et al., 2005; Lafon et al., 2006; Zhou et al., 2017; Cong et al., 2018). Typical                                                                                     |

absorption Ångström exponents for urban and industrial fossil fuel emissions fall within the range

1.0-1.5 (Millikan, 1961; Bergstrom et al., 2007), which is slightly lower than that of biomass

30 burning aerosols (1.5-2.5) (Kirchstetter et al., 2004; Bergstrom et al., 2007). In this study, we note

|---|---------------------------------------|-------------------|
| λ | 带格式的                                  | [399]             |
| / | 带格式的                                  | [400]             |
|   | 带格式的                                  | [398]             |
|   | 带格式的                                  | [401]             |
|   | 带格式的                                  | [403]             |
|   | 带格式的                                  | [402]             |
|   | 带格式的                                  | [404]             |
|   | 带格式的                                  | [405]             |
|   | 删除的内容: [Please consider spelling this | s term            |
|   | 带格式的                                  | [408]             |
|   | 带格式的                                  | [406]             |
|   | 带格式的                                  | [409]             |
|   | 带格式的                                  | [410]             |
**= \$6                     | [411]             |**
|   | 带借五时 带格式的                             | [412]             |
|   | 带格式的                                  | [413]             |
|   | 带格式的                                  | [415]             |
|   | 删除的内容: [Your list of parameters in th | is [416]          |
|   | 带格式的                                  | [410]             |
|   | 删除的内容:1                               |  [ · _ · ] |
|   | 带格式的                                  | [418]             |
|   | 带格式的                                  | [419]             |
|   | 带格式的                                  | [420]             |
|   | 带格式的                                  | [421]             |
|   | 带格式的                                  | [422]             |
|   | 带格式的                                  | [423]             |
|   | 带格式的                                  | [424]             |
|   | 带格式的                                  | [425]             |

that the absorption Ångström exponent  $(\hat{A}_{loc})$  comprises both BC and non-BC impurities trapped on the filters. Calculations of  $\hat{A}_{loc}$  and of  $\hat{A}_{non-BC}$  are described by Doherty et al. (2014). Specifically,  $\hat{A}_{non-BC}$  is calculated as a linear combination of the contributions to light absorption made by OC and Fe;

5

 $\dot{A}_{non-BC} = F_{OC} \times \dot{A}_{OC} + F_{Fe} \times \dot{A}_{Fe}$ (2)

**2.3 Chemical analysis**

Major, metallic elements (Al, Cr, Mn, Fe, Ni, Cu, Zn, Cd, and Pb) were analysed on an X-7 10 Thermo Electrical inductively coupled plasma mass spectrometer (ICP\_MS) at the Institute of Tibetan Plateau Research, Beijing, China, The detection limits are 0.238 ng ml-1 for Al, 0.075 ng ml-1/4v for Cr, 0.006 ng ml-1/4v for Mn, 4.146 ng ml-1/4v for Fe, 0.049 ng ml-1/4v for Ni, 0.054 ng ml-1/4v for Cu, 0.049 ng ml-1 for Zn, 0.002 ng ml-1 for Cd, and 0.002 ng ml-1 for Pb. Prior to measurement, melted samples were acidified  $(pH_2 < 2)$  with ultra-pure  $HNO_3$  and left to settle 15 for 48 hours. The relative deviation between most of the measured values and the standard reference values is within 10%. Details of these procedures are given in Li et al. (2009) and Cong et al. (2010), We used a Dionex 320 ion chromatograph to measure major anions (Cl.  $NO_{2a}^{-}NO_{3a}^{-}$  and  $SO_{4}^{2-}$  and cations ( $Na_{a}^{+}NH_{4a}^{+}K_{a}^{+}Mg_{a}^{2+}$  and  $Ca^{2+}$ ) in filtrated water samples. The apparatus, which is housed at the Institute of Tibetan Plateau Research in Beijing, is equipped 20 with a CS12 column for cations and an AS11 column for anions and has a detection limit for all measured ions of  $1 \mu g \cdot l^{-1}$ . We also measured concentrations of Sea, salt MD, and biosmoke K (KBiosmoke) to assess the mass contributions of the major components in our ice samples. Specifically, Sea salt was estimated according to the protocol described by Pio et al. (2007); Sea salt =  $Na_{Ss}^{+} + Cl^{-} + Mg_{S}^{2+} + Ca_{Ss}^{2+} + K_{Ss}^{+} + SO_{4Ss}^{2-}$ 25  $= Na_{Ss}^{+} + Cl^{-} + 0.12 \times Na_{Ssr}^{+} + 0.038 \times Na_{Ssr}^{+} + 0.038 \times Na_{Ssr}^{+} + 0.25 \times Na_{Ssr}^{+} + 0.25 \times Na_{Ssr}^{+} + 0.25 \times Na_{Ssr}^{+} + 0.025 \times Na_{Ssr}^{+} +$ (3)  $Na_{Ss} = Na_{Total} - Al \times (Na/Al)_{Crust}$ (4)

30

|-------------------|-----------------------------------------------|
|                   | ( 带格式的 [474]                                  |

| [530]                                                        |
|--------------------------------------------------------------|
|                                                              |
| [531]                                                        |
| [532]                                                        |
| rials                                                        |
|                                                              |
| [533]                                                        |
| [534]                                                        |
| estimated as                                                 |
| [535]                                                        |
|                                                              |
| [536]                                                        |
| straightforward method, and                                  |
| [538]                                                        |
|                                                              |
| [539]                                                        |
| [540]                                                        |
| [541]                                                        |
|                                                              |
| [542]                                                        |
|                                                              |
| [543]                                                        |
| [544]                                                        |
| [545]                                                        |
| llows                                                        |
| [546]                                                        |
| smoke=K Total -K Dust -K Ss |
| [548]                                                        |
| [549]                                                        |
|                                                              |
|                                                              |
|                                                              |
| [551]                                                        |
|                                                              |

... [550]

... [552]

... [553]

... [554]

... [555]

... [556]

|    | 2.5 Source apportionment                                                                                         |
|----|------------------------------------------------------------------------------------------------------------------|
|    | PMF 5.0 is a receptor model used to determine ILAP source apportionment when source emission                     |
|    | profiles are unavailable (Paatero and Tapper, 1994). We employed a PMF procedure similar to that          |
|    | described by Hegg et al. (2009, 2010), in which mass concentrations and chemical species                         |
| 5  | uncertainties are provided as the input. Our final data set contained 189 samples with 18 elements,              |
|    | only those elements with high recovery were used, for PMF analysis. For each sample, uncertainty                 |
|    | values (Unc) for individual variables were estimated from an empirical equation expressed as;                    |
|    | $Unc = \sqrt{(\sigma \times c)^2 + (MDL^2)} $ (10)                                                               |
|    | Where <math>\sigma</math> is the standard deviation, c represents the mass concentrations of the relative |
| 10 | species, and the MDL depicts the method detection limited.                                                       |
|    | Although we ran the PMF model for between three and six factors, including six random seeds, we                  |
|    | found that the most meaningful results for our TP sites were generated by a three-factor solution.               |
|    | Indeed, Q values (modified values) for this three-factor solution (both robust and true) were closest            |
|    | to the theoretical values of any factor number for which the model was run.                                      |
| 15 |                                                                                                                  |
|    | 3 Results and discussion                                                                                         |
|    | 3.1 Aerosol optical depth                                                                                        |
|    | Aerosol optical depth (AOD) represents both the transport pathways and deposition of dry                         |
|    | aerosols, which in turn provide vital information on potential ILAP sources. As shown in                         |
| 20 | Figure 3, QY, QM, MK, and YZF glaciers are located on the northern TP, whereas XD, HRQ, and                      |
|    | GR glaciers are located in the plateau's southern regions. Therefore, to elaborate on the sources of             |
|    | ILAPs for each TP study site, we assessed the spatial distribution of averaged 500 nm AOD, derived               |
|    | from Aqua-MODIS between 2013 and 2015. According to Ramanathan et al. (2007), anthropogenic                      |
|    | AOD, also referred to as atmospheric brown cloud (ABC) on the south side of the Himalayas, is                    |
| 25 | greater than 0.3. Consequently, AOD (500 nm) values of >0.3 and <0.1 are considered                              |
|    | representative of anthropogenic haze and background conditions, respectively                                     |
|    |                                                                                                                  |
|    | We observed considerably higher $\Lambda OD$ over the western TP than over the central TP. For                   |

We observed considerably higher AOD over the western TP than over the central TP For example, values for QY, QM, MK, and YZF glaciers ranged from 0.25 to 0.3 suggestive of anthropogenic influence, whereas values for HRQ, XD, and GR glaciers were considerably lower

|--------|------------------------------------------------|
| (

(<0.125). Although the elevated AOD over the western TP might serve to enhance glacial retreat there (Engling and Gelencser, 2010), we note that AOD over the TP in general was significantly, lower than in southern Asia, particularly over the Indo-Gangetic Plain during the cold season. This pattern aligns closely with previous measurements (Cong et al., 2009; Ming et al., 2010; Yang et

al., 2012; Lüthi et al., 2015). 5

**3.2 Regional averages of optical parameters**

Table 1 compiles the ice  $C_{BC}^{est}$ ,  $C_{BC}^{max}$ ,  $C_{BC}^{equiv}$ ,  $f_{mon-BC}^{est}$ ,  $\hat{A}_{tot_2}$  and  $\hat{A}_{non-BC}$ , data for each glacier. The Jowest median  $C_{BC}^{est}$  (23–26 ng g-1) was observed on HRQ and GR glaciers, southern TP, during the wet season, whereas the highest values (187-165 ng g-1) occurred on MK and YZF glaciers on

10 the central TP. Relative to the wet season, the measured concentrations of CBC were markedly higher during the dry season for all seven glaciers. The lowest overall BC concentration was recorded on XD Glacier ( $C_{BC}^{est} = \sim 10 \text{ ng g}_{-1}^{-1}$ ), whereas the maximum values of  $C_{BC}^{est}$  (3100 ng g-1).  $C_{BC}^{max}$  (3600 ng g-1), and  $C_{BC}^{equiv}$  (4700 ng g-1) all corresponded to GR Glacier. Median  $\hat{A}_{tot}$  typically exceeded 1.0 at all seven sites (Fig. 4, Table 1).

15

The ice samples exhibited  $A_{tot}$  and  $A_{non-BC}$  values of 1.4-3.7 and 1.9-5.8, respectively (Table S1). As shown in Figure 4a, the median values of  $A_{tot}$  for QY, MK, XD, and GR glaciers were 2.62, 2,64, 2,18, and 2.46, respectively, and the estimated contributions of non-BC ILAPs to absorption were approximately 41%, 44%, 36%, and 48%, respectively. Relatively high values

were observed in samples from QM (2.76), YZF (2.95), and HRQ (2.87) glaciers. Accordingly, the estimated from those regions were 44%, 48%, and 48%, respectively. With the exception of HRQ Glacier, our data set exhibits a clear south-to-north increase in  $A_{pon-BC}$  over the TP (Fig. 4b). Histograms depicting  $A_{tot}$  by region are shown in Figure 5.

25

20

XD Glacier exhibited the greatest degree of  $A_{tot}$  variability, not only in the higher values  $(\sim 2-4)$ , but also at the lower end of the range (<2). This broad distribution is indicative of the complicated sources of particulate light absorption. For instance, Wang et al. (2013) reported that higher  $\underline{A}_{iot}$  values (approximately 3.5-4.5) are strongly correlated with local soils, whereas fossil

30 fuel combustion has an absorption Ångström exponent of  $\leq 2$  (Millikan, 1961; Fialho et al., 2005).

|---|-------------------------------------------------------|
|   | (带格式的)[661]
(#找卡你)                                 |
|   | 带备式的 [662]                                            |
|   | 常俗式的 … [667] 世格式的 □ [667]                             |
|   | ##15,65[668]
|   | #格式的 [670]                                            |
|   |                                                       |

A significant fraction of the total absorption on XD Glacier, therefore, is attributed not only to BC (49%; Fig. 7), but also to non-BC absorbers (51%) linked to OC and MD. In contrast, A tot values for all other sites typically ranged from 2 to 3. The values of Anon-BC and Atot for each site are also given in Figure S3.

5

Figure 6 shows the regional variability in BC, OC, and Fe concentrations during the wet and dry. seasons. Although we observed clear, differences in median and average JLAP concentrations among the seven glaciers, we also note that overall, ILAPs exhibited a similar pattern of variability throughout our study area. With the exception of QY and QM glaciers, we collected ice samples

10 during both the wet and the dry seasons. On average, BC and OC concentrations at HRQ, XD, and GR glaciers were several orders of magnitude higher during the dry season than during the wet season. This pattern is consistent with the findings from the middle Himalayas of Cong et al. (2015), who reported that the dry season is characterised by a distinctly higher carbonaceous aerosol level than that of the wet season, despite similar air mass pathways.

15

Lüthi et al. (2015) demonstrated that the atmospheric brown cloud over Southern, Asia can cross the Himalayas, transporting polluted air masses to the TP and potentially impacting regional glacier mass balance. In our data set, however, there is no apparent difference in JLAP mixing ratios between the wet and dry seasons for two adjacent (MK and YZF) glaciers. We

- attribute this pattern to the fact that with the exception of long-range pathways, local air pollutants 20 can also impact ILAP availability on the central TP. For instance, although the prevailing air masses over the MK and YZF glaciers originate from the arid western TP and Taklimakan Desert. regions, Huang et al. (2018) concluded that the concentration of trace elements at YZF Glacier, and thus that YZF Glacier is less influenced by human activity. In close agreement with Ming et al.
- (2013), our median values of  $C_{BC}^{est}$  and  $C_{OC}$  (referred to as the mass concentration of OC) exhibit a 25 gradually decreasing trend from north to south, and the mass concentrations of BC are higher for northern TP glaciers than for their southern counterparts,

To help quantify the regional ILAP status of each glacier, Table 2 contains statistics on snow and 30 ice samples collected both during our present investigation and during previous studies of TP

13

|---|-----------------------------------------------|
|   | 【带格式的[763]                                    |
|   | ( 带格式的 [764]                                  |
|   | · 〒1日本山 · · · · · · · · · · · · · · · · · · · |

|---|----------------------------------------------------|
|   |                                                    |
|   |                                                    |
|   |                                                    |
|   | [647]
|   | 则必历中空: 41                                          |

glaciers. During our visit to YZF Glacier, we collected twelve ice samples from depths between 15 and 45 cm (Table S1). As shown in Figure S4,  $C_{BC,values}^{est}$  this region typically ranged from ~100 to 1000 ng  $g_{10}^{-1}$  with several values of <100 ng  $g_{10}^{-1}$  A striking feature of this data set is the relatively high  $C_{BCA}^{max}$  (1600 ng g-1) and  $C_{OCA}$  (9160 ng g-1) in the surface layer at site 41.

Judging by the high value of  $\int_{non-BC}^{est} (0.56)$  for this site, we suggest that these data indicate that light 5 absorption at this site is influenced not only by BC but also potentially by OC and MD

For YZF Glacier,  $A_{tot}$  typically varied between ~2 and 3.7, and the average  $f_{non-BC}^{est}$  close to 50%, which together suggest that ILAPs at this site are heavily influenced by anthropogenic air

10 pollution. We also observed large variations in  $C_{OC}$  with values ranging from ~10 to 17,000 ng  $g_{AC}^{=1}$  With the exception of site 23,  $C_{BC_{A}}^{est}$  values for MK Glacier were considerably lower than those of YZF Glacier (range 20-670 ng g-1, median 130 ng g-1, Fig. S5). MK Glacier gave a median  $C_{OC,Of} \sim 600 \text{ ng g}_{A}^{-1}$  whereas the fraction of total particulate light absorption attributable to non-BC constituents was typically  $\sim 16\% - 62\%$ ,  $A_{non-BC}$  (5.12) at this site is very similar to that of YZF

15 Glacier, (5.06).

> $\mathcal{L}_{BC}^{est}$  values for QY Glacier (Fig. S6) are similar to those of MK Glacier, ranging from ~20 to 720 ng  $g_{-1}^{-1}$  (excluding the highest value of 1900 ng  $g_{-1}^{-1}$  at site 13). The fraction of total particulate light absorption due to the non-BC constituent  $f_{non-BC}^{est}$  was typically ~20%-70%, with a median

- value of 41%. Together with the lower  $A_{tot}$  (2.6), this information indicates that BC plays a 20 dominant role in influencing light absorption in this region. Compared with the other TP glaciers, we note that the vertical ILAP profiles on QY Glacier were collected during the 2014 and 2015 wet seasons (Table S1). The mixing ratios of OC and Fe were 80-10,100 ng  $g_{11}^{-1}$  and 20-340 ng  $g_{12}^{-1}$ respectively. Figure S7 shows that the vertical profiles of the mass-mixing ratios of BC, OC, and
- Fe were more variable for XD Glacier than for the other six glaciers. With the exception of the 25 surface layer at sites 53 and 54,  $C_{BC}^{est}$  typically ranged from 10 to 280 ng  $g_{e}^{-1}$ , indicating that XD Glacier is the cleanest site in our study. At sites 56–58,  $\int_{non-BC}^{est}$  was less than 38%, and  $\hat{A}_{tot}$  ranged from 1, to 2.5, consistent with the combustion of fossil fuels due to industrial activity,

**3.3 Scavenging and washing efficiencies 30**

Previous studies have demonstrated how ILAPs become trapped and integrated into the snowpack as a result of melting and sublimation, thereby enriching surface concentrations of these particles (Conway et al., 1996; Painter et al., 2012; Doherty et al., 2013). For instance, Doherty et al. (2013) reported that JLAP scavenging by snow meltwater Jeads to elevated concentrations of BC in the

- 5 surface layer. Similarly, Flanner et al. (2007, 2009) concluded that amplified ablation due to the concentration of BC in melting snow serves to further reduce the snow albedo, thus providing a positive feedback to radiative forcing. However, the impact of multiple melting processes on ILAPs located at greater depths in the glacier surface, remains unclear,
- 10 On QY Glacier, we observed a marked increase in ILAP mixing ratios with depth. Although this result may appear inconsistent with those of Doherty et al. (2013), we note that Xu et al. (2012) observed high concentrations of BC at the snow surface and at depth, which those authors attributed to meltwater percolation and the deposition of superimposed ice in the snowpack. A further prominent feature in our data set is the elevated surface
- 15 mixing ratio of  $C_{BC}^{est}$  at sites 52–54 on XD Glacier, relative to deeper layers, which we attribute to the dry/wet deposition of BC on the surface samples. We propose that the clear difference in vertical profiles between QY and XD glaciers is a function of ILAP deposition. Specifically, QY Glacier was sampled during the wet season, when higher temperatures and stronger melting potentially serve to concentrate ILAPs in the basal layers. In contrast, because we
- 20 sampled XD Glacier during both the wet and dry seasons, ILAP concentrations decrease with depth during the dry season as a function of scavenging (Figs. S7a–g) but increase during the wet season because of the concentration effect (Figs. S7h-i), Because the single-layer samples are not shown, the vertical profiles of  $C_{BC}^{est}$  for QM, HRQ, and GR glaciers are plotted in Figure S8. With the exception of those sites included in Figure S8d–e, i, and h, the sampled glaciers exhibit the
- 25 trapping and scavenging effects of a higher surface-layer BC content resulting from melting processes.

**3.4 JLAP contributions to particulate light absorption,**

The fractional contributions of BC, OC, and Fe (presumably in the form of goethite) to total absorption (450 nm) are depicted for each glacier in Figure 7, with further details of BC, OC, and

|-----------------------------|--|
|                             |  |

|-------------------|----------------------------------------------------------|
1931] |
|                   | 已移动(插入) [3]                                              |
|                   | 已上移 [3]: the vertical profiles of $C_{BC}^{est}$         |
|                   | መስታ ( )                                                  |
|                   | (一) 带格式的 [0/1]                                           |

Fe concentrations given in Table S1. BC plays a dominant role in particulate light absorption, with average values ranging from ~44% to 54% across all seven glacier sites. Although OC represents the second highest absorber, we noted significant variability (between 25% and 46% on average) in its contribution to total light absorption during the 2013–2015 field campaign. For those glaciers located on the eastern TP (QY, YZF, and HRQ glaciers), the relative contributions of BC and OC

5 located on the eastern TP (QY, YZF, and HRQ glaciers), the relative contributions of BC and OC to total absorption are broadly similar. The highest fraction of BC (54%) was measured on QM Glacier, on the western TP.

Complementing the BC and OC contributions, light absorption on TP glaciers is also
 influenced by Fe. According to our data, the average fraction of total light absorbed by Fe ranges from approximately 11% to 31% across all seven glaciers, with the highest values recorded on GR Glacier. This finding indicates that MD plays a key role in the spectral absorption properties of ILAPs on TP glaciers. The relative contributions of BC, OC, and Fe to total light absorption for all surface-ice samples are presented in Figure S9 and Table

**15 1.**

**3.5 Enrichment factor**

EF values ranging from 0.1 to 10 represent significant input from crustal sources, whereas values of >10 indicate major contributions from anthropogenic activity. According to our EF analysis (Fig.

- 20 §), mean values for Fe are less than 5 for all seven glaciers, suggesting a primarily crustal origin. This result supports the findings of previous studies in northern China (Wang et al., 2013) and North America (Doherty et al., 2014), which indicate that light-absorbing particles in snow are dominated by local soil dust. Similar to Fe, other trace metals with mean EF values of ≥5.0 are moderately to highly enriched because of anthropogenic emissions (Hsu et al., 2010). For example,
- 25 Pacyna and Pacyna (2001) reported that Cr is derived chiefly from the combustion of fossil fuels, which is also a primary source of Cu Pb and Zn however, are Jinked to traffic-related combustion and coal burning (Christian et al., 2010; Contini et al., 2014). In summary, the high EF values for Cu, Zn, and Cd in our ice samples provide clear evidence that TP glaciers are being affected by anthropogenic pollution.

30

|-----------------------------------------------------------------------------------------------------------------------------------------------------------------------------------------------------------------------------------------------------------------------------------------------------------------------------------------------------------------------------------------------------------------------------------------------------------------------------|
… [958]                                                                                                                                                                                                                                                                                                                                                                                                                         |
… [963]                                                                                                                                                                                                                                                                                                                                                                                                                          |
|                                                                                                                                                                                                                                                                                                                                                                                                                                                                             |

[976]

[revised manuscript text omitted]

|------------------|-------------------------------------------|----------------------|
| 1                | 带格式的                                      | [1103]               |
|                  | 带格式的                                      | [1104]               |
|                  | 带格式的                                      | [1105]               |
| $\left  \right $ | 带格式的                                      | [1106]               |
|                  | 带格式的                                      | [1107]               |
|                  | 带格式的                                      | [1108]               |
|                  | 带格式的                                      | [1109]               |
|                  | 带格式的                                      | [1110]               |
|                  | 带格式的                                      | [1111]               |
|                  | 带格式的                                      | [1112]               |
|                  |                                           | [1113]               |
|                  |                                           | [1114]               |
|                  | 删除的内容: traction in the YZF glacier.       | [1115]               |
|                  |                                           | [1116]               |
|                  |                                           | 54.4.4 77            |
|                  |                                           | [1117]               |
|                  | ////////////////////////////////////      | [4440]               |
|                  |                                           | [1118]               |
|                  | 加际时内存.weit                                |                      |
|                  | 【带格式的
(带格式的                            | [1119]               |
|                  | 甲伯 八印
刪除的由密: in the V7E and MK glacier | [1120]
rs. In the |
|                  |                                           |                      |
|                  |                                           | [1121]               |
|                  | ШРП ЦЦЦЦЦ
刪险的内突: glacier               | [1122]               |
|                  | 带格式的                                      | [1100]               |
|                  | ー H C C C C C C C C C C C C C C C C C C   | [1123]               |
|                  | 带格式的                                      | [110/]               |
|                  | 删除的内容: was up to                          | [±±೭4]               |
|                  | 带格式的                                      | [1125]               |
|                  |                                           |                      |

anthropogenic pollutants. The largest contributors of light-absorbing insoluble particles for TP glaciers, however, include local MD and industrial pollution sources, followed by the burning of biomass. In summary, both natural MD and anthropogenic emissions constitute non-negligible sources of ILAPs for TP glaciers,

5

Data availability. All datasets and codes used in this study can be obtained by contacting Xin Wang (wxin@lzu.edu.cn).

The Supplement related to this article is available online at https://XXXX-supplement.

10

Author contributions. BX and MW designed the experiments. XW prepared the manuscript with contributions from all co-authors.

Competing interests. The authors declare that they have no conflicts of interest.

15

Acknowledgements. This research was supported by the National Key Research and Development Program on Monitoring, Early Warning and Prevention of Major Natural Disaster (2018YFC1506005), the National Natural Science Foundation of China (grants 41775144, 41522505, 41771091, 41675065 and 41875091), and the Fundamental Research Funds for the

20 Central Universities (lzujbky-2018-k02).

Edited by: Mark Flanner Reviewed by: two anonymous referees

|                                                                                                                                                                                                                                                                                                                                                                                                                                                                                                                                                                                                                                                                                                                                                                                                                                                                                                                                                                                                                                                                                                                                                                                                                                                                                                                                                                                                                                                                                                                                                                                                                                                                                                                                                                                                                                                                                                                                                                                                                                                                                                                                | 删除的内容: the light absorption by     |                 |
|--------------------------------------------------------------------------------------------------------------------------------------------------------------------------------------------------------------------------------------------------------------------------------------------------------------------------------------------------------------------------------------------------------------------------------------------------------------------------------------------------------------------------------------------------------------------------------------------------------------------------------------------------------------------------------------------------------------------------------------------------------------------------------------------------------------------------------------------------------------------------------------------------------------------------------------------------------------------------------------------------------------------------------------------------------------------------------------------------------------------------------------------------------------------------------------------------------------------------------------------------------------------------------------------------------------------------------------------------------------------------------------------------------------------------------------------------------------------------------------------------------------------------------------------------------------------------------------------------------------------------------------------------------------------------------------------------------------------------------------------------------------------------------------------------------------------------------------------------------------------------------------------------------------------------------------------------------------------------------------------------------------------------------------------------------------------------------------------------------------------------------|------------------------------------|-----------------|
| Å                                                                                                                                                                                                                                                                                                                                                                                                                                                                                                                                                                                                                                                                                                                                                                                                                                                                                                                                                                                                                                                                                                                                                                                                                                                                                                                                                                                                                                                                                                                                                                                                                                                                                                                                                                                                                                                                                                                                                                                                                                                                                                                              | 带格式的                               | [1186]          |
| 4                                                                                                                                                                                                                                                                                                                                                                                                                                                                                                                                                                                                                                                                                                                                                                                                                                                                                                                                                                                                                                                                                                                                                                                                                                                                                                                                                                                                                                                                                                                                                                                                                                                                                                                                                                                                                                                                                                                                                                                                                                                                                                                              | 删除的内容: in                          |                 |
| -(                                                                                                                                                                                                                                                                                                                                                                                                                                                                                                                                                                                                                                                                                                                                                                                                                                                                                                                                                                                                                                                                                                                                                                                                                                                                                                                                                                                                                                                                                                                                                                                                                                                                                                                                                                                                                                                                                                                                                                                                                                                                                                                             | 带格式的                               | [1187]          |
| 1                                                                                                                                                                                                                                                                                                                                                                                                                                                                                                                                                                                                                                                                                                                                                                                                                                                                                                                                                                                                                                                                                                                                                                                                                                                                                                                                                                                                                                                                                                                                                                                                                                                                                                                                                                                                                                                                                                                                                                                                                                                                                                                              | 删除的内容: originated from the         |                 |
|                                                                                                                                                                                                                                                                                                                                                                                                                                                                                                                                                                                                                                                                                                                                                                                                                                                                                                                                                                                                                                                                                                                                                                                                                                                                                                                                                                                                                                                                                                                                                                                                                                                                                                                                                                                                                                                                                                                                                                                                                                                                                                                                
| Ý                                                                                                                                                                                                                                                                                                                                                                                                                                                                                                                                                                                                                                                                                                                                                                                                                                                                                                                                                                                                                                                                                                                                                                                                                                                                                                                                                                                                                                                                                                                                                                                                                                                                                                                                                                                                                                                                                                                                                                                                                                                                                                                              | 带格式的                               | [1189]          |
| N                                                                                                                                                                                                                                                                                                                                                                                                                                                                                                                                                                                                                                                                                                                                                                                                                                                                                                                                                                                                                                                                                                                                                                                                                                                                                                                                                                                                                                                                                                                                                                                                                                                                                                                                                                                                                                                                                                                                                                                                                                                                                                                              | 删除的内容: biomass                     |                 |
|                                                                                                                                                                                                                                                                                                                                                                                                                                                                                                                                                                                                                                                                                                                                                                                                                                                                                                                                                                                                                                                                                                                                                                                                                                                                                                                                                                                                                                                                                                                                                                                                                                                                                                                                                                                                                                                                                                                                                                                                                                                                                                                                | 删除的内容: source. Therefore, the      |                 |
|                                                                                                                                                                                                                                                                                                                                                                                                                                                                                                                                                                                                                                                                                                                                                                                                                                                                                                                                                                                                                                                                                                                                                                                                                                                                                                                                                                                                                                                                                                                                                                                                                                                                                                                                                                                                                                                                                                                                                                                                                                                                                                                                | 带格式的                               | [1188]          |
|                                                                                                                                                                                                                                                                                                                                                                                                                                                                                                                                                                                                                                                                                                                                                                                                                                                                                                                                                                                                                                                                                                                                                                                                                                                                                                                                                                                                                                                                                                                                                                                                                                                                                                                                                                                                                                                                                                                                                                                                                                                                                                                                | 带格式的                               | [1190]          |
|                                                                                                                                                                                                                                                                                                                                                                                                                                                                                                                                                                                                                                                                                                                                                                                                                                                                                                                                                                                                                                                                                                                                                                                                                                                                                                                                                                                                                                                                                                                                                                                                                                                                                                                                                                                                                                                                                                                                                                                                                                                                                                                                | 带格式的                               | [1191]          |
|                                                                                                                                                                                                                                                                                                                                                                                                                                                                                                                                                                                                                                                                                                                                                                                                                                                                                                                                                                                                                                                                                                                                                                                                                                                                                                                                                                                                                                                                                                                                                                                                                                                                                                                                                                                                                                                                                                                                                                                                                                                                                                                                
|                                                                                                                                                                                                                                                                                                                                                                                                                                                                                                                                                                                                                                                                                                                                                                                                                                                                                                                                                                                                                                                                                                                                                                                                                                                                                                                                                                                                                                                                                                                                                                                                                                                                                                                                                                                                                                                                                                                                                                                                                                                                                                                                | 带格式的                               | [1192]          |
|